# The contrasted phytoplankton dynamics across a frontal system in the southwestern Mediterranean Sea

Roxane Tzortzis[1], Andrea M. Doglioli[1], Monique Messié[2], Stéphanie Barrillon[1], Anne A. Petrenko[1], Lloyd Izard[3], Yuan Zhao[4], Francesco d'Ovidio[3], Franck Dumas[5], and Gérald Gregori[1]

[1]Aix Marseille Univ., Université de Toulon, CNRS, IRD, MIO UM 110, 13288, Marseille, France
[2]Monterey Bay Aquarium Research Institute, Moss Landing, CA, United States
[3]Sorbonne Université, CNRS, IRD, MNHN, Laboratoire d'Océanographie et du Climat: Expériments et Approches Numériques (LOCEAN-IPSL), Paris, France
[4]CAS Key Laboratory of Marine Ecology and Environmental Sciences, Institute of Oceanology, Chinese Academy of Sciences, Qingdao, People's Republic of China
[5]SHOM, Service Hydrographique et Océanographique de la Marine, 13 rue de Chatellier, CS592803, 29228 Brest, CEDEX 2, France

**Correspondence:** Roxane TZORTZIS (roxane.tzortzis@mio.osupytheas.fr)

**Abstract.**

Numerical simulations have shown that finescale structures such as fronts are often suitable places for the generation of vertical velocities, transporting subsurface nutrients to the euphotic zone and thus modulating phytoplankton abundance and community structure. In these structures, direct in situ estimations of the phytoplankton growth rates are rare ; although difficult to obtain, they provide precious information on the ecosystem functioning. Here, we consider the case of a front separating two water masses characterized by several phytoplankton groups with different abundances, in the southwestern Mediterranean Sea. In order to estimate possible differences in growth rates, we measured the phytoplankton diurnal cycle in these two water masses as identified by an adaptive and Lagrangian sampling strategy. A size-structured population model was then applied to these data to estimate the growth and loss rates for each phytoplankton group identified by flow cytometry, showing that these two population parameters are significantly different on the two sides of the front, and consistent with the relative abundances. Our results introduce a general method for estimating growth rates at frontal systems, paving the way for in situ exploration of finescale biophysical interactions.

## 1 Introduction

Phytoplankton forms the basis of the marine food web (Sterner and Hessen, 1994) and it is responsible for half of the primary production of the planet (Field et al., 1998), while its biomass is only $\leq 1$ % of the global biomass (Winder and Cloern, 2010). Thanks to photosynthesis, phytoplankton fuels the ocean and the atmosphere in free $O_2$ and it fixes and exports the $CO_2$ into the ocean depth (Field et al., 1998; De La Rocha and Passow, 2007). This process called biological carbon pump is critical for global ocean sequestration of carbon and therefore for the modulation of atmospheric $CO_2$. The biological carbon pump is modulated by the size structure of the phytoplankton community. Small or large phytoplankton species are associated

with different efficiencies for particle export, remineralization, and transfer to the deep ocean (Boyd and Newton, 1999; Guidi et al., 2009; Hilligsøe et al., 2011; Mouw et al., 2016, etc). That is why, it is primordial to understand the factors that rule phytoplankton abundance and diversity.

We use here the term "finescale" to refer to ocean dynamical processes induced by mesoscale interactions and frontogenesis (Capet et al., 2008b, a; McWilliams, 2016; McGillicuddy, 2016; Lévy et al., 2018). Finescale structures are characterized by
a small Rossby number, horizontal scale of the order of 1–100 km, and a short lifetime (days–weeks). Numerical simulations and remote sensing observations have demonstrated that finescale lifetime is often similar to the phytoplankton growth timescale, suggesting that finescale processes can affect and modulate the phytoplankton community. Different physical processes associated with finescale structures are able to generate vertical velocities, such as deformations of the flow and spatial inhomogeneities (Giordani et al., 2006), eddy perturbation (Martin and Richards, 2001; Pilo et al., 2018), linear Ekman
pumping (McGillicuddy et al., 1998; Gaube et al., 2015), or eddy-wind interactions (McGillicuddy et al., 2007). Previous studies have well established that vertical motions impact biogeochemistry (Mahadevan and Tandon, 2006; Mahadevan, 2016; McGillicuddy, 2016). Upward vertical velocities drive deep nutrients into the euphotic layer and also move the phytoplankton cells along the water column resulting in changing light conditions. However, most of the in-situ studies related to the physical-biological coupling at finescale have focused on extreme situations occurring in coastal upwelling regions (Ribalet
et al., 2010) or in boundary currents (Clayton et al., 2014, 2017), where intense fronts and dramatic contrasts in water properties are found but that are not representative of the global ocean. Indeed, vast oceanic regions are dominated by weak fronts continuously created, moved and dissipated, which separate different water masses with similar properties. The ephemeral nature of these finescale structures makes them particularly challenging to sample. As a consequence, some recent cruises have used remote sensing and numerical simulations to define the sampling strategy allowing to target and measure finescale features
with physical sensors at high frequency (Shcherbina et al., 2015; Pascual et al., 2017; Petrenko et al., 2017). Concerning the biological variables, although progress in the understanding of phytoplankton cell cycle has been obtained from incubation, sample manipulation (Worden and Binder, 2003) and models (Geider et al., 1997; MacIntyre et al., 2000), performing in situ measurements at high frequency and resolution is a necessity to better understand these biological processes and their responses to the environment.

An efficient solution is to lead Lagrangian cruises using automated flow cytometers sampling at high frequency in order to resolve the phytoplankton diurnal cycle in situ, which is challenging using more conventional methods such as cultures or counting by optical microscopy (Thyssen et al., 2008; Fontana et al., 2018). This is the solution chosen by the PROTEVSMED-SWOT cruise. This cruise was performed in the southwestern Mediterranean Sea, south of the Balearic Islands (Dumas, 2018; Garreau et al., 2020) with the aim to study the physical and biological coupling at finescale. This area is characterized by
the presence of both fresh surface waters coming from the Atlantic (AW) and more saline waters from the Mediterranean region (Millot, 1999; Millot et al., 2006). AW enters the Mediterranean Sea through the Strait of Gibraltar and then forms a counterclockwise circulation along the continental slope of the western Mediterranean basin, caused by the combination of the Coriolis effect and the topographical forcing (Millot, 1999; Millot and Taupier-Letage, 2005; Millot et al., 2006). In the southwest part of the basin, this circulation is dominated by the Algerian Current (AC), which can form meanders and

mesoscale eddies due to baroclinic and barotropic instabilities (Millot, 1999). These eddies spread over the basin and join the study area south of the Balearic Islands, carrying with them the newly arrived AW, known as younger AW. In this region, the younger AW encounters the older AW sometimes also called resident AW (Balbín et al., 2012) or local AW (Barceló-Llull et al., 2019). The older AW is AW modified by cooling and evaporation during its progression along the northern part of the western Mediterranean basin. The encounter between these two AW often generates finescale frontal structures (Balbín et al., 2014). To our knowledge, except for the works of Balbín et al. (2012, 2014) and the glider experiments of Cotroneo et al. (2016) and Barceló-Llull et al. (2019), very few studies have been performed in this region and these frontal finescale structures have been scarcely sampled due to the difficulty of performing in situ experiments over these short-lived and small features. During the PROTEVSMED-SWOT cruise a Lagrangian adaptive sampling strategy was performed across a moderately energetic front separating two distinct AW at different stage of mixing (Tzortzis et al., 2021). The AW located south of the front is characterized by absolute salinity ($S_A$) between 37 $\mathrm{g\,kg^{-1}}$ and 37.5 $\mathrm{g\,kg^{-1}}$, corresponding to the younger AW recently entered into the Mediterranean Sea. Whereas north of the front, the AW referred to the older AW, is characterized by a higher $S_A$ (37.5 $\mathrm{g\,kg^{-1}}$ to 38 $\mathrm{g\,kg^{-1}}$). Tzortzis et al. (2021) have also observed contrasted phytoplankton abundances in these two water masses, with the smallest phytoplankton such as *Synechococcus* dominating south of the front in the younger AW, while microplankton was more abundant north of the front in the older AW. As a consequence, our previous study constitutes an important improvement in the understanding of the role of frontal structures at finescale on phytoplankton distribution in a moderately energetic ocean. Nevertheless, open questions remain concerning the mechanisms generating this observed distribution. Is it exclusively driven by the dynamics of the ocean currents? What is the role of biological processes? In the present study, we attempt to shed some light on these questions, and in particular on the patterns of phytoplankton abundances observed by automated flow cytometry during the PROTEVSMED-SWOT in the frontal structure, using the size-structured population model of Sosik et al. (2003). The specific objective of our study is to assess whether the observed contrasted abundances across the front were due to different growth and loss rates. Using high-frequency flow cytometry measurements across the front dividing two water masses, we were able to separately analyze each phytoplankton functional group and reconstruct their biovolume dynamics over a diel cycle in each water mass.

## 2 Materials and methods

### 2.1 The Sampling strategy

The PROTEVSMED-SWOT cruise, dedicated to the study of finescale dynamics, was conducted in the south of the Balearic Islands between April 30th and May 18th 2018, on board the R/V *Beautemps-Beaupré* (Fig. 1a). This cruise followed an adaptive Lagrangian strategy to measure at high spatial and temporal resolution several physical and biological variables with both in situ sensors and analysis of the sea surface water intake. The vessel route was designed ad-hoc on the basis of daily remote sensing dataset provided by the Software Package for an Adaptive Satellite-based Sampling for Oceanographic cruises (SPASSO, https://spasso.mio.osupytheas.fr, last access: July 3, 2023). SPASSO used altimetry-derived currents from the Mediterranean regional product (nrt_med_allsat_phy_l4) AVISO ("Archiving, Validation and Interpretation

of Satellite Oceanographic", https://www.aviso.altimetry.fr, last access: July 3, 2023) and ocean color observations. Chloro-phyll a concentrations ([chla], level 3, 1 km resolution, MODISAqua and NPPVIIRS sensors combined (after May 27,

2017) into a new product called MULTI) were provided by CMEMS, "Copernicus Marine Environment Monitoring Service", https://marine.copernicus.eu, last access: July 3, 2023. In addition, CLS provided the surface Chl concentration composite products, with the support of the CNES. They were constructed using a simple weighted average over the previous 5 days of data gathered by the Suomi/NPP/VIIRS sensor. SPASSO generated maps of dynamical and biogeochemical structures in both near real time (NRT) and delayed time (DT). Maps of [chla] allowed us to identify two water masses, characterized by distinct

[chla] values and separated by a zonal front at around 38° 30' N. This front was also detected using in situ horizontal velocities, temperature and salinity, as described in Tzortzis et al. (2021). These two water masses were sampled along a designated route of the ship, represented in black in Fig. 1b. Special attention was paid to adapting the temporal sampling in order to measure the phytoplankton diel cycle in each water mass. This was achieved by continuously sampling across both water masses along transects. While the ship did not remain in each water mass for 24h, day-to-day variability remained low and measurements

from several days were combined into one diel cycle (Fig. 1c). The shape depicted by the ship's track led us to call these areas north–south (NS) hippodrome (bold black line in Fig. 1b) performed between 11 May and 13 May 2018.

## 2.2  In situ measurements

During the cruise, the irradiance (wavelengths between 400 and 1000 nm) was measured by a CMP6 pyranometer (Kipp and Zonen; https://www.campbellsci.fr/cmp6, last access: July 3, 2023). Temperature and salinity were measured by a thermos-

alinograph (TSG). The TSG was equipped with two sensors: a CTD Sea-Bird Electronics SBE 45 sensor installed in the wet lab, connected to the surface water and which continuously pumped seawater at 3 m depth ; and an SBE 38 temperature sensor installed at the entry of the water intake. The TSG measurements were taken every 30 min, which corresponds to around 2 km spatial resolution at typical ship speeds. The data were converted into conservative temperature ($\Theta$) and absolute salinity ($S_A$) using the TEOS-10 standards of McDougall et al. (2012). To automatically sample and analyze phytoplankton cells, an

automated CytoSense flow cytometer (CytoBuoy, b.v. ; (Dubelaar et al., 1999; Dubelaar and Gerritzen, 2000)) was installed on board and connected to the seawater circuit of the TSG. The flow cytometer sampled the seawater in a dedicated small container called "subsampler". The subsampler isolates the seawater every 30 min which allows us to ignore the movement of the ship, while the flow cytometer performed its analysis. Between two consecutive samples the subsampler was flushed continuously by the seawater circuit of the ship in order to clean and renew the seawater. A sheath fluid made of 0.1 µm filtered

seawater stretched the sample in order to separate, align, center and drive the individual particles (i.e. cells) through a laser beam (488 nm wavelength). Several optical signals were recorded when each particle crossed the laser beam: the forward angle light scatter (FWS) and 90° side-ward angle scatter (SWS), related to the size and the structure (granularity) of the particles. Two distinct fluorescence emissions induced by the light excitation were also recorded, a red fluorescence (FLR) induced by chlorophyll a content and an orange fluorescence (FLO) induced by the phycoerythrin pigment content. The CytoUSB software

(Cytobuoy b.v.) was used to configure and control the flow cytometer and set two distinct protocols. The first protocol (FLR6) was dedicated to the analysis of the smaller phytoplankton, using a red fluorescence (FLR) trigger threshold fixed at 6 mV,

and a volume analyzed set up at 1.5 mL. The second protocol (FLR25) targeted nanophytoplankton and microphytoplankton with a FLR trigger level fixed at 25 mV and an analyzed volume of 4 mL. The FLR trigger was used to discriminate the red fluorescing phytoplanktonic cells from other particles (such as heterotrophic prokaryotes, nanoflagellates, ciliates, etc.).

Recorded data were analyzed with the CytoClus software (Cytobuoy b.v.) which retrieves information from the 4 pulse shapes curves (FWS, SWS, FLO, FLR) obtained for every single cell. These curves were then projected into distinct two-dimensional planes (cytograms) by computing the curves' integral. Using a combination of various cytograms (e.g., FWS vs. FLR, FLO vs. FLR) allows us to determine optimal cell clusters, i.e, cells sharing similar optical properties (see Fig. A1). The identification of phytoplankton functional groups is described in Appendix. These clusters have been demonstrated in the literature to rep-

resent phytoplankton functional groups (PFGs) (Dubelaar and Jonker, 2000; Reynolds, 2006; Thyssen et al., 2008; Edwards et al., 2015; Thyssen et al., 2022). Finally, the PFGs abundance (cells per milliliter) and mean light scatter and fluorescence intensities were extracted from each sample.

## 2.3 The size-structured population model

We used the size-structured population model described by Sosik et al. (2003) and adapted by Dugenne et al. (2014) and Marrec

et al. (2018), to estimate the in situ growth rates of every phytoplankton group identified by the CytoSense flow cytometer, in the older AW and the younger AW. Before applying the model, we reconstructed a daily cycle of 24 h in the two water masses for each phytoplankton group. We use the term reconstruction because the ship did not spend 24 h in a row in each water mass but sailed along two routes, each forming a sort of racetrack passing alternately through the two water masses (Fig. 1b, 1c). By eliminating the dates and keeping the associated sampling times, the 24-hour diel cycle can be reconstructed for each water

body (Fig. 1c). This relies on the hypothesis that the phytoplankton community and dynamics remained similar over the two days, and that hydrology and physics for each water mass remained alike during sampling. We also reconstructed the 24-hour irradiance in the two water masses (Fig. A2), because one of the most important parameters of this model is irradiance, since cell growth is dependent on light exposure due to photosynthesis.

The model of Sosik et al. (2003) uses as input the phytoplankton cell volume (biovolume) derived from cell light scatter intensities (FWS) (Eq. 1). Biovolumes were estimated using coefficients previously obtained by measuring a set of silica beads with the flow cytometer following the same settings used for phytoplankton analysis. The coefficients $\beta_0$ and $\beta_1$ used to convert FWS (arbitrary units, a.u.) to biovolume $v$ (μm$^3$) were derived from a log-log regression between FSW and silica bead volumes. These methods come from the studies of Koch et al. (1996) and Foladori et al. (2008).

$$v = exp(\beta_0) \times FWS^{\beta_1} \qquad (1)$$

with in our case $\beta_1$ = 0.9228 and $\beta_0$ = - 5.8702

In the size-structured population model, cells are classified into several size classes according to their dimensions at time $t$. Classes are logarithmically spaced as follows: for $i$ in 1,2,...,$m$ $v_i = v_1 2^{(i-1)\Delta v}$ where $\Delta v$ is constant and chosen to ensure that size classes cover the entire observed biovolume $v$, from $v_1$ to $v_m$ (Fig. 2). For *Synechococcus*, $\Delta v$ = 1/6 with $\Delta v$ constant and $m$ = 40, so that the model size classes encompassed our full measured size distributions (0.0279-2.5209 µm).

At any time $t$, the number of cells in size classes $\mathbf{N}$ (and $\mathbf{w}$ its corresponding normalized distribution), was projected to $t + dt$ via matrix multiplication (Eq. 2):

$$\mathbf{N}(t + dt) = \mathbf{A}(t)\mathbf{N}(t) \qquad and \qquad \mathbf{w}(t + dt) = \frac{\mathbf{A}(t)\mathbf{N}(t)}{\sum \mathbf{A}(t)\mathbf{N}(t)} \tag{2}$$

We chose $dt$ = 10 min (i.e., 10/60 h) as Sosik et al. (2003) and Dugenne et al. (2014), because cells of a specific phytoplankton group are unlikely to grow more than one size class over such a small time duration.

$\mathbf{A}(t)$ is a tridiagonal transition matrix that contains:

1) $\gamma$ : the probability of cellular growth

2) $\delta$: the probability of cells entering mitosis

3) the cells stasis, i.e., the probability for cells to maintain their state (i.e size) in equilibrium during the temporal projection.

Probability of cellular growth

The probability of cells growing to the next size class ($\gamma$) depends only on the light intensity (irradiance) necessary for photosynthesis, expressed as (Eq. 3):

$$\gamma(t) = \gamma_{max} \cdot (1 - exp(-E(t)/E^*)) \tag{3}$$

$\gamma_{max}$: maximum proportion of cells growing (dimensionless quantity)

$E$: irradiance ($\mu E\,m^{-2}\,s^{-1}$)

$E^*$: irradiance normalizing constant ($\mu E\,m^{-2}\,s^{-1}$)

Probability of cells entering mitosis

According to Dugenne et al. (2014), $\delta$ expresses a proportion (between 0 and 1) modeled by the combination of two normal distributions ($\mathcal{N}$). One is linked to the cell size, the other is linked to the time of cell division. Both imply an optimum, reached at $\bar{v}$ and $\bar{t}$ respectively, for cell division above which the cell size and the timing of division is suboptimal (Eq. 4).

$$\delta(t,v) = \delta_{max}\mathcal{N}(\bar{v},\sigma_v^2)\mathcal{N}(\bar{t},\sigma_t^2) \tag{4}$$

$\gamma_{max}$: maximum proportion of cells entering mitosis (dimensionless quantity)

$\bar{v}$: mean of the size normal distribution ($\mu m^3$)

$\sigma_v$: standard deviation of the size normal distribution ($\mu m^3$)

$\bar{t}$: mean of the time normal distribution (h)

$\sigma_t$: standard deviation of the time normal distribution (h)

Cells stasis

A third functional proportion is included in the transition matrix $\mathbf{A}(t)$, to represent cell stasis. Since this function illustrates a non-transition, it is modeled by the proportion of cells that neither divided nor grew between $t$ and $t + dt$ (Eq. 5).

$$[1 - \gamma(t)][1 - \delta(t,v)] \tag{5}$$

Optimal parameters

The set of parameters, $\theta$ is estimated by maximum likelihood function, assuming errors between observed $\mathbf{w}$ and predicted $\hat{\mathbf{w}}$ normalized size distributions (Eq. 6, 7, 8). Their standard deviations are estimated by a Markov Chain Monte Carlo approach (Geyer, 1992; Neal, 1993) that sample $\theta$ from their prior density distribution, obtained after running 200 optimizations on

bootstrapped residuals to approximate the parameter posterior distribution using the normal likelihood. (The likelihood function represents the probability of random variable realizations conditional on particular values of the statistical parameters).

$$\theta = [\gamma_{max}, E^*, \delta_{max}, \bar{v}, \sigma_v, \bar{t}, \sigma_t] = argmin(\sum(\theta)) \tag{6}$$

$$\sum(\theta) = \sum_t^{t+dt}\sum_{i=1}^{m}(\mathbf{w}(t) - \hat{\mathbf{w}}(t,\theta))^2 \tag{7}$$

$$\hat{\mathbf{N}}(t,\theta) = \mathbf{A}(t - dt,\theta)\mathbf{N}(t - dt) \tag{8}$$

$\hat{\mathbf{w}}$ is computed from $\hat{\mathbf{N}}$ following Eq. 2. The fit of the model is quantified using two numbers: the loss rate ($\sum(\theta)$, lower indicates better fit), and the correlation between the observed and modeled mean biovolumes $\bar{v}_{obs}$ and $\bar{v}_{mod}$ over the diel cycle

$(corr(\bar{v}_{obs}, \bar{v}_{mod})$, higher indicates better fit). Table 1 provides the model parameters being optimized.

Growth rate and loss rate


Once optimal parameters are identified, the model estimates a population intrinsic growth rate $\mu_{size}$, and a specific loss rate $l$, integrated over a 24 h period. The method uses the fact that the observed size distribution $\mathbf{N}$ is the result of both growth and loss processes, while the time projection of the initial size distribution $\mathbf{N}(0)$ using the model, $\hat{\mathbf{N}}$, is only the result of growth processes. The growth rate is calculated at each time step following Eq. 9, and integrated over 24 h. 200 iterations by a Markov

Chain Monte Carlo were run to estimate the standard deviation of group-specific growth rates.

$$\mu_{size}(t) = \frac{1}{dt} \ln \left( \frac{\sum_{i=1}^{m} \hat{\mathbf{N}}_i(t + dt)}{\sum_{i=1}^{m} \hat{\mathbf{N}}_i(t)} \right) \tag{9}$$

$i$: i th size class

$\hat{\mathbf{N}}$: predicted size distribution (cells cm$^{-3}$)

$m$: number of size classes

$dt$: time step (h)

$\mu_{size}$: growth rates (d$^{-1}$)

An independent growth rate estimation was obtained as $\mu_{ratio} = ln(\bar{v}_{max}/\bar{v}_{min})$ where $\bar{v}_{min}$ and $\bar{v}_{max}$ are the minimum and maximum of the mean observed biovolume $\bar{v}_{obs}$ over the diel cycle (Marrec et al., 2018). $\mu_{ratio}$ represents a minimum estimate

of the daily growth rate, that would be observed if cells synchronously only grew from the time $\bar{v}_{min}$ is observed (typically dawn) to the time $\bar{v}_{max}$ is observed (typically dusk), and only divided while $\bar{v}$ decreases. Since the model allows for any cell to grow, divide or be at equilibrium over the entire integration period (asynchronous populations), $\mu_{size}$ is expected to be higher than $\mu_{ratio}$. In practice, $\mu_{ratio}$ is sensitive to noise in the data and is only provided here as an alternative estimate of the growth rate that does not rely on the model.


The population loss rate $l$ is obtained by difference between the intrinsic growth rate $\mu_{size}(t)$ and the temporal change in logarithmic observed size distribution $\mathbf{N}$, which represents the net growth rate $r(t) = \mu_{size}(t) - l(t)$ so that:

$$\bar{l} = \int^{t} \mu_{size}(t) - \frac{1}{dt} \ln \frac{\mathbf{N}(t + dt)}{\mathbf{N}(t)} \tag{10}$$

# 3 Results

## 3.1 Spatio-temporal distribution of phytoplankton abundances in the two water masses

The sampling strategy adopted during PROTEVSMED-SWOT enabled us to sample two water masses with different properties. The map of the satellited-derived surface [chla] shows higher concentration in the northern part of the sampling route, corresponding to the older AW, than in the southern part, corresponding to the younger AW (Fig. 1b). Figure 3 shows the properties of the sea surface water as a function of time (from 11 May 00:00 to 13 May 12:00 UTC) along the sampling route. The older AW is characterized by a colder temperature and higher salinity than the younger AW. Figure 3 also displays the abundances of each phytoplankton group over these two water masses. *Synechococcus* and Pico2 are the most abundant. They present a clear surface distribution pattern, with high abundances in the warm and low salinity water, corresponding to the young AW. A similar distribution is observed for Pico1, Pico3 and RNano but with lower abundances than *Synechococcus* and Pico2. The abundances of SNano, PicoHFLR and Cryptophyte show less contrast along the cruise than the previous groups, nonetheless the highest abundances can be distinguished in the younger AW, in particular during the second and third passage (transect) across this water mass. Finally, microphytoplankton is the less abundant group, but it clearly shows a contrast between the two water masses, opposite to that of the other phytoplankton groups.

## 3.2 Phytoplankton cellular growth and division in the two water masses

The phytoplankton diurnal cycle was reconstructed in the two water masses using the size-structured population model originally developed by Sosik et al. (2003). Figures 4, 5, 6 represent the phytoplankton size distribution (i.e., biovolume) observed in situ and predicted by the model over 24 h for *Synechococcus*, RNano and SNano, respectively. From the predicted biovolume it is possible to derive specific growth ($\mu_{size}$) and loss ($l$) rates, summarized in Table 2 for the different phytoplankton groups in the two water masses, along with metrics of model performance. We also attempted to model the diurnal cycle for the picophytoplankton groups, i.e., Pico1, Pico2, Pico3, and PicoHFLR. However, their very noisy size distributions prevented us from obtaining reliable growth rate estimates. Similarly, microphytoplankton and Cryptophytes were not abundant enough to allow a reliable determination of their abundances and cell cycles. These cytometric groups are thus not considered further in this study.

For *Synechococcus*, in the older AW the prediction of the model (i.e., predicted biovolume) is similar to the observed size distribution (i.e., observed biovolume). Both display a day-long large size-class distribution centered approximately on 0.3 $\mu m^3$. In the younger AW (Fig. 4a, c) the distributions of observed and predicted biovolume are narrower than in the older AW and centered approximately on 0.2 $\mu m^3$ (Fig. 4b, d). As a consequence, the older AW is populated by larger cells of *Synechococcus* (mean observed biovolume $\bar{v}_{obs} = 0.38 \pm 0.04$ $\mu m^3$) than in the younger AW (mean biovolume $\bar{v}_{obs} = 0.21 \pm 0.04$ $\mu m^3$) (Table 2). Growth and loss rates also differ between the two water masses. In the older AW, the large cells of *Synechococcus* have a growth rate $\mu_{size} = 0.24 \pm 0.91$ $d^{-1}$ and a loss rate $l = 0.36$ $d^{-1}$, whereas in younger AW the smaller cells are characterized by higher growth ($\mu_{size} = 0.68 \pm 1.56$ $d^{-1}$) and loss ($l = 0.48$ $d^{-1}$) rates.

Relative to *Synechococcus*, cell size distribution and growth and loss rates are less contrasted between the older and younger AW for SNano (Fig. 6) and even more so RNano (Fig. 5). The mean observed RNano biovolumes are similar in the older and younger AW (63.5 ± 2.67 μm$^3$ and 61.2 ± 5.23 μm$^3$, respectively) (Table 2). For SNano, similar to *Synechococcus*, the older AW is predominantly composed of larger cells ($\bar{v}_{obs}$ = 85.0 ± 1.98 μm$^3$) than in the younger AW ($\bar{v}_{obs}$ = 63.8 ± 4.45 μm$^3$).

270  For both Nano groups, growh rates are generally very low in both water masses ($\mu_{size}$ < 0.1 d$^{-1}$). Loss rates are higher than growth rates, except for RNano in the younger AW (negative loss rate implying an external input of cells such as by advection). However, the corresponding optimization factor is the highest observed across the 6 modelisations, indicating this result is subject to caution.

## 4  Discussion

275  ### 4.1  The phytoplankton diurnal cycle

Although it has been clearly demonstrated that phytoplankton plays a fundamental role in the ocean ecosystem functioning (Watson et al., 1991; Field et al., 1998; Allen et al., 2005), numerous questions remain about their population dynamics in relation to finescale structures.

Coupling high-resolution in-situ flow cytometry measurements in two contrasted water masses with the size-structured 280  population model developed by Sosik et al. (2003) allowed us to characterize the structure of phytoplankton and to reconstruct its diel cycle of cell growth and division on both sides of a finescale front. The growth and loss rates ($\mu_{size}$ and $l$) found for *Synechococcus* are of the same order of magnitude as those obtained by Marrec et al. (2018) in the northwestern Mediterranean Sea using the same method. In section 3.2, we showed that the largest cells of *Synechococcus* were found in the older AW. These *Synechococcus* cells are characterized by a larger range of biovolume and lower growth and loss rates than those located 285  in the younger AW (Table 2). They are in average larger than in the younger AW as they grow slower at the population scale and divide less. Conversely, in the younger AW the distribution of the *Synechococcus* biovolume is narrower, which could be explained by cells being more active, more homogeneous in terms of size (biovolume) and better synchronized, leading to a smaller spread of the cell biovolume (Fig. 4b,d) with a dominance of small *Synechococcus* cells (Fig. 3). This also explains why higher abundances of *Synechococcus* are found in the younger AW (Fig. 3). Interestingly, the resulting net growth rate 290  (growth minus loss) is negative in the older AW, positive in the younger AW.

Results are more difficult to interpret for the nanoplankton groups RNano and SNano, expected to be mostly dominated by diatoms in the Mediterranean Sea (Marty et al., 2002; Siokou-Frangou et al., 2010; Navarro et al., 2014; El Hourany et al., 2019), especially in frontal systems (Claustre et al., 1994). RNano and SNano diel cycles are not as well-defined as for *Synechococcus*, leading to very small estimates of growth rates by the model. Optimization factors (linked to the mean 295  squared difference between observed and predicted normalized size distributions) are relatively high and/or temporal correlations between observed and predicted mean biovolume relatively low, indicating these results must be considered with caution. Nevertheless, our results suggest much lower growth and loss rates for nanoplankton than for *Synechococcus* and potentially

higher growth rates in the younger AW, similar to *Synechococcus* (excluding the likely unrealistic loss rate obtained for RNano in the younger AW).

## 4.2 Influence of the frontal system on the phytoplankton dynamics

Our previous article (Tzortzis et al., 2021) provided a description of the hydrodynamics and the hydrology of the region. In the following, we attempt to establish the potential link between the characteristics of the two AW separated by the front, the physical forcings associated with this frontal structure and the particular distribution of phytoplankton in terms of cells size and abundances. Figure 7 summarizes the physical forcing evidenced in this frontal area in the previous publication during the PROTEVSMED-SWOT cruise, superimposed with the biovolumes and the abundances of the different phytoplankton groups sampled in situ by the automated flow cytometer.

The older AW is characterized by larger cells of *Synechococcus* and nanophytoplankton with low abundances, low intrinsic growth rates and negative net growth rates, suggesting an older, declining population, whereas the younger AW is dominated by small cells with high abundances, and at least for *Synechococcus* high intrinsic growth rates and a positive net growth rate, suggesting a slightly growing or stable population (nanoplankton results in the younger AW are subject to caution as optimization factors are relatively high). Furthermore, microphytoplankton (i.e largest type of phytoplankton) is more abundant in older AW than in the younger AW. The early experimental works of Marshall and Orr (1928); Jenkin (1937); Huisman (1999) have well established that the light and nutrients are essential for phytoplankton growth. The reconstruction of the circadian cycle indicates that irradiance was similar in the two water masses (Fig. 4, 5 and 6, red lines), with corresponding daily total irradiance of 286 and 299 $\mu$E m$^{-2}$ for the older AW and the younger AW, respectively (Fig. A2). That is why, the availability of light seems not to be the principal cause explaining the difference of phytoplankton dynamics and its distribution in the two AW. An other possible explanation is that these two water masses are characterized by different nutrient concentrations, thus favoring certain phytoplankton groups. Bethoux (1989) and Schroeder et al. (2010) have observed that the older AW is slightly more enriched with nutrients than the younger AW because the older AW receives nutrient inputs from the continent (river discharges, rain, wind) during its circulation across the Mediterranean basin. Unfortunately, it was not possible for both technical and funding reasons to perform nutrient measurements during the 2018 cruise, so that we cannot conclusively assess nutrient patterns during the cruise. Assuming that the nutrient distribution across the two water masses was similar to what was previously measured by Bethoux (1989) and Schroeder et al. (2010), we propose that higher nutrient concentrations in the older AW explain the observed phytoplankton cell size and abundances distributions. Our hypothesis is supported by similar observations by Jacquet et al. (2010) and Mena et al. (2016) who also found the highest abundances of the small phytoplankton (*Synechococcus* and picophytoplankton) in the most oligotrophic waters, i.e., the younger AW. Furthermore, previous studies have shown that the proportion of picophytoplankton in the total phytoplankton biomass is higher in oligotrophic regions than in mesotrophic or eutrophic regions (Zhang et al., 2008; Cerino et al., 2012). Indeed, their better surface:size ratio due to their small size confers them a better capacity to inhabit areas with very low nutrient concentration compared to larger phytoplankton (Kiørboe, 1993; Marañón, 2015). Since our study area is always oligotrophic (Moutin et al., 2012), a small variation of the nutrient concentration (typically $\leq 0.1$ $\mu$M of nitrate) is sufficient to generate higher abundance of picophytoplankton.

Some studies have attempted to link hydrological condition and the phytoplankton dynamic (Qasim et al., 1972; Brunet et al., 2006; Marañón et al., 2012, e.g.,). However, their results showed that the influence of these hydrological parameters on the phytoplankton growth and distribution was difficult to estimate, compared to the effects of nutrient availability and radiation exposure.

Other physical processes occurring at the front can explain the different dynamics of phytoplankton groups. The work of Lévy et al. (2001); Pidcock et al. (2016); Mahadevan (2016) have highlighted that the availability of light and nutrient is driven by physical dynamics such as vertical velocities. The computation of the vertical motions in the frontal area, as represented in Fig. 7 (see also Fig. A3), show the presence of upwellings and downwellings in the frontal area. However, due to the lack of nutrients measurements during the cruise, we are not able to quantify the impact of these vertical velocities.

### 4.3 Limitations of the study and recommendations

Growth and loss rates were estimated using the size-structured population model originally developed by Sosik et al. (2003), which was fitted to a measured diel cycle of cell size distributions. More precisely, the rates were calculated based on the fitted size distribution predicted by the model, and its comparison with the observed size distribution. Because of this, results are sensitive to noise in the measured size distributions. We could not obtain reliable results for the picophytoplankton groups due to noisy distributions, because they probably contained several taxa with differing dynamics (Siokou-Frangou et al., 2010; Le Moal et al., 2011). To take into account this constraint, in future experiments, sorting by flow cytometry and identification with a microscope and/or genetics analysis should be planned to identify taxa in the various phytoplanktonic groups defined by flow cytometry. Nevertheless, these techniques are not easily applicable to large numbers of samples contrary to automated flow cytometry, which means that a careful selection of samples will be necessary.

For the taxonomic groups where reasonable size distributions could be estimated over a diel cycle, the model fit was evaluated using two metrics: the optimization loss rate $\sum(\theta)$ and the correlation between the observed and modeled mean biovolumes over the diel cycle $corr(\bar{v}_{obs}, \bar{v}_{mod})$ (Table 2). These metrics, as well as visual comparisons of the modeled and observed size distributions (Figures 4, 5, 6), indicate differing degrees of confidence in our estimated growth and loss rates, with the highest confidence obtained for *Synechococcus*. In future experiments, these rate estimates could be improved by more accurately measuring the phytoplankton diel cycle (i.e., by continuously sampling the same water mass over 24 h rather than by compiling several days to reconstruct a diel cycle). Furthermore, coupling flow cytometry with NanoSIMS analysis, as in the works of Bonnet et al. (2016) and Berthelot et al. (2019), could be also useful to get independent estimates of the growth rates, although the cost and the successive incubations required by this methodology are not adapted to high-frequency sampling of finescale ocean structures.

Overall, while estimating growth and loss rates by fitting a model to automated flow cytometry data remains limited by our capacity to accurately resolve the size distribution of independent phytoplankton groups over a full diel cycle, the method used here has several advantages. Other methods involve incubations that are dependent on the accurate reproduction of the marine environment in incubators. By contrast, automated flow cytometry as applied here measures the temporal evolution of phytoplankton in situ. The automated CytoSense flow cytometer, deployed underway, requires little maintenance or manipulation

during the cruise contrary to time-consuming sampling and incubations. As such, while growth and loss rates obtained from automated flow cytometry would benefit from independent validation, they have the potential to provide in situ estimates of biological rates that are traditionally difficult to measure.

## 5   Conclusion and perspectives

Phytoplankton structure and dynamics are a complex result of many interacting biological and physical phenomena. Finescale structures, and in particular fronts, generate vertical velocities which displace phytoplankton cells and nutrients in the water column, thus influencing phytoplankton communities. These mechanisms are only partially understood because the spatial scale of these structures and their ephemeral nature make them particularly difficult to monitor in situ. In particular the specific growth rates for the various phytoplankton groups, while being a key quantity for explaining the community structure, is chal-

lenging to obtain. In this study, we addressed this problem by monitoring the dynamics of several phytoplankton groups in two distinct water masses both in terms of hydrology and phytoplankton abundances. In the Mediterranean sea, the low nutrient content is indeed the perfect condition when addressing this question, because even weak horizontal or vertical nutrient redistributions associated with the finescale circulation are likely to result in a biological response (Talmy et al., 2014; Hashihama et al., 2021).

The originality of our work resides in the combination of (i) a Lagrangian sampling strategy to track adaptively a frontal region for several days; (ii) a high spatiotemporal resolution of the phytoplankton community thanks to flow cytometry; and (iii) the use of a size-structured population model for reconstructing the diurnal cycle of several phytoplankton groups and for identifying contrasted dynamics in the two water masses.

For *Synechococcus* and nanophytoplankton, we found higher cell size in the older AW located north of the front, associated

with lower abundances. A possible explanation is that the older AW is more enriched in nutrients than the younger AW, thus favoring larger cells. This remains a hypothesis because of a lack of nutrient data. Besides the employment of a Lagrangian adaptive strategy, another novelty of our study is that we applied the Sosik et al. (2003) model on several phytoplankton groups identified by flow cytometry, whereas previous studies only applied it to *Synechococcus* and *Prochlorococcus* (Ribalet et al., 2010; Hunter-Cevera et al., 2014; Marrec et al., 2018; Fowler et al., 2020) or to certain types of diatoms (Dugenne et al.,

2014). We obtained good results for *Synechococcus* and nanophytoplankton. Our analysis has been less conclusive for the other phytoplankton groups because of the presence of confounding effects, and in particular in the limitation of flow cytometry in taxonomic resolution. A further limitation of our study has been the lack of concomitant nutrient data, which hindered the possibility of testing mechanistic hypothesis leading to the emergence of type dominance.

Built over the experience that we acquired with the study presented here, the recent BioSWOT-Med cruise has been lead

in the South Western Mediterranean in spring 2023 to address these limitations. This cruise added to the methodological approach presented here high-resolution, high-precision nutrient measurements (necessary considering the oligotrophy of the Mediterranean Sea), as well as a better taxonomic resolution based on DNA metabarcoding and microscopy. The BioSWOT-Med cruise also included zooplankton and virus sampling aimed at estimating zooplankton grazing and viral lysis on the

different phytoplankton groups to better characterize the cell loss. In the long term, we believe that these types of studies pave

the way to the direct integration of growth rates in biogeochemical models (Cullen et al., 1993; Baklouti et al., 2006) and should eventually provide a better assessment of the biogeochemical contribution of phytoplankton in oligotrophic ecosystems as well better predict its evolution in the context of global change.

*Data availability.* The physical data are open access and available at https://www.seanoe.org/data/00512/62352/ (last access: July 3, 2023) (Dumas, 2018). Flow cytometry data are available by request to the corresponding author.

*Author contributions.* RT post-processed the in situ observations, performed the analysis of the results and led the writing of the manuscript. AD and GG designed the Lagrangian experiment and collected the in situ data together with FD. AP, SB and FdO provided land support concerning the sampling strategy. LI carried out the analysis of flow cytometry data. MM and YZ provided their expertise about the flow cytometry analysis and the results obtained with the size-structured population model. All the authors discussed the results and contributed to the writing of the manuscript.

*Competing interests.* The contact author has declared that neither they nor their co-authors have any competing interests.

*Acknowledgements.* This work was supported by the CNES in the framework of the BIOSWOT-AdAC project (https://www.swot-adac.org/, last access: July 3, 2023) by the MIO Axes Transverses program (AT-COUPLAGE) and the Sino-French IRP (CNRS-CAS) DYF2M program. The chlorophyll $a$ product is produced by CLS. The authors thank the SHOM and the crew of the RV *Beautemps-Beaupré* for shipboard operations. The authors thank also Melilotus Thyssen for providing the CytoBuoy® flow cytometer and her help in flow cytometry data

analysis. The flow cytometer was funded by the CHROME project, Excellence Initiative of Aix-Marseille University – A*MIDEX, a French 11 Investissements d'Avenir program. SPASSO is operated and developed with the support of the SIP (Service Informatique de Pythéas) and in particular Christophe Yohia, Julien Lecubin, Didier Zevaco and Cyrille Blanpain (Institut Pythéas, Marseille, France). The project leading to this publication received funding from the European FEDER Fund under project number 1166-39417. Roxane Tzortzis is financed by a MENRT PhD grant (École Doctorale Sciences de l'environnement – ED 251, Aix-Marseille University).

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

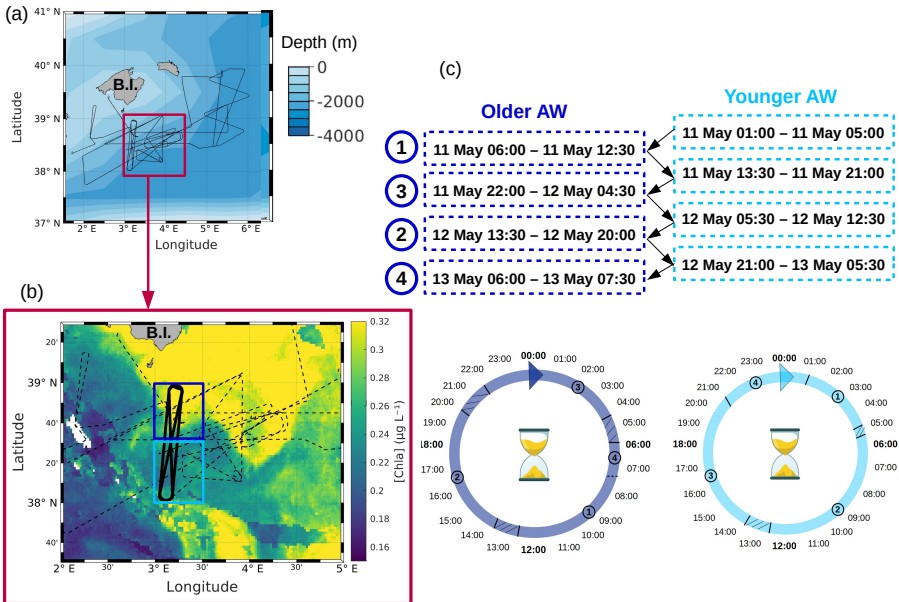

**Figure 1.** (a) Route of the RV *Beautemps-Beaupré* during the PROTEVSMED-SWOT cruise. The purple box encloses a (b) zoom of the sampling region with overlaid chlorophyll-a concentration ($\mu$g L$^{-1}$) of 11 May 2018. In panel (b) black dotted line represents the route of the ship and the bold black line represents the route of the Lagrangian sampling across the older AW (delimited by the box in dark blue) and the younger AW (delimited by the box in light blue). (c) Dates of the transects across the older AW and the younger AW, used to reconstruct a day of 24 h period in each water mass.

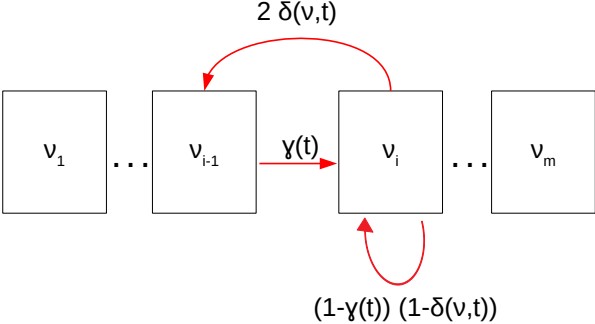

**Figure 2.** Cell cycle stages in the size-structured population model. Cells may grow to the next size class ($\gamma$) or be at equilibrium $(1 - \gamma(t))(1 - \delta(v,t))$. Above a particular size, cells are large enough to divide in two daughter cells with probability ($\delta$). Figure adapted from Sosik et al. (2003).

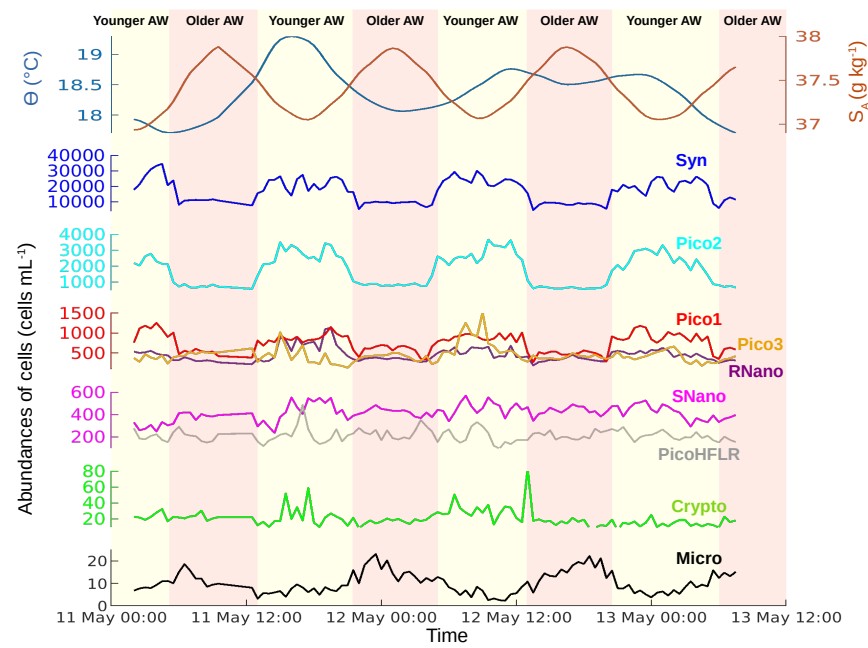

**Figure 3.** Temporal evolution of sea surface conservative temperature ($\Theta$) in °C, absolute salinity ($S_A$) in $\mathrm{g\,kg^{-1}}$, and phytoplankton abundances in $\mathrm{cells\,mL^{-1}}$, from 11 May 00:00 to 13 May 12:00 (UTC). Vertical colors correspond to the two water masses separated by the front (see Fig. 1).

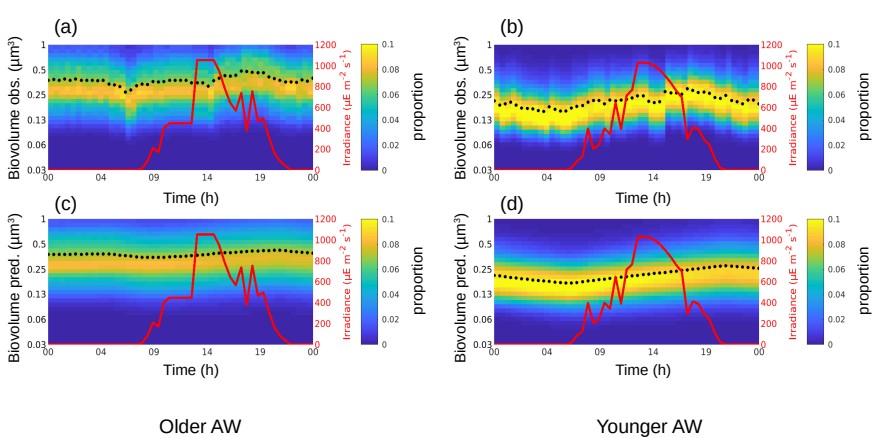

**Figure 4.** The background color represents the *Synechococcus* cell size distribution (i.e., biovolume in μm$^3$) observed (a, b) and predicted by the model (c, d) in the older AW (a, c) and in the younger AW (b, d) during 24 h. The black dots represent the mean of the biovolume ($\bar{v}_{obs}$ and $\bar{v}_{mod}$) and the red line represents the irradiance (μE m$^{-2}$ s$^{-1}$).

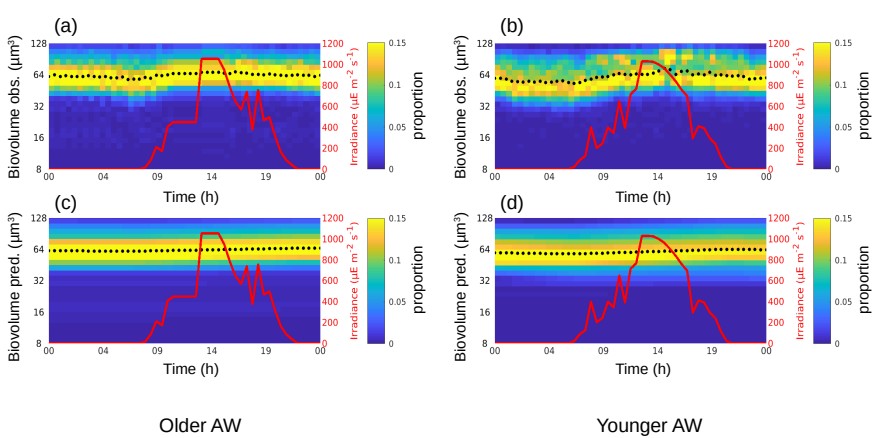

**Figure 5.** Same as Fig. 4 for RNano.

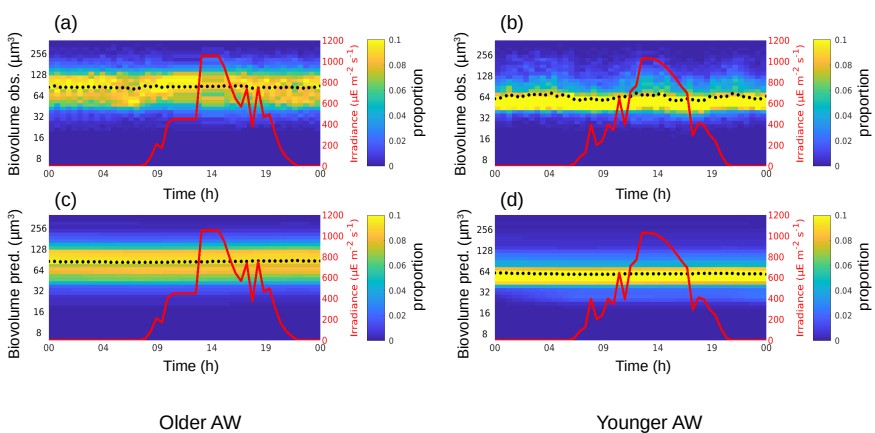

**Figure 6.** Same as Fig. 4 for SNano.

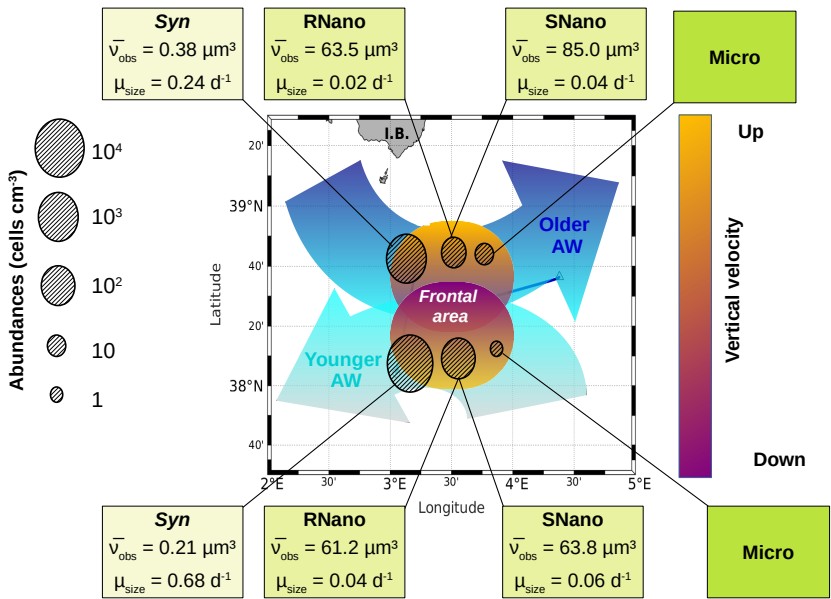

**Figure 7.** The contrasted distribution of phytoplankton in the frontal area. The circles represent the abundances of the several phytoplankton groups in the two water masses separated by the front. The boxes indicate the biovolume observed ($\bar{v}_{obs}$) and the growth rates ($\mu_{size}$) for each phytoplankton group, as estimated from the model. Figure adapted from Tzortzis et al. (2021).

**Table 1.** Model parameters being optimized.

| Parameters | Definition | Interval | Units |
|:---:|:---:|:---:|:---:|
| $\gamma_{max}$ | Max proportions of cells in growing phase | $[0,1]$ | $\emptyset$ |
| $E^*$ | Irradiance normalizing constant | $[0,\infty[$ | $\mu E\,m^{-2}\,s^{-1}$ |
| $\delta_{max}$ | Max proportions of cells in mitosis | $[0,1]$ | $\emptyset$ |
| $\bar{v}$ | Mean of size density distribution | $[v_{min}, v_{max}]$ | $\mu m^3$ |
| $\sigma_v$ | Standard deviation of size density distribution | $[10^{-06},\infty[$ | $\mu m^3$ |
| $\bar{t}$ | Mean of temporal density distribution | $[1, 24\frac{1}{dt}+1]$ | hours |
| $\sigma_t$ | Standard deviation of temporal density distribution | $[10^{-06},\infty[$ | hours |

**Table 2.** Means of biovolumes observed ($\bar{v}_{obs}$) and modelized ($\bar{v}_{mod}$) in µm$^3$, growth rates ($\mu_{size}$, $\mu_{ratio}$ in d$^{-1}$) and loss rate ($l$, in d$^{-1}$) for the phytoplankton groups, in the older and younger AW, as well as model fit parameters (see section 2.3).

|  | *Synechococcus* | RNano | SNano |
|---|---|---|---|
| Older AW | $\bar{v}_{obs} = 0.38 \pm 0.04$ | $\bar{v}_{obs} = 63.5 \pm 2.67$ | $\bar{v}_{obs} = 85.0 \pm 1.98$ |
|  | $\bar{v}_{mod} = 0.38 \pm 0.02$ | $\bar{v}_{mod} = 63.5 \pm 1.79$ | $\bar{v}_{mod} = 84.7 \pm 1.38$ |
|  | $\mu_{size} = 0.24 \pm 0.91$ | $\mu_{size} = 0.02 \pm 0.20$ | $\mu_{size} = 0.04 \pm 0.26$ |
|  | $\mu_{ratio} = 0.59$ | $\mu_{ratio} = 0.17$ | $\mu_{ratio} = 0.11$ |
|  | $l = 0.36$ | $l = 0.07$ | $l = 0.11$ |
|  | $\sum(\theta) = 0.05$ | $\sum(\theta) = 0.139$ | $\sum(\theta) = 0.067$ |
|  | $corr(\bar{v}_{obs}, \bar{v}_{mod}) = 0.60$ | $corr(\bar{v}_{obs}, \bar{v}_{mod}) = 0.46$ | $corr(\bar{v}_{obs}, \bar{v}_{mod}) = -0.05$ |
| Younger AW | $\bar{v}_{obs} = 0.21 \pm 0.04$ | $\bar{v}_{obs} = 61.2 \pm 5.23$ | $\bar{v}_{obs} = 63.8 \pm 4.45$ |
|  | $\bar{v}_{mod} = 0.22 \pm 0.03$ | $\bar{v}_{mod} = 60.6 \pm 2.17$ | $\bar{v}_{mod} = 59.1 \pm 0.61$ |
|  | $\mu_{size} = 0.68 \pm 1.56$ | $\mu_{size} = 0.04 \pm 0.28$ | $\mu_{size} = 0.06 \pm 0.19$ |
|  | $\mu_{ratio} = 0.63$ | $\mu_{ratio} = 0.33$ | $\mu_{ratio} = 0.24$ |
|  | $l = 0.48$ | $l = -0.12$ | $l = 0.23$ |
|  | $\sum(\theta) = 0.153$ | $\sum(\theta) = 0.417$ | $\sum(\theta) = 0.247$ |
|  | $corr(\bar{v}_{obs}, \bar{v}_{mod}) = 0.65$ | $corr(\bar{v}_{obs}, \bar{v}_{mod}) = 0.56$ | $corr(\bar{v}_{obs}, \bar{v}_{mod}) = 0.15$ |

## APPENDIX

### Identification of the phytoplankton functional groups by flow cytometry

Up to 9 groups of phytoplankton have been identified on the cytograms (Fig. A1), thanks to their light scatter (forward scatter FWS, and sideward scatter SWS) and fluorescence intensities (red fluorescence FLR, and orange fluorescence FLO). These groups have been called using the conventional names used by flow cytometrists, i.e., some groups relate to taxonomy (*Synechococcus*, Cryptophytes) while others relate to a range of sizes (picoeukaryotes, nanoeukaryotes) as described by Sieburth et al. (1978). *Synechococcus* (Syn on Fig. A1c) is a prokaryotic picophytoplankton that can be distinguished from the other picophytoplankton owing to its high FLO intensity, induced by phycoerythrin pigment content. Cryptophytes (Crypto on Fig. A1c) were also discriminated from the other groups as they also produce a characteristic orange fluorescence induced by phycoerythrin. Concerning the other phytoplankton groups, 4 eukaryotic picophytoplankton groups were put in evidence: Pico1 (on Fig. A1c) characterized by lower FLR and FLO intensities than *Synechococcus*, Pico2 and Pico3 (on Fig. A1d) with higher FWS, SWS and FLR intensities than Pico1, PicoHFLR (on Fig. A1a) has a high FLR signal induced by chla. We defined 2 distinct nanophytoplankton groups (SNano and RNano) according to their high FLR and FLO intensities. SNano exhibits higher SWS/FWS ratio and SWS intensities than RNano (Fig. A1b and Fig. A1a). Finally, microphytoplankton (Micro) is characterized by the highest FLR and FWS intensities (Fig. A1c).

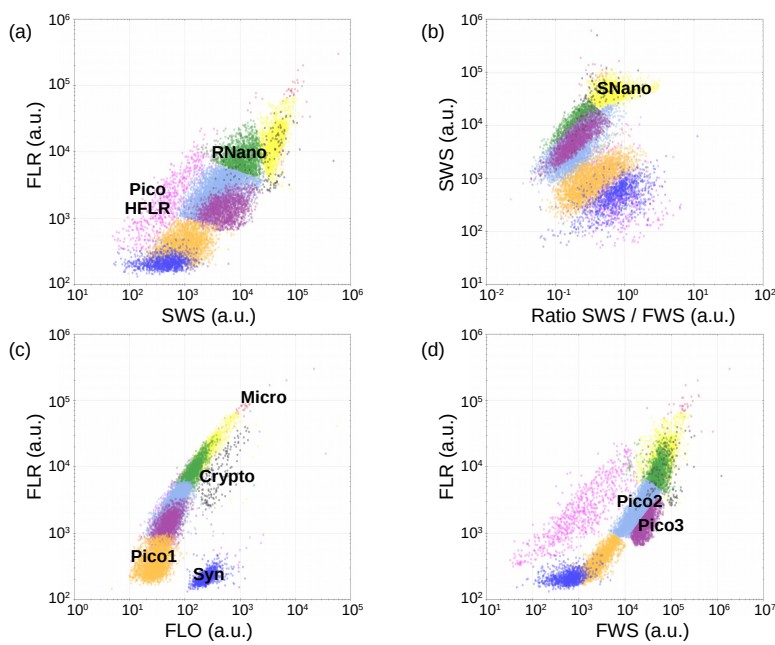

**Figure A1.** Cytograms obtained with the CytoSense automated flow cytometer. *Synechococcus* are in dark blue (Syn), the picophytoplankton with lowest FLO in orange (Pico1), the picophytoplankton with intermediate FWS in light blue (Pico2), the picophytoplankton with highest FWS in purple (Pico3), the picophytoplankton with a high red fluorescence in pink (PicoHFLR), the nanophytoplankton with high SWS/FWS ratio in yellow (SNano) and higher SWS intensities than the other nanophytoplankton (RNano) in green, the Cryptophytes in grey (Crypto) and the microphytoplankton in red (Micro). The flow cytometry units for both fluorescence and light scatter are arbitrary (a.u). Figure extracted from Tzortzis et al. (2021).

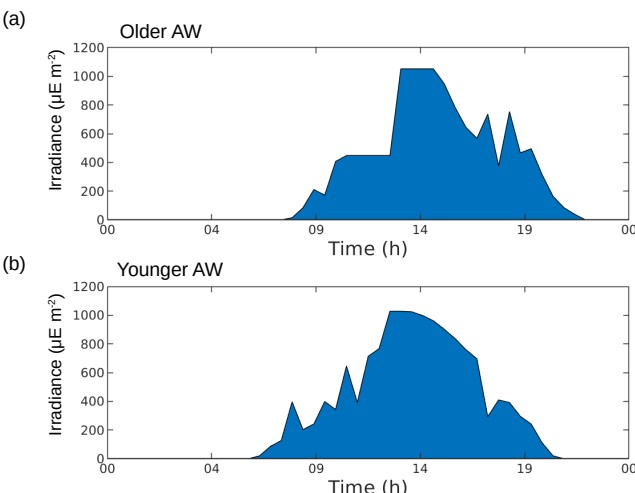

**Figure A2.** Reconstruction of irradiance during 24 h in the older AW (a) and the younger AW (b). Computation of trapezoidal integration of irradiance, in the older AW, E1 = 286 $\mu$E m$^{-2}$ and in the younger AW, E2 = 299 $\mu$E m$^{-2}$.

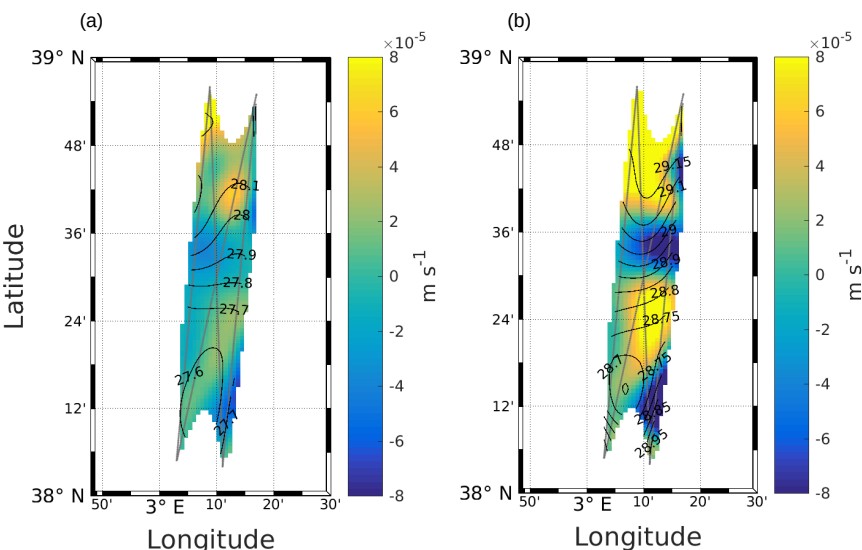

**Figure A3.** Vertical velocities at 25 m (a) and 85 m (b), calculated with the omega equation. Figure extracted from Tzortzis et al. (2021).