# Peer review of "The contrasted phytoplankton dynamics across a frontal system in the southwestern Mediterranean Sea"

_EGUsphere, 2022_

## Author Comment (AC2)

**Authors's reply to Anonymous Referee #1 comments on on egusphere-2022-1008 Tzortzis et al.**

Dear referee,

Thank you very much for your constructive comments and suggestions, as well as your corrections of the grammar and spelling in order to improve the quality of the manuscript. According to the Biogeosciences guidelines, hereinafter we address the main points you raised. We are confident that following your remarks and suggestions, as well as the ones from the other anonymous referee, we can improve our manuscript for publication in Biogeosciences. Your comments are in black, our replies in blue.

**General Comments**

In the manuscript "The contrasted phytoplankton dynamics across a frontal system in the southwestern Mediterranean Sea", Tzortzis et al. compare the phytoplankton communities at two different water masses separated by a frontal region. The work is original. The sampling design to analyze the two water masses separated by a front, and the phytoplanktonic community that characterizes them, is very interesting. Especially having a tool like the CytoBuoy.

We thank you for these positive comments about our work.

However, the manuscript needs improvement in many aspects before it can be considered for publication. In general, it is a disorganized text. The story does not flow, the paragraphs do not focus on clear topics, there are very long and confusing sentences, there are methodological descriptions in the results and results in the discussion... All sections should be carefully reviewed and improved. Especially the introduction and discussion.

We apologize for not being clear enough. We will rework each section in order to make our scientific message flow, taking into account your useful suggestions and those of the second referee.

The study area is barely described or named, why is it so relevant to focus on that particular front (besides the scope of the satellite)?

We will improve the description of the study area, including more information on the general dynamics of the Algerian sub-basin and on the specific situation encountered during our 2018 cruise. The importance of the frontal area studied in the present article is due to the fact that most of the in-situ studies related to the physical-biological coupling at finescale have focused on extreme situations occurring in boundary currents, where intense fronts and dramatic contrasts in water properties are found but are not representative of the global ocean. Indeed, vast oceanic regions are dominated by weak fronts continuously created,

moved and dissipated, which separate different water masses with similar properties. Very few studies exist in these regions due to the difficulty of performing in situ experiments over these short-lived and small features. In our previous work Tzortzis et al. (2021) we showed that the fine-scale front observed during our 2018 cruise in an oligotrophic and moderately energetic region (the SW Mediterranean) also drove phytoplankton diversity patterns. In the present study we aim at explaining how, by combining the available hydrological and cytometric data recorded at high spatio-temporal frequency.

Moreover, in the first paragraph of the introduction, the authors mention the context of climate change. How will climate change affect the presence and intensity of the fronts? And in turn, how a possible change in the intensity and frequency of the fronts will affect the associated phytoplankton communities? As a reader, I feel like I am being shown an interesting image, but in black and white instead of full color.

The impact of climate change on the ocean is a topic of current research, but many uncertainties remain (Collins et al., 2010 ; Cheng et al., 2022 ; Yao et al., 2022, etc). The predictions concerning the consequences of global warming on ocean circulation are already observable in the Southern Ocean, where the Subantarctic Front is particularly affected (Jia-Rui Shi et al., 2021).
Future climate-driven changes to ocean circulation patterns have the potential to alter dispersal pathways, potentially affecting the ability of species to track and adapt to climate change (Wilson et al., 2016). This impact may be very important in the Mediterranean Sea (MerMex Group, 2011).
The fact that the links between finescale and biological processes are still unknown (see below for more explanation and references) implies that the impact of climate change on them is speculative.
The objective of this study is to contribute to improving our knowledge of the finescale physical-biological coupling, by focusing on the distribution of the phytoplankton and its possible explanation (cell cycle). Our study does not aim to understand the consequence of climate change-induced changes in fronts. Since our study is not related to climate change, we decided to remove the reference to climate change in the Introduction in the reworked manuscript.

**Specific Comments**

**Introduction**

The authors describe in ~15 lines the relevance of phytoplankton and ocean fronts. In my opinion, more information is needed. Knowing the abundance and diversity of phytoplankton is important, but its role in the carbon cycle should also be highlighted, which changes depending on whether the community is dominated by small or large species. Moreover, an oceanic front is not defined, the authors describe briefly the

physical-biological interaction. I also miss an intro to the study region.

We will improve this part following your suggestions. Here we provide a draft of this section that we will be included in the reworked version of the manuscript:

"Phytoplankton forms the basis of the marine food web and plays a crucial role in biogeochemical processes, including the efficiency of the biological carbon pump, i.e., fixing CO2 and exporting it into the ocean depth (Field et al., 1998 ; De La Rocha and Passow, 2007). This process is critical for global ocean sequestration of carbon and therefore for the modulation of atmospheric $CO_2$. Furthermore, the biological carbon pump is also modulated by the size structure of the phytoplankton community. Indeed small or large phytoplankton species are associated with different efficiencies for particle export, remineralization, and transfer to deep ocean (Boyd and Newton, 1999 ; Guidi et al., 2009 ; Hilligsøe et al., 2011 ; Mouw et al., 2016, etc). Finally, phytoplankton is responsible for half of the primary production of the planet (Field et al., 1998), while its biomass is only < 1 % of the global biomass (Winder et al., 2010). Thanks to the photosynthesis, phytoplankton fuels the ocean in free $O_2$.
That is why it is primordial to understand the factors that rule phytoplankton abundance and diversity, particularly in the context of climate change (Hays et al., 2005 ; Behrenfeld et al., 2006 ; Winder and Sommer, 2012 ; Bates et al., 2018, etc). Indeed, a consequence of the climate change on the phytoplankton is to modulate its ability to export $CO_2$ into the deep ocean (Laws et al., 2000). Furthermore, the studies of Lomas et al. (2009), Menkes et al. (2016) and Barrillon et al. (2023) have shown that impulse events such as storms can generate mixing and stirring of the surface layer and trigger transitional peaks in primary production, mainly explained, for the first two references, by nitracline shoaling and grazer dilution. The occurrence of such intense events will tend to increase in the context of global change, impacting even further the functioning of the biological carbon pump."

Furthermore, below we provide a brief description of the study area:

"The region south of the Balearic Islands is characterized by the presence of both fresh surface waters coming from the Atlantic (AW) and more saline waters from the Mediterranean region (Millot, 1999; Millot et al., 2006).
AW enters the Mediterranean Sea through the Strait of Gibraltar and then forms a counterclockwise circulation along the continental slope of the western Mediterranean basin, caused by the combination of the Coriolis effect and the topographical forcing (Millot, 1999; Millot and Taupier-Letage, 2005; Millot et al., 2006). In the southwest part of the basin, this circulation is dominated by the Algerian Current (AC), which can form meanders and mesoscale eddies due to baroclinic and barotropic instabilities (e.g., Millot, 1999). These eddies spread over the basin and join the study area south of the Balearic Islands, carrying with them the newly arrived AW, known as younger AW. In this region, the younger AW encounters the older AW sometimes also called resident AW or local AW (Balbín et al., 2012). The older AW is AW modified by cooling and evaporation during its progression along the northern part of the western Mediterranean basin. The presence of these two types

of water makes this area particularly suitable for the formation of frontal structures (Balbín et al., 2014)."

After that brief introduction, the difficulty of an in situ study is described, and without continuing the story fluently, they begin to talk about an oceanographic campaign/project. I think the information is relevant, although the more technical details should be indicated in the methodology.

We will move the technical details of the campaign in the methodology, and improve the transition between the difficulty of an in situ study and our particular campaign.

Finally, the last paragraph is confusing. There is a lot of information, but not all makes sense, and it is kind of disorganized. The last paragraph of the introduction should clearly define the objectives of the study and how the authors would answer them.

We will clarify this part in the reworked manuscript following these suggestions:
"The objectives of this study are to assess whether the observed contrasted abundances across the front are due to different growth and loss rates. Using high-frequency flow cytometry measurements across the front dividing two water masses, we were able to separately analyze each phytoplankton functional group and reconstruct their biovolume dynamics over a diel cycle in each water mass."

Ln 16: Phytoplankton are essential for marine ecosystems, but not really for the functioning of the oceans... Oceans can function without life.
Ln 16-17: Revise the sentence. The $CO_2$ assimilated by phytoplankton can be exported to deep waters when they die or are partially eaten, being decomposed at depth; that's the biological pump. But not when they are eaten by higher trophic layers.

Our sentence was unclear and we plan to modify it as follow:

"Phytoplankton plays a crucial role in the oceans by regulating climate and forming the base of the food chain (Sterner et al., 1994). Its capacity to perform photosynthesis influences the global carbon cycle, by fixing $CO_2$ and exporting it into the ocean depth through the biological pump. Phytoplankton production also supports higher trophic levels, impacting ecosystem functioning."

Ln 20-22: Add references.

We plan to add here the following references: (Clayton et al., 2014 ; Mahadevan 2016 ; Lévy et al., 2018).

Ln 22-23: Revise the sentence. It seems that the idea the authors are trying to convey is that the temporal scale of growth/evolution of the phytoplankton community is due

to a fine-scale coupling. It seems that the fronts are the ideal environment for phytoplankton when it is not necessarily true.

We plan to better explain our point of view by adding a sentence as follows:
"Indeed, the phytoplankton dynamics temporal scale is of the same order of magnitude as the one of finescale processes such as eddies, filaments or fronts, suggesting the possibility of a close coupling between phytoplankton growth and finescale forcing. This does not necessarily imply that phytoplankton can grow only in presence of fronts, but fronts create favorable conditions for phytoplankton and life in general (Clayton et al., 2014 ; Mahadevan 2016 ; Lévy et al., 2018)."

Ln 27: Here there is a change of topic, please, start a new paragraph.

We will rework the manuscript to take into account your suggestions.

Ln 24-27: Revise sentence. First, the sentence is too long.
In my opinion, the use of "could" makes the facts described less solid. It is established that fine-scale frontal structures induce vertical velocity. Is there any study where no vertical velocity is associated with these structures?

We will modify it in the manuscript, following your grammar suggestion.

Different physical processes associated with finescale structures are able to generate vertical velocities, such as deformations of the flow and spatial inhomogeneities (Giordani et al., 2006), eddy perturbation (Martin & Richards, 2001 ; Pilo et al., 2018), linear Ekman pumping (McGillicuddy et al., 1998 ; Gaube et al., 2015), or eddy-wind interactions (McGillicuddy et al., 2007). That is why to our knowledge the majority of studies about vertical velocities have focused on finescale structures (fronts or eddies) because these are suitable places for the formation of vertical motions (Rudnick, 1996 ; Pascual et al., 2015 ; Rousselet et al., 2019 ; Barceló-Llull et al., 2021).

Vertical velocities do not modulate light availability. Vertical velocities move the phytoplankton cells along the water column and depending on the "resulting" depth they will have more or less light.

We plan to add refs and to modify the sentence as follow:
"Previous studies have well established that vertical motions impact biogeochemistry (Mahadevan & Tandon, 2006 ; Mahadevan, 2016 ; McGillicuddy, 2016). Upward vertical velocities drive deep nutrients into the euphotic layer and also move the phytoplankton cells along the water column resulting in changing light conditions."

Ln 33-35: I do not understand this sentence. Are you saying that little is known about the phytoplankton diel cycle? Not only there are laboratory experiments, but also models, in particular individual-based models, that study this fact. For example, several

studies by Geider et al.

We just want to highlight the importance of performing in situ measurements at high frequency to sample the phytoplankton cycle in natural conditions. Although a lot of knowledge has been obtained from laboratory experiments or models, only a few studies have performed in situ measurements at high frequency and resolution. This is a reason to lead cruises using Lagrangian sampling strategy and automated flow cytometer.

Thank you for your reference, we will include it in the sentence as follows:

"Although progress in the understanding of phytoplankton cell cycle has been obtained from incubation, sample manipulation (Worden and Binder, 2003) and models (Geider et al., 1997 ; MacIntyre et al., 2000), performing in situ measurements at high frequency and resolution is a necessity to better understand these biological processes and their responses to the environment. An efficient solution is to lead Lagrangian cruises using automated flow cytometers sampling at high frequency in order to resolve the phytoplankton diurnal cycle in situ, which is challenging using more conventional methods such as cultures or counting by optical microscopy (Thyssen et al., 2008 ; Fontana et al., 2018)."

Ln 44-48: Please, rephrase these sentences with a clearer and simpler message.

We plan to modify the sentence as follow:

"In our previous work, we identified the two water masses as both Atlantic Water (AW), but at different stages of mixing due to their different routes in the Western Mediterranean Sea. The AW located south of the front, characterized by absolute salinity ($S_A$) between 37 g kg$^{-1}$ and 37.5 g kg$^{-1}$, corresponds to AW more recently entered into the Mediterranean Sea by the Gibraltar strait, thus being named "younger AW". The AW found north of the front is characterized by a higher $S_A$ (37.5 g kg$^{-1}$ to 38 g kg$^{-1}$) and corresponds to the surface water having circulated in the cyclonic gyre of the western Mediterranean for some years and then is referred to as "older AW". This water is also referred to as "local AW" (Barceló-Llull et al., 2019) or "resident AW" (Balbín et al., 2012)."

Ln 46-48: Could the authors provide some details about the nutrient concentration of both water masses?

Unfortunately, it was not possible for both technical and funding reasons to perform nutrient measurements during the 2018 cruise, that is why we cannot provide nutrient concentrations of both water masses. Nevertheless we acknowledge the importance of this information and these measurements are planned for our future cruise planned this spring (mid-April to mid-May 2023).

Ln 49-50: This sentence looks like part of the results section. I understand that you are referring to Tzortzis et al. (2021), but it is not clear.

We will modify the sentence as follows:

"Coupled with these hydrological measurements, measurements of phytoplankton abundance by flow cytometry were also described in our previous study. Tzortzis et al. (2021) showed that contrasted phytoplankton abundances were observed in these two water masses, with the smallest phytoplankton such as *Synechococcus* dominating south of the front in the younger AW, while microplankton is more abundant north of the front in the older AW."

Ln 51: This study, or Tzortzis et al. (2021)? I imagine is Tzortzis et al. (2021), then this first sentence and probably the open questions should be in the previous paragraph.

We apologize for the lack of clarity. We will not change the paragraph but we will modify the sentence as follows:

"As a consequence, our previous study constitutes an important improvement in the understanding of the role of frontal structures at finescale on phytoplankton distribution in a moderately energetic ocean. Nevertheless, open questions remain concerning the mechanisms generating this observed distribution: Is it exclusively driven by the dynamics of the ocean currents ? What is the role of biological processes ?

In the present study, we attempt to explain the particular patterns of phytoplankton abundances observed by automated flow cytometry during the PROTEVSMED-SWOT cruise [...]."

**Materials and methods**

Ln 70-73: I don't think this information is relevant.

We will move the sentence about SWOT in the perspective section.

Ln 73-76: These facts should be in the introduction.

As suggested, we will move this part in the new version of the introduction (see above)

Ln 77: Please, consider indicating that the measures have a high spatial and temporal resolution.
Ln 77: I am not sure if you can use in situ sensors when they are on board. It is kind of repetitive.
Ln 77-70: Here you are describing in situ measurements, that are described in the next subsection.
Ln 80-82: Repetitive information (introduction).
Ln 76 and 83: Please, describe the sampling strategy details in a single paragraph.
Ln 84 and 85: Please, mention the source of the remote sensing datasets.

Following your suggestions we will modify Figure 1 (see page 21 of this document) and we will modify this part as follows:

"The PROTEVSMED-SWOT cruise followed an adaptive Lagrangian strategy to measure at **high spatial and temporal resolution** several physical and biological variables with both in situ sensors and analysis of the sea surface water intake. The vessel route was designed ad-hoc on the basis of daily remote sensing dataset provided by the Software Package for an Adaptive Satellite-based Sampling for Oceanographic cruises (SPASSO, https://spasso.mio.osupytheas.fr, last access: February 6, 2023). SPASSO used altimetry-derived currents **from AVISO** ("Archiving, Validation and Interpretation of Satellite Oceanographic", https://www.aviso.altimetry.fr, last access: January 18, 2023 ) and ocean color observations. Chlorophyll a concentrations ([chl*a*], level 3, 1 km resolution) were provided by **CMEMS**, "Copernicus Marine Environment Monitoring Service", https://marine.copernicus.eu, last access: January 18, 2023. In addition, ocean color composite maps were provided by **CLS with the support of the CNES**). They were constructed using a simple weighted average over the previous 5 d of data gathered by the Suomi/NPP/VIIRS sensor. SPASSO generated maps of dynamical and biogeochemical structures in both near real time (NRT) and delayed time (DT).

Maps of [chl*a*] allowed us to identify two water masses, characterized by distinct [chl*a*] values and separated by a zonal front at around 38° 30' N. This front was also detected using the in situ horizontal velocities, temperature and salinity, as described in our previous study. These two water masses were sampled along a designated route of the ship, represented in black in Fig. 1. Special attention was paid to adapting the temporal sampling in order to measure the phytoplankton diel cycle in each water mass. This was achieved by continuously sampling across both water masses along transects. While the ship did not remain in each water mass for 24h, day-to-day variability remained low and measurements from several days were combined into one diel cycle (Fig. 1). The shape depicted by the ship's track led us to call these area north–south (NS) hippodrome (bold black line in Fig. 1) performed between 11 May at 02:00 and 13 May at 08:30 UTC."

Ln 91-94: What are the temporal and spatial resolution of the temperature and salinity measurements?

Temperature and salinity were measured thanks to a thermosalinograph (TSG). The TSG was equipped with two sensors: a CTD Sea-Bird Electronics SBE 45 sensor installed in the wet lab, connected to the surface water and which continuously pumped seawater at 3 m depth ; and an SBE 38 temperature sensor installed at the entry of the water intake. The TSG measurements are achieved **each 30 min,** which corresponds to a **~ 2 km spatial resolution** at typical ship speeds.

Ln 94-115: One paragraph. Moreover, revise the information provided, there is some repetitiveness regarding the optical signals.
Ln 104: 1.5 cm$^3$ is the water volume analyzed? Please, consider expressing the volume in mL, in my experience, it is a more common unit used in this kind of study.
Ln 112: Again, please consider using cell per mL
Ln 115-116: Totally out of place.

We used a subunit of the international system (SI) as requested by Biogeosciences, but below we will convert it into mL as in the reworked manuscript, following your suggestions.

We have reworked this section (Ln 91-116):

"During the cruise, the irradiance (wavelengths between 400 and 1000 nm) was measured by a CMP6 pyranometer (Kipp and Zonen; https://www.campbellsci.fr/cmp6, last access: January 19, 2023). The sea surface temperature and salinity were measured continuously with a **frequency of 30 min along the ship route** by a thermosalinograph (TSG), which is an underway sensor able to pump seawater at 3 m. The data were converted into conservative temperature ($\Theta$) and absolute salinity ($S_A$) using the TEOS-10 standards of McDougall et al. (2012). To automatically sample and analyze the phytoplankton cells, an automated CytoSense flow cytometer (CytoBuoy b.v.) was installed on board and connected to the seawater circuit of the TSG. A sheath fluid made of 0.1 μm filtered seawater stretched the sample in order to separate, align, center and drive the individual particles (i.e. cells) through a laser beam (488 nm wavelength). Several optical signals were recorded when each particle crossed the laser beam: the forward angle light scatter (FWS) and 90° side-ward angle scatter (SWS), related to the size and the structure (granularity) of the particles. Two distinct fluorescence emissions induced by the light excitation were also recorded, a red fluorescence (FLR) induced by chlorophyll a content and an orange fluorescence (FLO) induced by the phycoerythrin pigment content. The CytoUSB software (Cytobuoy b.v.) was used to configure and control the flow cytometer and set two distinct protocols, **running sequentially every 30 min**. A total of 1164 samples were analyzed during the cruise. The first protocol (FLR6) was dedicated to the analysis of the smaller phytoplankton, using a red fluorescence (FLR) trigger threshold fixed at 6 mV, and a volume analyzed set up at **1.5 mL**. The second protocol (FLR25) targeted nanophytoplankton and microphytoplankton with a FLR trigger level fixed at 25 mV and an analyzed volume of **4 mL**. The FLR trigger was used to discriminate the red fluorescing phytoplanktonic cells from other particles (such heterotrophic prokaryotes, nanoflagellates, ciliates, etc.). Recorded data were analyzed with the CytoClus software (Cytobuoy b.v.) which retrieves information from the 4 pulse shapes curves (FWS, SWS, FLO, FLR) obtained for every single cell. These curves were then projected into distinct two-dimensional planes (cytograms) by computing the curves' integral. Using a combination of various cytograms (e.g., FWS vs. FLR, FLO vs. FLR) allows us to determine optimal cell clusters (i.e, cells sharing similar optical properties). These clusters have been demonstrated in the literature to represent phytoplankton functional groups (PFGs) (Dubelaar and Jonker, 2000; Reynolds, 2006; Thyssen et al., 2008; Edwards et al., 2015). Finally, the PFGs abundance (**cells per milliliter**) and mean light scatter and fluorescence intensities were extracted from each sample."

Ln 118-124: Please, make it clear that this size-structured population model was applied to every phytoplankton population/group identified previously using the CytoBuoy.

We applied the size-structured population model of Sosik et al. (2003) to every phytoplankton group identified by the CytoSense flow cytometer. However, we only analyzed the results obtained for *Synechococcus* and the two nanophytoplankton groups because the size distributions of the picophytoplankton groups were very noisy. Furthermore, microphytoplankton and Cryptophytes were not abundant enough to allow a reliable determination of their abundances and cell cycle.

Ln 126: To use the model, the light scatter signal (FWS) recorded for each cell by the flow cytometer must be converted to size (diameter) using a power law relationship (Sosik et al., 2003), and then to biovolume (v).
I imagine that to convert size into volume you are considering that all the species are spherical. Then, please consider indicating this fact and that you are converting the FWS signal to Equivalent Spherical Diameter.Please, indicate the units of both measurements. Also for the rest of the variables (t, E, g, μ*,...).
Ln 128: I am not sure if N is the number of cells in all the size classes or at each size class.
Ln 133: How many size classes were determined and how? Does it follow a log distribution?
Ln 135: I am not sure what exactly is "this probability". Is it the probability of cells growing in a time interval? Is it a probability or a proportion?
Ln 141: Instead of however, besides seems more appropriate.
Ln 141-142: Repetitive information.
Ln 147: I consider kind of inappropriate the use of a "decrease in cell size", it is a division. A phytoplanktonic cell decreases in size if the growth conditions are not optimal, and that is not an indication that there is a doubling event.
Ln 150-151: This sentence is confusing. Why do you talk about N(0) when is not used in the equations 5. (Two equations = two labels, please. Similarly, with equations 8)
Ln 153: A(t) is a tridiagonal transition matrix that contains.
Ln 159: Could you elaborate on what you mean by optimal parameters, please?
Ln 160: Standard deviations of the errors?
Ln 166: There is no information about this equation.
Ln 167-169: I do not understand this explanation.
Moreover, the definition of "bar l" (I do not know how to write the loss symbol here) confuses me. If it is the daily average population loss rate, how dt is 1 hour? On the other hand, what do you mean exactly by loss? The number of cells moving from one size class to another, or death?
What is the description of T1day NT0?

I have no experience using this kind of model, but any reader should be able to understand the methodology followed in the study without having to read previous studies. So please, review this section carefully and try to make it as clear as possible.

In order to follow all your comments above, we will completely rework section 2.3 of the manuscript, concerning the size-structured population model. We hope that this new version here below will clarify this approach.

**The size-structured population model**

We used the size-structured population model described by Sosik et al. (2003) and adapted by Dugenne et al. (2014) and Marrec et al. (2018), to estimate phytoplankton in situ growth rates.

The model uses as input the phytoplankton cell volume (biovolume) derived from cell light scatter intensities (FWS). Biovolumes were estimated using coefficients previously obtained by measuring a set of silica beads with the flow cytometer following the same settings used for phytoplankton analysis. The coefficients $\beta_0$ and $\beta_1$ used to convert FWS (arbitrary units, a.u.) to biovolume $v$ ($\mu m^3$) were derived from a log-log regression between FSW and silica bead volumes. These methods come from the studies of Koch et al. (1996) and Foladori et al. (2008).

$v = exp(\beta_0) \times FWS^{\beta_1}$     with in our case $\beta_1 = 0.9228$ and $\beta_0 = -5.8702$

In the size-structured population model, cells are classified into several size classes according to their dimensions at time $t$. $\Delta v$ is chosen in order to have enough number of classes $m$ to cover the entire observed biovolume $v$, from $v_1$ to $v_m$ (cf Figure A). Classes are logarithmically spaced as follows:

for i in 1,2,…,m      $v_i = v_1 \, 2^{(i-1)\Delta v}$      with $\Delta v$ constant

For *Synechococcus,* $\Delta v = \frac{1}{6}$ and $m = 40$, so that the model size classes encompassed our full measured size distributions.

[Figure]

Figure A: Cell cycle stages in the size-structured population model. Cells may grow to the next size class ($\gamma$) or be at equilibrium *(1- $\gamma$(t)) (1-$\delta$(v,t))*. Above a particular size, cells are large enough to divide in two daughter cells with probability (*$\delta$*). Figure adapted from Sosik et al. (2003).

At any time *t*, the number of cells in size classes $\vec{N}$ (and $\vec{w}$ its corresponding normalized distribution), was projected to *t + dt* via matrix multiplication:

$$\vec{N}(t + dt) = A(t)\ \vec{N}(t) \qquad \text{and} \qquad \vec{w}(t + dt) = \frac{A(t)\ \vec{N}(t)}{\sum A(t)\ \vec{N}(t)}$$

We chose *dt* = 10 min (i.e., $\frac{10}{60}$ h) as Sosik et al. (2003) and Dugenne et al. (2014), because for this time step, cells are unlikely to grow more than one size class.

*A(t)* is a tridiagonal transition matrix that contains (cf Figure B):
  1) *$\gamma$*: the probability of cellular growth
  2) *$\delta$*: the probability of cells entering mitosis
  3) the cells stasis, i.e., the probability for cells to maintain their state (i.e size) in equilibrium during the temporal projection.

$$\begin{array}{c c}
& \begin{array}{ccccc} v_1(t) & \dots & v_j(t) & \dots & v_m(t) \end{array} \\
\begin{array}{c} v_1(t+dt) \\ \vdots \\ v_j(t+dt) \\ \vdots \\ v_m(t+dt) \end{array} &
\left( \begin{array}{ccccc}
1-\gamma(t) & \dots & 2.\delta(v_j,t) & \dots & 0 \\
0 & \ddots & (1-\gamma(t)). & \ddots & 0 \\
& & (1-\delta(v_j,t)) & & \\
0 & \ddots & 0 & \ddots & 1-\delta(v_m,t)
\end{array} \right)
\end{array}$$

Figure B: Matrix transition A(t). (Figure extracted from Dugenne, 2017 (thesis)).

The temporal projection

$N_{|v=v1}\ (t+dt) = (1-\gamma(t)).N_{v=v1}(t) + 2\delta(v_j,\ t).N_{v=vj}(t)$

$N_{|v=vm}\ (t+dt) = (1-\delta(t)).N_{v=vm}(t) + \gamma(t).N_{v=vm-1}(t)$

Probability of cellular growth

The probability of cells growing to the next size class ($\gamma$) depends only on the light intensity (irradiance) necessary for photosynthesis, expressed as:

$\gamma(t) = \gamma_{max}\ [\ 1 - exp(-\ \frac{E(t)}{E^*})]$

$\gamma_{max}$: maximum proportion of cells growing (dimensionless quantity)
E: irradiance ($\mu E\ m^{-2}\ s^{-1}$)
E*: irradiance normalizing constant ($\mu E\ m^{-2}\ s^{-1}$)

Probability of cells entering mitosis

According to Dugenne et al. (2014), $\delta$ expresses a proportion (between 0 and 1) modeled by the combination of two Normal distributions ($\mathcal{N}$). One is linked to the cell size, the other is linked to the time of cell division. Both imply an optimum, reached at $\bar{v}$ and $\bar{t}$ respectively, for cell division above which the cell size and the timing of division is suboptimal.

$\delta(t,\ v) = \delta_{max}\ \mathcal{N}(\bar{v},\sigma_v^2)\ \mathcal{N}(\bar{t},\sigma_t^2)$

$\delta_{max}$: maximum proportion of cells entering mitosis (dimensionless quantity)
$\bar{v}$: mean of the size Normal distribution ($\mu m^3$)
$\sigma_v$: standard deviation of the size Normal distribution ($\mu m^3$)

$\overline{t}$: mean of the time Normal distribution (h)

$\sigma_t$: standard deviation of the time Normal distribution (h)

**Cells stasis**

A third functional proportion is included in the transition matrix *A(t)*, to represent cell stasis. Since this function illustrates a non-transition, it is modeled by the proportion of cells that neither divided nor grew between *t* and *t + dt*.

*[1 -γ(t)] [1 - δ(t, v)]*

**Optimal parameters**

The set of parameters, $\theta$ is estimated by maximum likelihood function, assuming errors between observed $\vec{w}$ and predicted $\widehat{w}$ normalized size distributions. Their standard deviations are estimated by a Markov Chain Monte Carlo approach (Geyer, 1992 ; Neal, 1993) that sample $\theta$ from their prior density distribution, obtained after running 200 optimizations on bootstrapped residuals to approximate the parameter posterior distribution using the normal likelihood. (The likelihood function represents the probability of random variable realizations conditional on particular values of the statistical parameters).

$$\theta = [\gamma_{max}, E^*, \delta_{max}, \overline{v}, \sigma_v, \overline{t}, \sigma_t] = argmin(\textstyle\sum(\theta))$$

$$\sum(\theta) = \sum_{t}^{t+dt} \sum_{i=1}^{m} (w(t) - \widehat{w}(t,\theta))^2$$

$$\widehat{w}(t,\theta) = A(t\text{-}dt, \theta)\, \vec{w}(t\text{-}dt) \quad \text{and} \quad \widehat{N}(t,\theta) = A(t\text{-}dt, \theta)\, \vec{N}(t\text{-}dt)$$

Table: Model parameters being optimized.

| Parameters | Definition | Interval | Units |
|---|---|---|---|
| $\gamma_{max}$ | Maximum proportions of cells in growing phase | [0, 1] | ∅ |
| $E^*$ | Irradiance normalizing constant | [0,∞[ | µE m$^{-2}$ s$^{-1}$ |

| $\delta_{max}$ | Maximum proportion of cells entering mitosis | [0,1] | Ø |
|---|---|---|---|
| $\overline{v}$ | Mean of the size Normal distribution | $[v_{min}, v_{max}]$ | µm³ |
| $\sigma_v$ | Standard deviation of the size Normal distribution | $[10^{-06}, \infty[$ | µm³ |
| $\overline{t}$ | Mean of the time Normal distribution | $[1, 24\frac{1}{dt}+1]$ | hours |
| $\sigma_t$ | Standard deviation of the time Normal distribution | $[10^{-06}, \infty[$ | hours |

Growth rate and loss rate

Once optimal parameters are identified, the growth rate is calculated using the time projection of the initial size distribution $\overset{\frown}{N}$. 200 iterations were run to estimate the standard deviation of group-specific growth rates.

$$\mu_{size} = \frac{1}{t+dt} ln\left(\frac{\sum\limits_{i=1}^{m} \widehat{N}_i(t+dt)}{\sum\limits_{i=1}^{m} \overset{\frown}{N}_i(t)}\right)$$

$\overset{\frown}{N}$: observed size distribution at $t = 0$ (cells cm⁻³)
$i$: iᵗʰ size class
$\widehat{N}$: predicted size distribution (cells cm⁻³)
$m$: number of size classes
$dt$: time step (h)
$t + dt$: temporal integration of the distribution projection (h), $t + dt = 24$ h
$\mu_{size}$: growth rates (day⁻¹)

The model estimates a population intrinsic growth rate $\mu_{size}$, and a specific loss rate, over a 24 h period.

The daily average population loss rate $\bar{l}$ is obtained by the difference between the intrinsic growth rates $\mu_{size}$ and the hourly logarithmic of observed size distribution $N$.

$$\bar{l} = \int\limits^{dt} \mu_{size}(dt) \ - \ \frac{1}{dt} \ ln(\frac{N^{\check{}}(t+dt)}{N^{\check{}}(t)})$$

As a consequence, the net growth rate is expressed as $r(t) = \mu_{size}(t) - l(t)$.

**Results**

Ln 176-188: This information should be included in the methodology section. Also, at the end of this explanation, it will be interesting to indicate how to convert the scatter signal to size and volume.The details about how every species was differentiated, in my opinion, are not necessary, therefore I propose the authors move it to the supplementary, together with Figure 2.

Following your suggestions, we will modify the text by moving: the description of how to identify the phytoplankton groups in the methodology section, and the remaining information and the figure to the supplementary material.

Ln 210-211: The information about the figures does not fit here. It will be more appropriate to move to the beginning of the next paragraph.

We will move it in the reworked version of the manuscript.

On the other hand, please explain the background information. Does it make reference to the proportion (percentage) of cells of each biovolume? If it is a percentage, why it does not vary between 0 and 1?

We will change the terms, instead of proportion we will use frequency; indeed, the background colors represent the number of particles of a specific biovolume. For a given time, the sum of the frequency is equal to the total number of particles.

Ln 211-216: How was reconstructed the 24-hour irradiance curve should be explained in the methodology.

We will move this part in the methodology.

Ln 221, 222, 231, and 232: Please, include the standard deviation value together with the mean value.

We will add that in the text and maintain it in the table, too.

Ln 217-233: Please, explain in this section why there is no information about the other 6 groups identified.

We will move the explanation from the Discussion in this section:
"We have also modeled the diurnal cycle for the picophytoplankton groups, i.e., Pico1, Pico2, Pico3, PicoHFLR. However, we obtained very noisy size distributions and couldn't obtain a valid measurement of the growth rates, hence these distributions are not considered further in this study. As microphytoplankton and Cryptophytes were not abundant enough to allow a reliable determination of their abundances and cell cycles, they were not taken into consideration in this study."

**Discussion**

Ln 236: Please, add some references.

Although it has been clearly demonstrated that phytoplankton plays a fundamental role in the ocean ecosystem functioning **(Watson et al., 1991 ; Field et al., 1998 ; Allen et al., 2005)**, numerous questions remain about their population dynamics in relation to finescale structures.

Ln 245-246: What do you mean by transiting in all the cell cycle stages? That they are growing and dividing?
Ln 254: What do you mean by extensive distribution?

We apologize for the lack of clarity, our intent here is to compare the *Synechococcus* distributions in the two different water masses. We will modify the text as follows:
"In the older AW the cells have all along the day a large size-class distribution centered approximately at 0.3 $\mu m^3$ while in the younger AW (Fig. 4a, 4c) the distribution is narrower and centered approximately at 0.2 $\mu m^3$ (Fig. 4b, 4d)."

Ln 265: Is it really the only reference for this fact?

There are others references, we will modify the sentence as follows:
"Picophytoplankton is often characterized by the presence of several taxa with potentially different effects on the population dynamics, whereas nanophytoplankton is mostly dominated by diatoms in the Mediterranean Sea **(Siokou-Frangou et al., 2010 ; Marty et al., 2002 ; Navarro et al., 2014 ; El Hourany et al., 2019)**."

Ln 269-274: Basically the same was said in the introduction.
Ln 275: Please, revise this sentence.

We will remove these sentences.

Ln 284: The fact that light and irradiance are essential for phytoplankton growth was known before 2001.

Here we referred to the fact that vertical velocities impact biogeochemistry, driving nutrients in the euphotic layers and the phytoplankton cells along the water column where these organisms will receive more or less light as a function of the depth (see our previous answer).

We agree that our sentence was not clear, so we will rework these sentences as follows:
"The early experimental works of Huisman, 1999 ; Jenkin et al., 1937 and Marshall et al., 1928 have well established that the light and nutrients are essential for phytoplankton growth. The availability of these two variables for the phytoplankton is driven by physical dynamics such as vertical velocities (Lévy et al., 2001 ; Pidcock et al., 2016 ; Mahadevan, 2016)."

Ln 285: Then is expected a higher nutrient concentration in the old AW? For that reason, there is a higher contribution of larger cells?

Yes, indeed we expect a higher nutrient concentration in the older AW since this water probably experiences more upwelling  and longer temporal vicinity of the coast along its route (Bethoux, 1989 ; Schroeder et al., 2010 ; see introduction). Moreover, it is known that a higher concentration of nutrients is favorable to larger cells. Indeed, their better surface:size ratio due to their small size confers them a better capacity to inhabit areas with very low nutrient concentration compared to larger phytoplankton (Kiørboe, 1993 ; Marañón, 2015).

**Conclusions and perspectives**

Ln 305-309: This is not a conclusion.

We used to begin the conclusion section with a short summary of the thematics. We will completely rework the Conclusion taking into account your comments and those of the other referee:

"Phytoplankton structure and dynamics are a complex result of many interacting biological and physical phenomena. Finescale structures, and in particular fronts, generate vertical velocities which displace phytoplankton cells and nutrients in the water column, thus influencing phytoplankton communities. These mechanisms are only partially understood because the spatial scale of these structures and their ephemeral nature make them particularly difficult to study in situ; as a consequence only a few studies have been performed in finescale frontal regions. The estimates of specific growth rates for the various phytoplankton groups is one of  the keys to better understand how environmental conditions affect phytoplankton dynamics. In this study, we followed the dynamics of several phytoplankton groups in two distinct water masses both in terms of hydrology and phytoplankton abundances, in order to explain their particular distribution.

The originality of our work resides in the fact that we used a size-structured population model applied in two water masses identified using a Lagrangian sampling strategy. To our knowledge that has never been done before. This strategy allowed us to reconstruct the diurnal cycle for several phytoplankton groups and to identify contrasted dynamics in the two water masses. For *Synechococcus* and nanophytoplankton, we found the higher cell size in the older AW located to the north of the front, associated with the lower abundances. A possible explanation is that the older AW is more enriched in nutrients than the younger AW, thus favoring the largest cells. This remains a hypothesis because of a lack of nutrient data. Furthermore, in the oligotrophic region such as the Mediterranean Sea, a narrow trophic variation of the nutrient concentration is sufficient to generate higher abundance of phytoplankton. Another novelty of our study is that we applied this model on several phytoplankton groups identified by flow cytometry, whereas previous studies only applied it to *Synechococcus* and *Prochlorococcus*. We obtained good results for *Synechococcus* and nanophytoplankton. However, our results were noisy for picophytoplankton groups probably because these latter contain several taxa with differing dynamics.

Our work paves the way for many research perspectives. Direct integration of growth rates in biogeochemical models (Cullen et al., 1993) should be taken into account for a better assessment of the biogeochemical contribution of phytoplankton in oligotrophic ecosystems and to better forecast its evolution in the context of global change. Furthermore, we plan future experiments again in the South Western Mediterranean in spring 2023, after the launch of the SWOT satellite which will provide high resolution altimetry-derived current. Involving high-resolution nutrient measurements (and also high-precision ones, considering the oligotrophy of the Mediterranean Sea), coupled with metabarcoding (to address the biodiversity of phytoplankton), zooplankton and virus sampling, we will improve the understanding of zooplankton grazing and viral lysis on the different phytoplankton groups. Furthermore, we aim to explore how biogeochemical and ecological role of the finescales in regions of weak circulation are different from the ones more documented in highly energetic regions like boundary currents. In the Mediterranean sea, the low nutrient content is indeed the perfect condition when addressing this question, because even weak horizontal or vertical nutrient redistributions associated with the finescale circulation are likely to result in a biological response (Talmy et al., 2014 ; Hashihama et al., 2021)."

**Technical corrections**

Ln 6: Delete the space between "numerous" and ";".
Ln 21: Add a comma after (days-weeks).
Ln 41: Delete parenthesis after altimetry.
Ln 87: Once the front "is" localized.
Ln 103: 1164 samples "were" analyzed.
Ln 122: Maybe light availability is more adequate?
Ln 129: Please, consider changing investigated by counted.
Ln 134 and141: ... between the time interval t...
Ln 135-136: ... necessary to carry out photosynthesis?
Ln 157: Is there a typo? The probability of division is not denoted by $\gamma$?

Ln 163-164: You already defined those symbols; it is kind of redundant to do it again.

Ln 185: [chla]?

Ln 207: Please, consider using disregarding instead of eliminating.

Ln 227: Observed biovolume (observed and in situ are kind of repetitive), and predicted biovolume (check also Ln 219).

Ln 219-220: all species populations in both water masses?

Ln 222: No comma before the parenthesis.

Ln 221, 223, 224, 229, 230, 240: l or "bar l"?

Ln 239-240: The structure of the phytoplankton community.

Delete the point after the manuscript title and after the abstract.

Please, use 1 or 2 decimal numbers for all the variables measurements, to keep the format along the manuscript (e.g., Ln 46, 48, 221).

Please, use the same format for the dates along the manuscript (e.g., Ln 70 and 93).

Please, revise the use of the word indeed, it is repeated quite often throughout the text.

Thank you, we will correct that in the reworked manuscript.

**Figure 1.**

Panel a is very small, impossible to appreciate the information. Moreover, the colormap scale is minuscule and does not indicate the variable (and units) that represents.

In panel b, it would be interesting to indicate where the sampling events took place.

In panel c, in my opinion, the clock diagrams are not necessary.

Legend: The purple box encloses a (b) zoom of the sampling region with overlaid chlorophyll-a concentration (units). ______. The red line represents _____, the dark blue box ______, and the light blue box _______.

I am not an English native, but I think that the lines and boxes are superimposed to the chl map. The other way around will not allow you to see lines and boxes.

We have modified Figure 1 (see below). In panel a, we have increased the size of the map and colormap. We have also changed the color of the route of the ship for more visibility, and we added the units. In panel b, we have also increased the map. The sampling events took place all along the transects, that is why for us it is not necessary to indicate that in addition. Concerning panel c, we kindly disagree, because we think that the clock diagrams help the reader to understand the reconstruction of a day of 24 h period thanks to the 4 transects in each water mass.

[Figure]

Figure 1: (a) Route of the RV Beautemps-Beaupré during the PROTEVSMED-SWOT cruise. The purple box encloses a (b) zoom of the sampling region with overlaid chlorophyll-a concentration (µg L⁻¹) of 11 May 2018. In panel (b) black dotted line represents the route of the ship and the bold black line represents the route of the Lagrangian sampling across the older AW (delimited by the box in dark blue) and the younger AW (delimited by the box in light blue). (c) Dates of the transects across the older AW and the younger AW, used to reconstruct a day of 24 h period in each water mass.

**Figure 2.**

As previously indicated, I do not consider this figure of relevance to the main text.

We will move this figure in supplementary information.

**Figure 2.**

Legend: Background colors indicate the two water masses…

We think that you speak about figure 3 ? In this case, indeed the background colors indicate the two water masses.

**Figures 4-6.**

Explain what represents the red line and the background color.
Correct all the color bars (by figure) to vary all in the same range.
We will modify the caption, the colorbars and the colormaps as follows:

[Figure]

Figure 4: The background color represents the *Synechococcus* cell size distribution (i.e., biovolume in μm³) observed (a, b) and predicted by the model (c, d) in the older AW (a, c) and in the younger AW (b, d) during 24 h. The line represents the irradiance (μE m⁻² s⁻¹).

[Figure]

Figure 5: The background color represents the RNano cell size distribution (i.e., biovolume in μm³) observed (a, b) and predicted by the model (c, d) in the older AW (a, c) and in the younger AW (b, d) during 24 h. The line represents the irradiance (μE m⁻² s⁻¹).

[Figure]

Figure 6: The background color represents the SNano cell size distribution (i.e., biovolume in μm³) observed (a, b) and predicted by the model (c, d) in the older AW (a, c) and in the younger AW (b, d) during 24 h. The line represents the irradiance (μE m⁻² s⁻¹).

**Figure 7.**

A very small figure, with some details difficult to appreciate. Even the legend is difficult to read.

We hope this new arrangement makes the figure clearer.

[Figure]

Figure 7: The contrasted distribution of phytoplankton in the frontal area. The circles represent the abundances of the several phytoplankton groups in the two water masses separated by the front. The boxes indicate the biovolume ($v$) and the growth rates ($\mu$) for each phytoplankton group, as estimated from the model. Figure adapted from Tzortzis et al. (2021).

**Table 2.**

Indicate also that there is information about the standard deviation.

As mentioned in the methodology, statistics are included in the model. The growth rates were estimated using the maximum likelihood function and 200 iterations were run to estimate the standard deviation of group-specific growth rates using a Markov Chain Monte Carlo (Geyer, 1992 ; Neal, 1993).

Define every variable on its own.
This will be done.

$\mu$ ratio should not be adimensional? The equation and its meaning are already defined in the text.
As described in the new version of the methods (see above) $\mu$ is a temporal rate.

Define the acronym PFG.
We mentioned in the manuscript (Ln 111) that PFG(s) means "phytoplankton functional groups".

**References**

Allen, John T., Louise Brown, Richard Sanders, C. Mark Moore, Alexander Mustard, Sophie Fielding, Mike Lucas et al., Diatom carbon export enhanced by silicate upwelling in the northeast Atlantic. *Nature* 437, no. 7059: 728-732, https://doi.org/10.1038/nature03948, 2005.

Balbín, R., Flexas, M. d. M., López-Jurado, J. L., Peña, M., Amores, A., and Alemany, F.: Vertical velocities and biological consequences at a front detected at the Balearic Sea, *Cont. Shelf. Res.*, 47, 28–41, https://doi.org/10.1016/j.csr.2012.06.008, 2012.

Balbín, R., López-Jurado, J. L., Flexas, M., Reglero, P., Vélez-Velchí, P., González-Pola, C., Rodríguez, J. M., García, A., and Alemany, F.: Interannual variability of the early summer circulation around the Balearic Islands: driving factors and potential effects on the marine ecosystem, *J. Mar. Syst.*, 138, 70–81, https://doi.org/10.1016/j.jmarsys.2013.07.004, 2014.

Barceló-Llull, B., Pascual, A., Ruiz, S., Escudier, R., Torner, M., and Tintoré, J.: Temporal and spatial hydrodynamic variability in the Mallorca channel (western Mediterranean Sea) from 8 years of underwater glider data, *J. Geophys. Res.-Oceans*, 124, 2769–2786, https://doi.org/10.1029/2018JC014636, 2019.

Barceló-Lull, B., Pascual, A., Sánchez-Román, A., Cutolo, E., d'Ovidio, F., Fifani, G., Ser-Giacomi, E., Ruiz, S., Mason, E., Cyr, F., Doglioli, A., Mourre, B., Allen, J. T., Alou-Font, E., Casas, B., Díaz-Barroso, L., Dumas, F., Gómez-Navarro, L., and Muñoz, C.: Fine-Scale Ocean Currents Derived From in situ Observations in Anticipation of the Upcoming SWOT Altimetric Mission, *Front. Mar. Sci.*, 8, 1070, https://doi.org/10.3389/fmars.2021.679844, 2021.

Barrillon, Stéphanie, Robin Fuchs, Anne A. Petrenko, Caroline Comby, Anthony Bosse, Christophe Yohia, Jean-Luc Fuda et al. Phytoplankton reaction to an intense storm in the north-western Mediterranean Sea. *Biogeosciences* 20, no. 1: 141-161, https://doi.org/10.5194/bg-20-141-2023, 2023.

Barton, A. D., Ward, B. A., Williams, R. G., and Follows, M. J.: The impact of fine-scale turbulence on phytoplankton community structure, *Limnol. Oceanogr.*, 4, 34–49, https://doi.org/10.1215/21573689-2651533, 2014.

Behrenfeld, Michael J., Robert T. O'Malley, David A. Siegel, Charles R. McClain, Jorge L. Sarmiento, Gene C. Feldman, Allen J. Milligan, Paul G. Falkowski, Ricardo M. Letelier, and Emmanuel S. Boss. Climate-driven trends in contemporary ocean productivity. *Nature* 444, no. 7120: 752-755, https://doi.org/10.1038/nature05317, 2006.

Bethoux, J.: Oxygen consumption, new production, vertical advection and environmental evolution in the Mediterranean Sea, *Deep-Sea Res.*, 36, 769–781, https://doi.org/10.1016/0198-0149(89)90150-7, 1989.

Boyd, P. W., and P. P. Newton. Does planktonic community structure determine downward particulate organic carbon flux in different oceanic provinces ? *Deep Sea Research Part I: Oceanographic Research Papers* 46, no. 1: 63-91, https://doi.org/10.1016/S0967-0637(98)00066-1, 1999.

Capotondi, Antonietta, Michael A. Alexander, Nicholas A. Bond, Enrique N. Curchitser, and James D. Scott. Enhanced upper ocean stratification with climate change in the CMIP3 models. *Journal of Geophysical Research: Oceans* 117, no. C4, https://doi.org/10.1029/2011JC007409, 2012.

Cheng, Lijing, Karina von Schuckmann, John P. Abraham, Kevin E. Trenberth, Michael E. Mann, Laure Zanna, Matthew H. England et al.: Past and future ocean warming. *Nature Reviews Earth & Environment*: 1-19, https://doi.org/10.1038/s43017-022-00345-1, 2022.

Clayton, S., Nagai, T., and Follows, M. J.: Fine scale phytoplankton community structure across the Kuroshio Front, *J. Plankton. Res.*, 36, 1017–1030, https://doi.org/10.1093/plankt/fbu020, 2014.

Clayton, S., Lin, Y.-C., Follows, M. J., and Worden, A. Z.: Co-existence of distinct Ostreococcus ecotypes at an oceanic front, *Limnol. Oceanogr.*, 62, 75–88, https://doi.org/10.1002/lno.10373, 2017.

Collins, Mat, Soon-Il An, Wenju Cai, Alexandre Ganachaud, Eric Guilyardi, Fei-Fei Jin, Markus Jochum et al. The impact of global warming on the tropical Pacific Ocean and El Niño. *Nature Geoscience* 3, no. 6: 391-397, https://doi.org/10.1038/ngeo868, 2010.

Dugenne, M., Thyssen, M., Nerini, D., Mante, C., Poggiale, J.-C., Garcia, N., Garcia, F., and Grégori, G. J.: Consequence of a sudden wind event on the dynamics of a coastal phytoplankton community: an insight into specific population growth rates using a single cell high frequency approach, *Front. Microbiol.*, 5, 485, https://doi.org/10.3389/fmicb.2014.00485, 2014.

Dugenne, Mathilde. Dynamique du phytoplancton en mer Méditerranée: Approches par mesures à haute fréquence, modélisation, et statistiques bayésiennes. PhD diss., Aix-Marseille, 2017.

Doherty, Shannon C., Amy E. Maas, Deborah K. Steinberg, Brian N. Popp, and Hilary G. Close. Distinguishing zooplankton fecal pellets as a component of the biological pump using compound‑specific isotope analysis of amino acids. *Limnology and oceanography* 66, no. 7: 2827-2841, https://doi.org/10.1002/lno.11793, 2021.

El Hourany, Roy, Marie Abboud‑abi Saab, Ghaleb Faour, Carlos Mejia, Michel Crépon, and Sylvie Thiria. Phytoplankton diversity in the mediterranean sea from satellite data using self‑organizing maps. *Journal of Geophysical Research: Oceans* 124, no. 8 (2019): 5827-5843, https://doi.org/10.1029/2019JC015131, 2019.

Field, C. B., Behrenfeld, M. J., Randerson, J. T., and Falkowski, P.: Primary production of the biosphere: integrating terrestrial and oceanic components, *Science*, 281, 237–240, https://doi.org/10.1126/science.281.5374.237, 1998.

Foladori, Paola, Alberto Quaranta, and Giuliano Ziglio. Use of silica microspheres having refractive index similar to bacteria for conversion of flow cytometric forward light scatter into biovolume. *Water research* 42, no. 14 : 3757-3766, https://doi.org/10.1016/j.watres.2008.06.026, 2008.

Fontana, S., Thomas, M. K., Moldoveanu, M., Spaak, P., and Pomati, F.: Individual-level trait diversity predicts phytoplankton community properties better than species richness or evenness, *The ISME journal*, 12, 356–366, https://doi.org/10.1038/ismej.2017.160, 2018.

Gaube, P., Chelton, D. B., Samelson, R. M., Schlax, M. G., & O'Neill, L. W.: Satellite observations of mesoscale eddy-induced Ekman pumping. *Journal of Physical Oceanography*, 45(1), 104–132, https://doi.org/10.1175/JPO-D-14-0032.1, 2015.

Geyer, Charles J. Practical markov chain monte carlo. *Statistical science* (1992): 473-483.

Giordani, H., Prieur, L., & Caniaux, G.: Advanced insights into sources of vertical velocity in the ocean. *Ocean Dynamics*, 56(5-6), 513–524, https://doi.org/10.1007/s10236-005-0050-1, 2006.

Guidi, Lionel, Lars Stemmann, George A. Jackson, Frédéric Ibanez, Hervé Claustre, Louis Legendre, Marc Picheral, and Gabriel Gorskya. Effects of phytoplankton community on production, size, and export of large aggregates: A world‑ocean analysis. *Limnology and Oceanography* 54, no. 6: 1951-1963, https://doi.org/10.4319/lo.2009.54.6.1951, 2009.

Huisman, Jef. Population dynamics of light‑limited phytoplankton: microcosm experiments. *Ecology* 80, no. 1: 202-210, https://doi.org/10.1890/0012-9658(1999)080[0202:PDOLLP]2.0.CO;2, 1999.

Hays, Graeme C., Anthony J. Richardson, and Carol Robinson. Climate change and marine plankton. *Trends in ecology & evolution* 20, no. 6: 337-344, https://doi.org/10.1016/j.tree.2005.03.004, 2005.

Hilligsøe, Karen Marie, Katherine Richardson, Jørgen Bendtsen, Lise-Lotte Sørensen, Torkel Gissel Nielsen, and Maren Moltke Lyngsgaard. Linking phytoplankton community size composition with temperature, plankton food web structure and sea–air $CO_2$ flux. *Deep Sea Research Part I: Oceanographic Research Papers* 58, no. 8: 826-838, https://doi.org/10.1016/j.dsr.2011.06.004, 2011.

Jenkin, Penelope M. Oxygen production by the diatom Coscinodiscus excentricus Ehr. in relation to submarine illumination in the English Channel. *Journal of the Marine Biological Association of the United Kingdom* 22, no. 1: 301-343, https://doi.org/10.1017/S0025315400012030, 1937.

Kiørboe, T.: Turbulence, phytoplankton cell size, and the structure of pelagic food webs, in: Adv. Mar. Biol., vol. 29, pp. 1–72, Elsevier, https://doi.org/10.1016/S0065-2881(08)60129-7, 1993.

Koch, Arthur L., Betsy R. Robertson, and Don K. Button. Deduction of the cell volume and mass from forward scatter intensity of bacteria analyzed by flow cytometry. *Journal of microbiological methods* 27, no. 1 : 49-61, https://doi.org/10.1016/0167-7012(96)00928-1, 1996.

Laws, Edward A., Paul G. Falkowski, Walker O. Smith Jr, Hugh Ducklow, and James J. McCarthy. Temperature effects on export production in the open ocean. *Global biogeochemical cycles* 14, no. 4: 1231-1246, https://doi.org/10.1029/1999GB001229, 2000.

Lévy, M., Jahn, O., Dutkiewicz, S., Follows, M. J., and d'Ovidio, F.: The dynamical landscape of marine phytoplankton diversity, *J. Roy. Soc. Interface*, 12, 20150481, https://doi.org/10.1098/rsif.2015.0481, 2015.

Lévy, M., Franks, P., and Smith, K.: The role of submesoscale currents in structuring marine ecosystems, *Nat. Commun.*, 9, 4758, https://doi.org/10.1038/s41467-018-07059-3, 2018.

Li, G., Cheng, L., Zhu, J. *et al.* Increasing ocean stratification over the past half-century. *Nat. Clim. Chang.* **10**, 1116–1123, https://doi.org/10.1038/s41558-020-00918-2, 2020.

Lomas, M., Roberts, N., Lipschultz, F., Krause, J., Nelson, D., and Bates, N.: Biogeochemical responses to late-winter storms in the Sargasso Sea. IV. Rapid succession of major phytoplankton groups, *Deep-Sea Res. Pt. I*, 56, 892–908, https://doi.org/10.1016/j.dsr.2009.03.004, 2009.

MacIntyre, Hugh L., Todd M. Kana, and Richard J. Geider. The effect of water motion on short-term rates of photosynthesis by marine phytoplankton. *Trends in plant science* 5, no. 1 (2000): 12-17, https://doi.org/10.1016/S1360-1385(99)01504-6, 2000.

Mahadevan, Amala, and Amit Tandon. An analysis of mechanisms for submesoscale vertical motion at ocean fronts. *Ocean Modelling* 14, no. 3-4: 241-256, https://doi.org/10.1016/j.ocemod.2006.05.006, 2006.

Mahadevan, A.: The impact of submesoscale physics on primary productivity of plankton, *Annu. Rev. Mar. Sci.*, 8, 161–184, https://doi.org/10.1146/annurev-marine-010814-015912, 2016.

Marañón, E.: Cell size as a key determinant of phytoplankton metabolism and community structure, *Annu. Rev. Mar. Sci.*, 7, 241–264, https://doi.org/10.1146/annurev-marine-010814-015955, 2015.

Marrec, P., Grégori, G., Doglioli, A. M., Dugenne, M., Della Penna, A., Bhairy, N., Cariou, T., Hélias Nunige, S., Lahbib, S., Rougier, G., Wagener, T., and Thyssen, M.: Coupling physics and biogeochemistry thanks to high-resolution observations of the phytoplankton community structure in the northwestern Mediterranean Sea, *Biogeosciences*, 15, 1579–1606, https://doi.org/10.5194/bg-15-1579-2018, 2018.

Marshall, Sheina M., and A. P. Orr. The photosynthesis of diatom cultures in the sea. *Journal of the Marine Biological Association of the United Kingdom* 15, no. 1: 321-360, https://doi.org/10.1017/S0025315400055703, 1928.

Martin, A. P., & Richards, K. J.: Mechanisms for vertical nutrient transport within a North Atlantic mesoscale eddy. *Deep Sea Research Part II:* Topical Studies in Oceanography, 48(4-5), 757–773, https://doi.org/10.1016/S0967-0645(00)00096-5, 2001.

Marty, Jean-Claude, Jacques Chiavérini, Marie-Dominique Pizay, and Bernard Avril. Seasonal and interannual dynamics of nutrients and phytoplankton pigments in the western Mediterranean Sea at the DYFAMED time-series station (1991–1999). *Deep Sea Research Part II: Topical Studies in Oceanography* 49, no. 11: 1965-1985, https://doi.org/10.1016/S0967-0645(02)00022-X, 2002.

McDougall, T., Jackett, D., Millero, F., Pawlowicz, R., and Barker, P.: A global algorithm for estimating Absolute Salinity., *Ocean Sci.*, 8, 1123–1134, https://doi.org/10.5194/os-8-1123-2012, 2012.

McGillicuddy, D. J. Jr, Robinson, A., Siegel, D., Jannasch, H., Johnson, R, Dickey, T., et al.: Influence of mesoscale eddies on new production in the Sargasso Sea. Nature, 394(6690), 263, https://doi.org/10.1038/28367, 1998.

McGillicuddy, D. J., Anderson, L. A., Bates, N. R., Bibby, T., Buesseler, K. O., Carlson, C. A., et al.: Eddy/wind interactions stimulate extraordinary mid-ocean plankton blooms. *Science*, 316(5827), 1021–1026, https://doi.org/10.1126/science.113625, 2007.

McGillicuddy, D. J.: Mechanisms of physical-biological-biogeochemical interaction at the oceanic mesoscale. *Annual Review of Marine Science*, 8, 125–159, https://doi.org/10.1146/annurev-marine-010814-015606, 2016.

The MerMex Group. Marine ecosystems' responses to climatic and anthropogenic forcings in the Mediterranean. *Progress in Oceanography* 91, no. 2: 97-166, https://doi.org/10.1016/j.pocean.2011.02.003, 2011.

Menkes, C. E., Lengaigne, M., Lévy, M., Éthé, C., Bopp, L., Aumont, O., Vincent, E., Vialard, J., and Jullien, S.: Global impact of tropical cyclones on primary production, *Global Biogeochem. Cy.*, 30, 767–786, https://doi.org/10.1002/2015GB005214, 2016.

Millot, C.: Circulation in the western Mediterranean Sea, *J. Mar. Syst.*, 20, 423–442, https://doi.org/10.1016/S0924-7963(98)00078-5, 1999.

Millot, C. and Taupier-Letage, I.: Circulation in the Mediterranean Sea, in: The Mediterranean Sea, edited by Saliot, A., Springer, Berlin, Heidelberg, Germany, 29–66, https://doi.org/10.1007/b107143, 2005.

Millot, C., Candela, J., Fuda, J.-L., and Tber, Y.: Large warming and salinification of the Mediterranean outflow due to changes in its composition, *Deep-Sea Res. Pt. I*, 53, 656–666, https://doi.org/10.1016/j.dsr.2005.12.017, 2006.

Mouw, Colleen B., Audrey Barnett, Galen A. McKinley, Lucas Gloege, and Darren Pilcher. Phytoplankton size impact on export flux in the global ocean. *Global Biogeochemical Cycles* 30, no. 10: 1542-1562, https://doi.org/10.1002/2015GB005355, 2016.

Mustard, Alexander T., and Thomas R. Anderson. Use of spherical and spheroidal models to calculate zooplankton biovolume from particle equivalent spherical diameter as measured by an optical plankton counter. *Limnology and Oceanography: Methods* 3, no. 3: 183-189, https://doi.org/10.4319/lom.2005.3.183, 2005.

Navarro, Gabriel, Séverine Alvain, Vincent Vantrepotte, and I. Emma Huertas. Identification of dominant phytoplankton functional types in the Mediterranean Sea based on a regionalized remote sensing approach. *Remote Sensing of Environment* 152: 557-575, https://doi.org/10.1016/j.rse.2014.06.029, 2014.

Neal, Radford M. Probabilistic inference using Markov chain Monte Carlo methods. Toronto, ON, Canada: Department of Computer Science, University of Toronto, 1993.

Pascual, Ananda, Simón Ruiz, Bruno Buongiorno Nardelli, Stéphanie Guinehut, Daniele Iudicone, and Joaquín Tintoré. Net primary production in the Gulf Stream sustained by quasi‑geostrophic vertical exchanges. *Geophysical Research Letters* 42, no. 2: 441-449, https://doi.org/10.1002/2014GL062569, 2015.

Pilo, G. S., Oke, P. R., Coleman, R., Rykova, T., & Ridgway, K.: Patterns of vertical velocity induced by eddy distortion in an ocean model. *Journal of Geophysical Research: Oceans*, 123, 2274–2292, https://doi.org/10.1002/2017JC013298, 2018.

Regnard, Paul. *Recherches expérimentales sur les conditions physiques de la vie dans les eaux*. G. Masson, 1891.

Ribalet, F., Marchetti, A., Hubbard, K. A., Brown, K., Durkin, C. A., Morales, R., Robert, M., Swalwell, J. E., Tortell, P. D., and Armbrust, E. V.: Unveiling a phytoplankton hotspot at a narrow boundary between coastal and offshore waters, *P. Natl. Acad. Sci. USA*, 107, 16571–16576, https://doi.org/10.1073/pnas.1005638107, 2010.

Rousselet, L., Doglioli, A. M., de Verneil, A., Pietri, A., Della Penna, A., Berline, L., et al.: Vertical motions and their effects on a biogeochemical tracer in a cyclonic structure finely observed in the Ligurian Sea. *Journal of Geophysical Research: Oceans*, 124, 3561–3574. https://doi.org/10.1029/2018JC014392, 2019.

Schroeder, K., Gasparini, G., Borghini, M., Cerrati, G., and Delfanti, R.: Biogeochemical tracers and fluxes in the Western Mediterranean Sea, spring 2005, *J. Marine Syst.*, 80, 8–24, https://doi.org/10.1016/j.jmarsys.2009.08.002, 2010.

Shi, Jia-Rui, Lynne D. Talley, Shang-Ping Xie, Qihua Peng, and Wei Liu. Ocean warming and accelerating Southern Ocean zonal flow. *Nature Climate Change* 11, no. 12: 1090-1097, https://doi.org/10.1038/s41558-021-01212-5, 2021.

Siokou-Frangou, I., Christaki, U., Mazzocchi, M. G., Montresor, M., Ribera d'Alcalá, M., Vaqué, D., and Zingone, A.: Plankton in the open Mediterranean Sea: a review, *Biogeosciences*, 7, 1543–1586, https://doi.org/10.5194/bg-7-1543-2010, 2010.

Sosik, H. M., Olson, R. J., Neubert, M. G., Shalapyonok, A., and Solow, A. R.: Growth rates of coastal phytoplankton from time-series measurements with a submersible flow cytometer, *Limnol. Oceanogr.*, 48, 1756–1765, https://doi.org/10.4319/lo.2003.48.5.1756, 2003.

Sterner, R. W., & Hessen, D. O. (1994). Algal nutrient limitation and the nutrition of aquatic herbivores. *Annual Review of Ecology and Systematics*, 25, 1–29. https://doi.org/10.1146/annurev.es.25.110194.000245, 1994.

Thyssen, M., Tarran, G. A., Zubkov, M. V., Holland, R. J., Grégori, G., Burkill, P. H., and Denis, M.: The emergence of automated high-frequency flow cytometry: revealing temporal and spatial phytoplankton variability, *J. Plankton Res.*, 30, 333–343, https://doi.org/10.1093/plankt/fbn005, 2008.

Tzortzis, R., Doglioli, A. M., Barrillon, S., Petrenko, A. A., d'Ovidio, F., Izard, L., Thyssen, M., Pascual, A., Barceló-Llull, B., Cyr, F., Tedetti, M., Bhairy, N., Garreau, P., Dumas, F., and Gregori, G.: Impact of moderately energetic fine-scale dynamics on the phytoplankton community structure in the western Mediterranean Sea, *Biogeosciences,* 18, 6455–6477, https://doi.org/10.5194/bg-18-6455-2021, 2021.

Watson AJ, Robinson C, Robinson JE, Williams PJ le B, Fasham MJR, Spatial variability in the sink for atmospheric carbon dioxide in the North Atlantic. *Nature* 350:50–53, https://doi.org/10.1038/350050a0, 1991.

Wilson, Laura J., Christopher J. Fulton, Andrew McC Hogg, Karen E. Joyce, Ben TM Radford, and Ceridwen I. Fraser. Climate‑driven changes to ocean circulation and their inferred impacts on marine dispersal patterns. *Global ecology and biogeography* 25, no. 8: 923-939, https://doi.org/10.1111/geb.12456, 2016.

Winder, Monika, and James E. Cloern. The annual cycles of phytoplankton biomass. *Philosophical Transactions of the Royal Society B: Biological Sciences* 365, no. 1555 : 3215-3226, https://doi.org/10.1098/rstb.2010.0125, 2010.

Winder, Monika, and Ulrich Sommer. Phytoplankton response to a changing climate. *Hydrobiologia* 698, no.1: 5-16, https://doi.org/10.1007/s10750-012-1149-2, 2012.

Worden, A. Z. and Binder, B. J.: Application of dilution experiments for measuring growth and mortality rates among Prochlorococcus and Synechococcus populations in oligotrophic environments, *Aquat. Microb. Ecol.*, 30, 159–174, https://doi.org/10.3354/ame030159, 2003.

Yamaguchi, Ryohei, and Toshio Suga. Trend and variability in global upper‑ocean stratification since the 1960s. *Journal of Geophysical Research: Oceans* 124, no. 12: 8933-8948, https://doi.org/10.1029/2019JC015439, 2019.

Yao, Yulong, Chunzai Wang, and Yao Fu. Global Marine Heatwaves and Cold‑Spells in Present Climate to Future Projections. *Earth's Future* 10, no. 11 (2022): e2022EF002787, https://doi.org/10.1029/2022EF002787, 2022.

Zhang, Jinlun, and Mike Steele. Effect of vertical mixing on the Atlantic Water layer circulation in the Arctic Ocean. *Journal of geophysical research: Oceans* 112, no. C4, https://doi.org/10.1029/2006JC003732, 2007.

---

## Author Comment (AC3)

**Authors's reply to Anonymous Referee #2 comments on on egusphere-2022-1008 Tzortzis et al.**

Dear referee,

Thank you very much for your constructive comments and suggestions, as well as your corrections of the grammar and spelling in order to improve the quality of the manuscript. According to the Biogeosciences guidelines, hereinafter we address the main points you raised. We are confident that following your remarks and suggestions, as well as the ones from the other anonymous referee, we can improve our manuscript for publication in Biogeosciences. Your comments are in black, our replies in blue.

**General Comments**

Fine-scale physical processes affect the community structures and productivity of marine plankton at various time scales, but the study to explore them are relatively limited due to technical difficulties. This study aims at untangling this problem by applying the combination of a semi-Lagrangian survey, semi-continuous sampling, and biomathematical models. This approach is novel, and the results obtained from the field survey in the Mediterranean Sea seem to be reasonable. I believe that their approach may open the door to the elucidation of complex physical processes that affect marine microbial ecology, though there are still some problems to be answered.

We thank you for this encouraging comment on our scientific approach.

The first problem is that the objective of the present study (this article) is ambiguous. I understand the final goal of their study, but the results obtained this time are too primitive for that. Based on the results obtained, the authors should reconstruct the objective(s) of the "present" study. The authors should be clear about whether this manuscript concentrated on the development of a new method or aimed to elucidate the effects of the frontal structure observed in the South Mediterranean Sea on phytoplankton structures to some degree. In addition to Introduction and Abstract, the title of the article possibly should be changed in association with that.

We acknowledge that we did not sufficiently detail our scientific questions. Our present study is closely related to our previous work (Tzortzis et al., 2021) on the description of the physical characteristics of a **frontal finescale structure** and its effects on the distribution of phytoplankton abundances, in the south part of the Balearic Islands (Mediterranean Sea). The importance of the frontal area studied in the present article is due to the fact that most of the in-situ studies related to the physical-biological coupling at finescale have focused on extreme situations occurring in boundary currents, where intense fronts and dramatic contrasts in water properties are met but are not representative of the global ocean. Indeed, on the contrary, vast oceanic regions are dominated by weak fronts continuously created, moved

and dissipated, and which separate different water masses with similar properties. Very few studies exist in these regions due to the difficulty of performing in situ experiments over these short-lived and small features. In our previous work Tzortzis et al. (2021) we showed that the fine-scale front observed during our cruise in 2018 in an oligotrophic region as the SW Mediterranean maintains the driving role on phytoplankton diversity also in these moderately energetic regions. Nevertheless, this first study did not explain this particular distribution of phytoplankton, and open questions remain: **Is it exclusively driven by the dynamics of the front ? Or, do biological processes also play a role ?**
In the present study, we attempt to explain the patterns of phytoplankton abundances by focusing on the cellular dynamic of these organisms using the size-structured population model developed by Sosik et al. (2003). This is possible thanks to the analysis of phytoplanktonic cells at the single cell level.

We agree that we did not develop enough the potential effect of the frontal dynamics on the structure of the phytoplankton community, especially in the Discussion. We will rework this part, detailing more in depth the potential effect of vertical velocities and water masses properties (temperature, salinity, nutrients) on the phytoplankton communities.

Furthermore, we did not sufficiently discuss the novelty of our methodology, and its implication for future oceanographic cruises. Although some studies have already used the size-structured population model (Sosik et al., 2003 ; Ribalet et al., 2010, Dugenne et al., 2014 ; Marrec et al., 2018) or other models (Geider et al., 1997 ; MacIntyre et al., 2000) to compute phytoplankton growth rates, the novelty of our study is its application in a context of a Lagrangian sampling strategy. Moreover, we applied this model on several phytoplankton groups (not only *Synechococcus* or *Prochlorococcus*, like most studies). To our knowledge, this has not been done before (except the study of Dugenne et al. (2014) which applied it to specific diatoms). We think our methodology applied here, paves the way for future studies.

The second problem is about the robustness and significance of the estimates of growth and loss rates. When we compare two or more values, the intervals of confidence or possible standard errors are indispensable. However, in the present manuscript, there are no remarks on that. If possible, please add the statistical information.

We agree that it is important to add intervals of confidence; in the previous literature this is not done, probably due to the difficulties in calculating them. In any case, as suggested also by the other referee, we have completely reworked the section concerning the methodology of the size-structured population model (section 2.3 in the manuscript). We provide this part further in the present document, in the case that you cannot consult our answer for the other referee. We hope that this new version clarifies the principle of this model. Indeed, statistics are already included in the model: the growth rates were estimated using the maximum likelihood function and 200 iterations were run to estimate the standard deviation of group-specific growth rates using a Markov Chain Monte Carlo (Geyer, 1992 ; Neal, 1993).

English grammatical errors are relatively frequent in this manuscript. The authors should

have it checked by a native speaker or some editorial service. For example, "Numerical simulation have shown" (L3), "Since several years" (L5), and "a precious information" (L7).

Thank you, we will rework the manuscrit with the help of a new co-author English speaker.

These are general comments on this manuscript. The followings are minor specific comments.

**Specific Comments**

L71 "satellite SWOT will be launched" is correct.

Sorry for the mistake. The SWOT satellite is now in orbit ! It was launched on December 16[th] 2022. We will modify this section following also the suggestions of the other referee.

L74 What do the authors mean by "moderate energy"? Which energy? And in which way is it important in the selection of the present study site?

Here, we mean that Mediterranean frontal structures are often less intense than those found in boundary currents such as the Kuroshio, that are able to generate vertical velocities in order of 30 m day$^{-1}$ (Clayton et al., 2014). By contrast, vertical velocities in the Mediterranean sea are in the order of 8 m day$^{-1}$ (Barceló-Llull et al., 2021, Tzortzis et al., 2021).

Below is a map of the surface eddy kinetic energy by Pascual et al. (2006), where the contrast between the Mediterranean sea and the western boundary currents is evident.
We will add this reference and explanation in the new version of the Methods section.

[Figure]

Figure: Eddy kinetic energy (EKE) estimated with 4 altimetric missions (Jason-1 + T/Pinterlaced + ERS-2/ENVISAT + GFO). Units are cm$^2$ s$^{-2}$ . Figure extracted from Pascual et al. (2006).

L91 "have been measured" should be "were measured".

Thank you, we will modify it when we rework the manuscrit.

L109 Was the categorization of phytoplankton populations (functional groups) on cytograms made manually on a somewhat arbitrary criterion or semi-automatedly using something like machine learning? How do the authors guarantee the validity and consistency of the categorization?

We have identified several groups of phytoplankton by flow cytometry. This categorization of phytoplankton groups on 2 dimensional plots (cytograms) was made **manually using the conventional criterion determined by flow cytometrists**. Phytoplankton groups were resolved on the basis of their light scatter (namely forward scatter FWS and sideward scatter SWS) and fluorescence (red FLR and orange FLO fluorescence ranges) properties. For instance, *Synechococcus* was unambiguously put in evidence thanks to its higher FLO intensity induced by the presence of phycoerythrin pigments. The optical characteristics of each phytoplankton cluster (or group) are provided in the literature (Dubelaar and Jonker, 2000 ; Reynolds, 2006 ; Thyssen et al., 2008 ; Edwards et al., 2015). We based our categorization on these previous studies. We identified typical phytoplankton groups of the Mediterranean Sea already found by previous works using flow cytometry (Thyssen et al., 2014 ; Marrec et al., 2018). Most of the publications using flow cytometry data to study planktonic cells perform the same way, and rely entirely on the literature and the expertise of the flow cytometrists.

Application of Artificial Intelligence as machine learning to cytometry data is currently under development in our laboratory and the recent work of Fuchs et al. (2022) provided promising results. Unfortunately, this approach is not yet mature enough, which is why we do not use it here.

L114 Show us the time and space (cruise length) ranges that a single sample covers.

We will provide a detailed information modifying the sentence as follow:
"The flow cytometer was connected to sea surface continuous flow, through the system of the thermosalinograph (TSG), (depth: 3 m). The flow cytometer sampled the seawater in a dedicated small container called "subsampler", that isolates the seawater during its analysis. Between two consecutive samples the subsampler was flushed continuously by the seawater circuit of the ship in order to clean and renew the seawater. The subsampler isolated the seawater every 30 min, and two distinct protocols (FLR6 and FLR25) were run sequentially: for FLR6 about 1.3 mL were analyzed in 420 s and for FLR25 about 4 mL were analyzed in 600 s. The use of the subsampler to isolate the volume of seawater subsampled by the flow cytometer allowed us to ignore the movement of the ship, while the flow cytometer performed its analysis. This way the volume analyzed corresponds to a point location rather than a volume spread on the ~2 km covered by the ship in 30 min".

L183 What do the authors mean by "put in evidence"?

We will change it by "identify": "Four eukaryotic picophytoplankton groups **were identified**."

L197 "A similar distribution is observed" should be "A similar distribution was observed".
Most of the sentences in this paragraph should be rewritten to past tense.

Thank you, we will carefully check verb tenses in the reworked manuscript.

L204 "In addition to the cell abundances measured along the route of the ship, the phytoplankton diurnal cycle in the two water masses was also reconstructed" This sentence means that the cell abundances were reconstructed first. But, of course, they were not "reconstructed". Rewrite.

We will rework this sentence as follows: "The phytoplankton diurnal cycle in the two water masses was reconstructed [...]."

L205 "each water mass" should be "either water mass"?

Thank you, we will modify it when we rework the manuscrit.

L207 "This adaptive Lagrangian approach allows sampling of the different functional groups of phytoplankton in each water mass" Different functional groups of phytoplankton in

different water masses can be sampled using another approach. I think that this is not the benefit of the adaptive Lagrangian approach. Explain it more appropriately.

We are sorry for the lack of clarity. Following the suggestions of the other referee, in the reworked manuscript we will move this part from the Discussion to the Methods section.
We also will modify the sentences as follow:

"The PROTEVSMED-SWOT cruise followed an adaptive Lagrangian strategy to measure at **high spatial and temporal resolution** several physical and biological variables with both in situ sensors and analysis of the sea surface water intake. The vessel route was designed ad-hoc on the basis of daily remote sensing dataset provided by the Software Package for an Adaptive Satellite-based Sampling for Oceanographic cruises (SPASSO, https://spasso.mio.osupytheas.fr, last access: February 6, 2023). SPASSO used altimetry-derived currents **from AVISO** ("Archiving, Validation and Interpretation of Satellite Oceanographic", https://www.aviso.altimetry.fr, last access: January 18, 2023 ) and ocean color observations. Chlorophyll a concentrations ([chl$a$], level 3, 1 km resolution) were provided by **CMEMS**, "Copernicus Marine Environment Monitoring Service", https://marine.copernicus.eu, last access: January 18, 2023. In addition, ocean color composite maps were provided by **CLS with the support of the CNES**). They were constructed using a simple weighted average over the previous 5 d of data gathered by the Suomi/NPP/VIIRS sensor. SPASSO generated maps of dynamical and biogeochemical structures in both near real time (NRT) and delayed time (DT).
Maps of [chl$a$] allowed us to identify two water masses, characterized by distinct [chl$a$] values and separated by a zonal front at around 38° 30' N. This front was also detected using the in situ horizontal velocities, temperature and salinity, as described in our previous study. These two water masses were sampled along a designated route of the ship, represented in black in Fig. 1. Special attention was paid to adapting the temporal sampling in order to measure the phytoplankton diel cycle in each water mass. This was achieved by continuously sampling across both water masses along transects. While the ship did not remain in each water mass for 24h, day-to-day variability remained low and measurements from several days were combined into one diel cycle (Fig. 1). The shape depicted by the ship's track led us to call these area north–south (NS) hippodrome (bold black line in Fig. 1) performed between 11 May at 02:00 and 13 May at 08:30 UTC."

L217 "Furthermore, the comparison between the biovolume observed in situ and the biovolume predicted by the model is sound and confirms that the model-predicted cell size distributions well recapitulated the diurnal cycle reflecting either growth or cell division."
Could the authors show any data or figure to support this?

We apologize for the lack of clarity. This sentence refers to Figure 4 in the manuscript. In order to also take into account the suggestion of the other referee, we will modify the text as follows:
"For *Synechococcus*, in the older AW the observed size distribution (i.e., observed biovolume) is similar to the prediction of the model (i.e., predicted biovolume). Both display

a day-long large size-class distribution centered approximately on 0.3 μm$^3$. In the younger AW (Fig. 4a, 4c) the distributions of biovolume observed and predicted are narrower than in the older AW and centered approximately on 0.2 μm$^3$ (Fig. 4b, 4d)."

L223 As mentioned in my General Comments, I request the authors to show the interval of confidence or something that can evaluate the robustness of the estimates presented by the present method. This will enable us to compare the values of different phytoplankton groups and water masses on a statistical basis. I can find something like that in Table 2, but I fail to see what it means. When the authors consider the interval, is it significant to discuss the "difference" between the two water masses?

In the manuscript, the standard deviation of the growth rates is indicated in Table 2, but in the reworked version we will also include it in the text. As previously mentioned, we will completely rework the methodological section concerning the size-structured population model, following the comments of the other referee.

**The size-structured population model**

We used the size-structured population model described by Sosik et al. (2003) and adapted by Dugenne et al. (2014) and Marrec et al. (2018), to estimate phytoplankton in situ growth rates.

The model uses as input the phytoplankton cell volume (biovolume) derived from cell light scatter intensities (FWS). Biovolumes were estimated using coefficients previously obtained by measuring a set of silica beads with the flow cytometer following the same settings used for phytoplankton analysis. The coefficients $\beta_0$ and $\beta_1$ used to convert FWS (arbitrary units, a.u.) to biovolume $v$ (μm$^3$) were derived from a log-log regression between FSW and silica bead volumes. These methods come from the studies of Koch et al. (1996) and Foladori et al. (2008).

$v = exp(\beta_0) \times FWS^{\beta_1}$    with in our case $\beta_1 = 0.9228$ and $\beta_0 = -5.8702$

In the size-structured population model, cells are classified into several size classes according to their dimensions at time $t$. $\Delta v$ is chosen in order to have enough number of classes $m$ to cover the entire observed biovolume $v$, from $v_1$ to $v_m$ (cf Figure A). Classes are logarithmically spaced as follows:

for i in 1,2,…,m      $v_i = v_1 \, 2^{(i-1)\Delta v}$      with $\Delta v$ constant

For *Synechococcus*, $\Delta v = \frac{1}{6}$ and $m = 40$, so that the model size classes encompassed our full measured size distributions.

[Figure]

Figure A: Cell cycle stages in the size-structured population model. Cells may grow to the next size class ($\gamma$) or be at equilibrium (*1- $\gamma$(t)) (1-$\delta$(v,t)*). Above a particular size, cells are large enough to divide in two daughter cells with probability (*$\delta$*). Figure adapted from Sosik et al. (2003).

At any time *t*, the number of cells in size classes $\vec{N}$ (and $\vec{w}$ its corresponding normalized distribution), was projected to *t + dt* via matrix multiplication:

$$\vec{N}(t + dt) = A(t)\ \vec{N}(t) \qquad \text{and} \qquad \vec{w}(t + dt) = \frac{A(t)\ \vec{N}(t)}{\sum A(t)\ \vec{N}(t)}$$

We chose *dt* = 10 min (i.e., $\frac{10}{60}$ h) as Sosik et al. (2003) and Dugenne et al. (2014), because for this time step, cells are unlikely to grow more than one size class.

*A(t)* is a tridiagonal transition matrix that contains (cf Figure B):
1) $\gamma$: the probability of cellular growth
2) $\delta$: the probability of cells entering mitosis
3) the cells stasis, i.e., the probability for cells to maintain their state (i.e size) in equilibrium during the temporal projection.

$$
\begin{array}{c}
\begin{array}{ccccc}
v_1(t) & \dots & v_j(t) & \dots & v_m(t)
\end{array}\\
\begin{array}{c}
v_1(t+dt)\\ \vdots \\ v_j(t+dt)\\ \\ \vdots \\ v_m(t+dt)
\end{array}
\left(
\begin{array}{ccccc}
1-\gamma(t) & \dots & 2.\delta(v_j,t) & \dots & 0\\
0 & \ddots & \begin{array}{c}(1-\gamma(t)).\\(1-\delta(v_j,t))\end{array} & \ddots & 0\\
0 & \ddots & 0 & \ddots & 1-\delta(v_m,t)
\end{array}
\right)
\end{array}
$$

Figure B: Matrix transition A(t). (Figure extracted from Dugenne, 2017 (thesis)).

The temporal projection

$N_{|v=v1} (t +dt) = (1 - \gamma(t)).N_{v=v1}(t) + 2\delta(v_j, t).N_{v=vj}(t)$

$N_{|v=vm} (t +dt) = (1 - \delta(t)).N_{v=vm}(t) + \gamma(t).N_{v=vm-1}(t)$

Probability of cellular growth

The probability of cells growing ($\gamma$) to the next size class depends only on the light intensity (irradiance) necessary for photosynthesis, expressed as:

$\gamma(t) = \gamma_{max} [ 1 - exp(- \frac{E(t)}{E^*})]$

$\gamma_{max}$: maximum proportion of cells growing (dimensionless quantity)
E: irradiance ($\mu E\ m^{-2}\ s^{-1}$)
E*: irradiance normalizing constant ($\mu E\ m^{-2}\ s^{-1}$)

Probability of cells entering mitosis

According to Dugenne et al. (2014), $\delta$ expresses a proportion (between 0 and 1) modeled by the combination of two Normal distributions ($\mathcal{N}$). One is linked to the cell size, the other is linked to the time of cell division. Both imply an optimum, reached at $\bar{v}$ and $\bar{t}$ respectively, for cell division above which the cell size and the timing of division is suboptimal.

$\delta(t, v) = \delta_{max}\ \mathcal{N}(\bar{v},\sigma_v^2)\ \mathcal{N}(\bar{t},\sigma_t^2)$

$\delta_{max}$: maximum proportion of cells entering mitosis (dimensionless quantity)
$\bar{v}$: mean of the size Normal distribution ($\mu m^3$)
$\sigma_v$: standard deviation of the size Normal distribution ($\mu m^3$)
$\bar{t}$: mean of the time Normal distribution (h)
$\sigma_t$: standard deviation of the time Normal distribution (h)

Cells stasis

A third functional proportion is included in the transition matrix $A(t)$, to represent cell stasis. Since this function illustrates a non-transition, it is modeled by the proportion of cells that neither divided nor grew between $t$ and $t + dt$.

*[1 -γ(t)] [1 - δ(t, v)]*

Optimal parameters

The set of parameters, $\theta$ is estimated by maximum likelihood function, assuming errors between observed $\vec{w}$ and predicted $\widehat{w}$ normalized size distributions. Their standard deviations are estimated by a Markov Chain Monte Carlo approach (Geyer, 1992 ; Neal, 1993) that sample $\theta$ from their prior density distribution, obtained after running 200 optimizations on bootstrapped residuals to approximate the parameter posterior distribution using the normal likelihood. (The likelihood function represents the probability of random variable realizations conditional on particular values of the statistical parameters).

$\theta = [\gamma_{max}, E^*, \delta_{max}, \bar{v}, \sigma_v, \bar{t}, \sigma_t] = argmin(\sum(\theta))$

$$\sum(\theta) = \sum_{t}^{t+dt} \sum_{i=1}^{m} (w(t) - \widehat{w}(t,\theta))^2$$

$\widehat{w}(t,\theta) = A(t-dt, \theta)\, \vec{w}(t-dt)$ and $\widehat{N}(t,\theta) = A(t-dt, \theta)\, N(t-dt)$

Table: Model parameters being optimized.

| Parameters | Definition | Interval | Units |
|---|---|---|---|
| $\gamma_{max}$ | Maximum proportions of cells in growing phase | $[0, 1]$ | $\varnothing$ |
| $E^*$ | Irradiance normalizing constant | $[0,\infty[$ | $\mu E\ m^{-2}\ s^{-1}$ |
| $\delta_{max}$ | Maximum proportion of cells entering mitosis | $[0,1]$ | $\varnothing$ |
| $\bar{v}$ | Mean of the size Normal distribution | $[v_{min}, v_{max}]$ | $\mu m^3$ |

| $\sigma_v$ | Standard deviation of the size Normal distribution | $[10^{-06}, \infty[$ | $\mu m^3$ |
|---|---|---|---|
| $\bar{t}$ | Mean of the time Normal distribution | $[1, 24\frac{1}{dt} + 1]$ | hours |
| $\sigma_t$ | Standard deviation of the time Normal distribution | $[10^{-06}, \infty[$ | hours |

Growth rate and loss rate

Once optimal parameters are identified, the growth rate is calculated using the time projection of the initial size distribution $N$. 200 iterations were run to estimate the standard deviation of group-specific growth rates.

$$\mu_{size} = \frac{1}{t+dt} ln\left(\frac{\sum\limits_{i=1}^{m} \widehat{N}_i(t+dt)}{\sum\limits_{i=1}^{m} N\vec{}_i(t)}\right)$$

$N$: observed size distribution at $t = 0$ (cells cm$^{-3}$)

$i$: i$^{th}$ size class

$\widehat{N}$: predicted size distribution (cells cm$^{-3}$)

$m$: number of size classes

$dt$: time step (h)

$t + dt$: temporal integration of the distribution projection (h), $t + dt = 24$ h

$\mu_{size}$: growth rates (day$^{-1}$)

The model estimates a population intrinsic growth rate $\mu_{size}$, and a specific loss rate, over a 24 h period.

The daily average population loss rate $\bar{l}$ is obtained by the difference between the intrinsic growth rates $\mu_{size}$ and the hourly logarithmic of observed size distribution $N$.

$$\bar{l} = \int\limits^{dt} \mu_{size}(dt) - \frac{1}{dt} ln\left(\frac{N\vec{}(t+dt)}{N\vec{}(t)}\right)$$

L223 What do the authors mean by a negative loss rate? I think that it should be shown as a positive value if the loss term is significant.

The model estimates a population intrinsic growth rate, $\mu_{size}$, and a specific loss rate, $l(t)$, over a 24 h period. The loss term includes both biological losses (grazing or death, always negative) but also physical losses (e.g., advection, which can be positive or negative, see Sosik et al., 2003).

L224 "a low division rate" should be "a low loss rate"?

Indeed it is "a low loss rate".

L225 We are not able to judge whether the difference is "significant", without an appropriate statistical figure. Did the authors conduct a statistical test? In which way? What was the level of significance?

We did not conduct a statistical test, but in the reworked manuscript we plan to measure the "fit" of the model, i.e. compare observed vs predicted cell distributions (e.g. Fig. 4a vs 4c). The idea is to recover the error used to define the "best fit" and the best parameters (i.e. error of the selected model).

L244 "largest cells of *Synechococcus* are dominant" This sounds unnatural. "large cells" or "larger cells" may sound more natural.

Thank you for your English corrections.

L245 "This is due to the fact that the older AW is composed of *Synechococcus* cells transiting in all the cell cycle stages all day long". That the older AW is composed of *Synechococcus* cells transiting in all the cell cycle stages all day long is not a "fact", but a suggestion or speculation derived from the present observation. The authors should be more careful about it.

The sentence will be reformulated as follows: "The model results suggest that in the older AW *Synechococcus* cells transit in all the cell cycle stages all day long."

L250 "The patchiness of a distribution" laterally means how frequently "patches" are observed in that distribution. It does not mean how dispersed it is over a wide range. This misunderstanding may be critical in this discussion.

We apologize for the lack of clarity. We will remove this sentence, as we don't think this information is relevant because it is kind of repetitive with what we wrote before, and modify this part also taking into account the comments of the other referee:
"The model results suggest that in the older AW *Synechococcus* cells transit in all the cell cycle stages all day long. Furthermore, in the older AW the cells display a day-long large

size-class distribution centered approximately on 0.3 $\mu m^3$ while in the younger AW (Fig. 4a, 4c) the distribution is narrower and centered approximately on 0.2 $\mu m^3$ (Fig. 4b, 4d)."

L257 Avoid using any contraction (including "couldn't") in academic writing.

Thank you for your English correction.

L258 What is an "important biodiversity"? I believe that biodiversity is always important.

We apologize for the misuse of the adjective "important". Of course, we do agree on the importance of biodiversity! We will modify the sentence as follows:
"Picophytoplankton is often characterized by the presence of several taxa with potentially different effects on the population dynamics, whereas nanophytoplankton is mostly dominated by diatoms in the Mediterranean Sea (Siokou-Frangou et al., 2010 ; Marty et al., 2002 ; Navarro et al., 2014 ; El Hourany et al., 2019). "

L261 Does this mean that the authors should have conducted molecular analysis (e.g. metabarcoding) to elucidate which taxonomic group each flow cytometric population is composed of? Although it requires flow sorting before analysis, is it a possible future plan? Anyway, the authors mention "this hypothesis" here, but I could not find any hypothesis to be tested from this paragraph. Please reconsider the issues to be discussed here and rearrange this paragraph.

As mentioned in the methodology section, several phytoplankton groups were identified by flow cytometry. This analysis allowed us to detect various groups of eukaryotic nanophytoplankton (RNano and SNano) and eukaryotic picophytoplankton (Pico1, Pico2, Pico3, PicoHFLR). *Synechococcus* is a prokaryotic picophytoplankton, but we have made a distinction between the picophytoplankton group and *Synechococcus* group because this latter was unambiguously resolved by flow cytometry thanks to its higher FLO intensity induced by the presence of phycoerythrin pigments. Idem for Cryptophytes which also have a peculiar and unambiguous optical signature.

Unfortunately, we did not conduct molecular analysis, that is why we are not able to identify the taxa contained in pico- and nanophytoplankton groups. In our future cruise (spring 2023), we plan to use metabarcoding and metagenomic analysis to address the biodiversity of phytoplankton. We will also perform zooplankton and virus sampling to understand the effect of zooplankton grazing and viral lysis on the different phytoplankton groups.

Following the suggestions of the other referee, we will move the sentences (L 256 - 258) in section 3.3. In the reworked manuscript, we will focus on the explanation of why the size distribution of picophytoplankton is noisy whereas we obtained a clear pattern for nanophytoplankton.

L269 The authors have used the term "finescale" and the rough definition appears here for the first time. From which have the authors derived this definition? We often used the term "mesoscale" to show this spatial scale in marine processes (Dickey and Bidigare, 2005, Scientia Marina). If this term was originally defined, the authors should have shown that in Introduction.

In the manuscript, we defined the term "finescale" in the Introduction (L 20-21): "ocean structures characterized by horizontal scale of the order of 1-100 km, with a short lifetime (days-weeks)". Following your comments, in the reworked manuscript we will develop this definition further.
Although several studies used the term "mesoscale", in our case "finescale" seems more appropriate. Indeed, by using this term, we include a fraction of the mesoscale processes (e.g. eddies), with scales close to the first internal Rossby radius, and the submesoscale processes, with scales smaller than the first internal Rossby radius (e.g. fronts) (Capet et al., 2008a ; Capet et al., 2008b ; McWilliams, 2016 ; Lévy et al., 2018).

L272 What are "many important oceanic processes including biogeochemical cycles and biodiversity"? Unless specified, we cannot judge whether "this suggests the possibility of a close coupling between the finescale forcing and the phytoplankton distribution and growth." Honestly, I could not understand what the authors are to discuss in this paragraph. In different water masses, phytoplankton community structures are different almost every time. We usually attribute this to different water properties that can affect phytoplankton physiology, including salinity, temperature, turbidity, and nutrient concentrations, rather than to temporal and/or spatial scales of physical processes. I am afraid that there may be a large discrepancy between the final goal of this (overall) study and possible conclusions extracted from the present results.

We agree that phytoplankton is affected by water masses properties. In this part, we were not clear enough, we propose to reformulate these sentences as follows:
"The temporal scale of finescale processes (days-weeks) is of the same magnitude as biogeochemical processes and phytoplankton cellular cycle. The rapid evolution of these finescale structures influence the phytoplankton community, suggesting the possibility of a close coupling between finescale forcing and phytoplankton distribution and growth."

L284 How much of the two figures (Figs. 7 and A2) was extracted from the original version in Tzortzis et al. (2021)? If it is a copy of the original, the authors should not use it again but should just cite it. And the authors say "in the frontal area upwellings and downwellings occur with different intensities", but I think that it is not reflected in Fig. 7. From this figure, I could not find any difference in the vertical velocity of the two water masses.

Figures 2 and A2 were indeed extracted from Tzortzis et al. (2021). Following your suggestion and those of the first referee, we will move these figures into supplementary information. Concerning Figure 7, we have adapted this figure from Tzortzis et al. (2021),

which is why we think that it should stay in the manuscript. Following the suggestions of the other referee, we have reworked this figure (see below) for clarity.

[Figure]

Figure 7: The contrasted distribution of phytoplankton in the frontal area. The circles represent the abundances of the several phytoplankton groups in the two water masses separated by the front. The boxes indicate the biovolume ($v$) and the growth rates ($\mu$) for each phytoplankton group. Figure adapted from Tzortzis et al. (2021).

L286 The authors intended to say "spatial", not "special"? Even if so, the authors did not show "spatial" distribution in this paper. They just showed "temporal variations" in phytoplankton populations while covering two water masses.

We intended to say "special", but "particular" is more appropriate.

L289 "high phytoplankton size" is not an appropriate term.

Thank you for your corrections, maybe "the largest cells" is better.

L290 "picophytoplankton are more abundant in oligotrophic regions". This is a problematic description. First, it is true that the proportion of picophytoplankton in the total phytoplankton biomass becomes higher in the oligotrophic region compared with that in the mesotrophic or eutrophic regions. However, the absolute biomass or abundance of picophytoplankton is not always higher in the oligotrophic area. Generally speaking,

*Prochlorococcus*, which are adapted to ultraoligotrophic environments, are most abundant in oligotrophic waters. However, *Synechococcus* and eukaryotic picophytoplankton are more abundant in the mesotrophic region. Additionally, within the narrow trophic variation of the oligotrophic regions (typically < 0.1 μM of nitrate), a higher concentration of nutrients is sometimes related to the higher abundance of these picophytoplankton populations. Because the Mediterranean Sea is widely depleted with surface nutrients, discussion is not such a simple one as "picophytoplankton are more abundant in oligotrophic regions." I admit that this description is true for the study area, as shown in previous studies (Jacquet et al., 2010; Mena et al., 2016) as well, but it is not always related to generalization.

Thank you for your analysis. We will rework this part taking into account your comments, as follows:

"In our study, the older AW is characterized by larger cells of *Synechococcus* and nanophytoplankton with low abundances, whereas the younger AW is dominated by small cells with high abundances. Furthermore, microphytoplankton (i.e largest type of phytoplankton) is more abundant in older AW than in the younger AW. A possible explanation is that these two water masses do not have the same nutrient concentration, thus favoring certain phytoplankton groups.
Bethoux (1989) and Schroeder et al. (2010) have observed that the older AW is slightly more enriched with nutrients than the younger AW because during its circulation across the Mediterranean basin, the older AW receives nutrient inputs from the continent (river discharges, rain, wind). While in our study we do not have nutrient data, we can suppose that the nutrient distribution across the two water masses should be similar to the one measured during the previous studies of Bethoux (1989) and Schroeder et al. (2010).
We propose that the enhancement in nutrient in the older AW explains the corresponding phytoplankton cell size and abundances distributions. Our hypothesis is supported by the fact that our results are in agreements with those of Jacquet et al. (2010) and Mena et al. (2016) which also found the highest abundances of the small phytoplankton (*Synechococcus* and picophytoplankton) in the most oligotrophic waters, i.e., the younger AW. Furthermore, previous studies have shown that the proportion of picophytoplankton in the total phytoplankton biomass is higher in the oligotrophic region compared with that in the mesotrophic or eutrophic regions (Zhang et al., 2008 ; Cerino et al., 2012). Indeed, their better surface:size ratio due to their small size confers them a better capacity to inhabit areas with very low nutrient concentration compared to larger phytoplankton (Kiørboe, 1993 ; Marañón, 2015). Since our study area is always oligotrophic (Moutin et al., 2012), a small variation of the nutrient concentration (typically < 0.1 μM of nitrate) is sufficient to generate higher abundance of picophytoplankton."

L295 "If in our study we do not have nutrient data" I do not understand the intention. Are the authors unclear whether they have nutrient data themselves?

We will modify the sentence as follows:

"Unfortunately, it was not possible for both technical and funding reasons to perform nutrient measurements during the 2018 cruise, which is why we cannot provide nutrient concentrations of both water masses. We acknowledge the importance of this information and these measurements are planned for our future cruise this year."

L305 Here the authors abandoned the trial to estimate the effects of physical processes on irradiance received by phytoplankton, but is it impossible to compare them from the results of vertical velocity in the two water masses?

Following also the suggestion of the other referee, we will modify the beginning of this section as follows:
"Previous studies have well established that vertical motions impact biogeochemistry (Mahadevan & Tandon, 2006 ; Mahadevan, 2016 ; McGillicuddy, 2016). Upward vertical velocities drive deep nutrients into the euphotic layer and also move the phytoplankton cells along the water column resulting in changing light conditions."

L309 Although the authors succeeded in estimating intrinsic growth rates of various phytoplankton populations in the two different water masses using novel methodologies, the conclusion remarks here seem too superficial and primitive. The authors did not discuss the validity or robustness of the methodology or did not discuss the interactive connections among physical fields, chemical environments, and phytoplankton growth with quantitative comparisons.

We will completely rework the Conclusion taking into account your comments and those of the other reviewer:

"Phytoplankton structure and dynamics are a complex result of many interacting biological and physical phenomena. Finescale structures, and in particular fronts, generate vertical velocities which displace phytoplankton cells and nutrients in the water column, thus influencing phytoplankton communities. These mechanisms are only partially understood because the spatial scale of these structures and their ephemeral nature make them particularly difficult to study in situ; as a consequence only a few studies have been performed in finescale frontal regions. The estimates of specific growth rates for the various phytoplankton groups is one of the keys to better understand how environmental conditions affect phytoplankton dynamics. In this study, we followed the dynamics of several phytoplankton groups in two distinct water masses both in terms of hydrology and phytoplankton abundances, in order to explain their particular distribution.
  The originality of our work resides in the fact that we used a size-structured population model applied in two water masses identified using a Lagrangian sampling strategy. To our knowledge that has never been done before. This strategy allowed us to reconstruct the diurnal cycle for several phytoplankton groups and to identify contrasted dynamics in the two water masses. For *Synechococcus* and nanophytoplankton, we found the higher cell size in the older AW located to the north of the front, associated with the lower abundances. A possible explanation is that the older AW is more enriched in nutrients than the younger AW,

thus favoring the largest cells. This remains a hypothesis because of a lack of nutrient data. Furthermore, in the oligotrophic region such as the Mediterranean Sea, a narrow trophic variation of the nutrient concentration is sufficient to generate higher abundance of phytoplankton. Another novelty of our study is that we applied this model on several phytoplankton groups identified by flow cytometry, whereas previous studies only applied it to *Synechococcus* and *Prochlorococcus*. We obtained good results for *Synechococcus* and nanophytoplankton. However, our results were noisy for picophytoplankton groups probably because these latter contain several taxa with differing dynamics.

Our work paves the way for many research perspectives. Direct integration of growth rates in biogeochemical models (Cullen et al., 1993) should be taken into account for a better assessment of the biogeochemical contribution of phytoplankton in oligotrophic ecosystems and to better forecast its evolution in the context of global change. Furthermore, we plan future experiments again in the South Western Mediterranean in spring 2023, after the launch of the SWOT satellite which will provide high resolution altimetry-derived current. Involving high-resolution nutrient measurements (and also high-precision ones, considering the oligotrophy of the Mediterranean Sea), coupled with metabarcoding (to address the biodiversity of phytoplankton), zooplankton and virus sampling, we will improve the understanding of zooplankton grazing and viral lysis on the different phytoplankton groups. Furthermore, we aim to explore how biogeochemical and ecological role of the finescales in regions of weak circulation are different from the ones more documented in highly energetic regions like boundary currents. In the Mediterranean sea, the low nutrient content is indeed the perfect condition when addressing this question, because even weak horizontal or vertical nutrient redistributions associated with the finescale circulation are likely to result in a biological response (Talmy et al., 2014 ; Hashihama et al., 2021)."

**References**

Barceló-Llull, Bàrbara, Ananda Pascual, Antonio Sánchez-Román, Eugenio Cutolo, Francesco d'Ovidio, Gina Fifani, Enrico Ser-Giacomi et al.: Uncovering fine-scale ocean currents from in situ observations to anticipate SWOT satellite mission capabilities, *Frontiers in Marine Science*, https://doi.org/10.3389/fmars.2021.679844, 2021.

Capet, X., McWilliams, J. C., Molemaker, M. J., and Shchepetkin, A.: Mesoscale to submesoscale transition in the California Current System. Part I: Flow structure, eddy flux, and observational tests, *J. Phys. Oceanogr.*, 38, 29–43, https://doi.org/10.1175/2007JPO3671.1, 2008.

Capet, X., McWilliams, J. C., Molemaker, M. J., and Shchepetkin, A.: Mesoscale to submesoscale transition in the California Current System. Part II: Frontal processes. *J. Phys. Oceanogr.,* 38, 44-64, https://doi.org/10.1175/2007JPO3672.1, 2008.

Cerino, F., Aubry, F. B., Coppola, J., La Ferla, R., Maimone, G., Socal, G., and Totti, C.: Spatial and temporal variability of pico-, nano-and microphytoplankton in the offshore waters of the southern Adriatic Sea (Mediterranean Sea), *Cont. Shelf Res.*, 44, 94–105, https://doi.org/10.1016/j.csr.2011.06.006, 2012.

Cullen, J. J., Geider, R., Ishizaka, J., Kiefer, D., Marra, J., Sakshaug, E., and Raven, J.: Towards a general description of phytoplankton growth for biogeochemical models, in: Towards a model of ocean biogeochemical processes, pp. 153–176, Springer, Berlin, Heidelberg, https://doi.org/10.1007/978-3-642-84602-1_7, 1993.

Dubelaar, G. B. and Jonker, R. R.: Flow cytometry as a tool for the study of phytoplankton, *Sci. Mar.*, 64, 135–156, https://doi.org/10.3989/scimar.2000.64n2135, 2000.

Dugenne, M., Thyssen, M., Nerini, D., Mante, C., Poggiale, J.-C., Garcia, N., Garcia, F., and Grégori, G. J.: Consequence of a sudden wind event on the dynamics of a coastal phytoplankton community: an insight into specific population growth rates using a single cell high frequency approach, *Front. Microbiol.*, 5, 485, https://doi.org/10.3389/fmicb.2014.00485, 2014.

Edwards, K. F., Thomas, M. K., Klausmeier, C. A., and Litchman, E.: Light and growth in marine phytoplankton: allometric, taxonomic, and environmental variation, *Limnol. Oceanogr.*, 60, 540–552, https://doi.org/10.1002/lno.10033, 2015.

Fuchs, Robin, Melilotus Thyssen, Véronique Creach, Mathilde Dugenne, Lloyd Izard, Marie Latimier, Arnaud Louchart et al. Automatic recognition of flow cytometric phytoplankton functional groups using convolutional neural networks. *Limnology and Oceanography: Methods* 20, no. 7: 387-399, https://doi.org/10.1002/lom3.10493, 2022.

Geyer, Charles J. Practical markov chain monte carlo. *Statistical science* (1992): 473-483.

Hashihama, F., Saito, H., Kodama, T., Yasui-Tamura, S., Kanda, J., Tanita, I., Ogawa, H., Woodward, E. M. S., Boyd, P. W., and Furuya, K.: Cross-basin differences in the nutrient assimilation characteristics of induced phytoplankton blooms in the subtropical Pacific waters, *Biogeosciences*, 18, 897–915, https://doi.org/10.5194/bg-18-897-2021, 2021.

Kiørboe, T.: Turbulence, phytoplankton cell size, and the structure of pelagic food webs, in: Adv. Mar. Biol., vol. 29, pp. 1–72, *Elsevier,* https://doi.org/10.1016/S0065-2881(08)60129-7, 1993.

Lévy, Marina, Peter JS Franks, and K. Shafer Smith. The role of submesoscale currents in structuring marine ecosystems. *Nature communications* 9, no. 1, https://doi.org/10.1038/s41467-018-07059-3, 2018.

Mahadevan, Amala, and Amit Tandon. An analysis of mechanisms for submesoscale vertical motion at ocean fronts. *Ocean Modelling* 14, no. 3-4: 241-256, https://doi.org/10.1016/j.ocemod.2006.05.006, 2006.

Mahadevan, A.: The impact of submesoscale physics on primary productivity of plankton, Annu. Rev. Mar. Sci., 8, 161–184, https://doi.org/10.1146/annurev-marine-010814-015912, 2016.

Marañón, E.: Cell size as a key determinant of phytoplankton metabolism and community structure, *Annu. Rev. Mar. Sci.*, 7, 241–264, https://doi.org/10.1146/annurev-marine-010814-015955, 2015.

Marrec, P., Grégori, G., Doglioli, A. M., Dugenne, M., Della Penna, A., Bhairy, N., Cariou, T., Hélias Nunige, S., Lahbib, S., Rougier, G., Wagener, T., and Thyssen, M.: Coupling physics and biogeochemistry thanks to high-resolution observations of the phytoplankton community structure in the northwestern Mediterranean Sea, *Biogeosciences*, 15, 1579–1606, https://doi.org/10.5194/bg-15-1579-2018, 2018.

McGillicuddy, D. J.: Mechanisms of physical-biological-biogeochemical interaction at the oceanic mesoscale. Annual Review of Marine Science, 8, 125–159, https://doi.org/10.1146/annurev-marine-010814-015606, 2016.

McWilliams, J. C.: Submesoscale currents in the ocean, Philos. T. Roy. Soc. A, 472, 20160117, https://doi.org/10.1098/rspa.2016.0117, 2016.

Moutin, Thierry, France Van Wambeke, and Louis Prieur. Introduction to the Biogeochemistry from the Oligotrophic to the Ultraoligotrophic Mediterranean (BOUM) experiment. *Biogeosciences* 9, no. 10 : 3817-3825, https://doi.org/10.5194/bg-9-3817-2012, 2012.

Mustard, Alexander T., and Thomas R. Anderson. Use of spherical and spheroidal models to calculate zooplankton biovolume from particle equivalent spherical diameter as measured by an optical plankton counter. *Limnology and Oceanography: Methods* 3, no. 3: 183-189, https://doi.org/10.4319/lom.2005.3.183, 2005.

Neal, Radford M. Probabilistic inference using Markov chain Monte Carlo methods. Toronto, ON, Canada: Department of Computer Science, University of Toronto, 1993.

Pascual, Ananda, Yannice Faugère, Gilles Larnicol, and Pierre‑Yves Le Traon. Improved description of the ocean mesoscale variability by combining four satellite altimeters. *Geophysical Research Letters* 33, no. 2, https://doi.org/10.1029/2005GL024633, 2006.

Reynolds, C. S.: The ecology of phytoplankton, Cambridge University Press, 2006.

Ribalet, F., Marchetti, A., Hubbard, K. A., Brown, K., Durkin, C. A., Morales, R., Robert, M., Swalwell, J. E., Tortell, P. D., and Armbrust, E. V.: Unveiling a phytoplankton hotspot at a narrow boundary between coastal and offshore waters, *Proc. Nat. Acad. Sci.* USA, 107, 16 571–16 576, https://doi.org/10.1073/pnas.1005638107, 2010.

Talmy, D., Blackford, J., Hardman-Mountford, N., Polimene, L., Follows, M., and Geider, R.: Flexible C: N ratio enhances metabolism of large phytoplankton when resource supply is intermittent, *Biogeosciences*, 11, 4881–4895, https://doi.org/10.5194/bg-11-4881-2014, 2014.

Thyssen, M., Tarran, G. A., Zubkov, M. V., Holland, R. J., Grégori, G., Burkill, P. H., and Denis, M.: The emergence of automated high-frequency flow cytometry: revealing temporal and spatial phytoplankton variability, *J. Plankton Res.*, 30, 333–343, https://doi.org/10.1093/plankt/fbn005, 2008.

Thyssen, Melilotus, Gerald J. Grégori, Jean-Michel Grisoni, Maria Luiza Pedrotti, Laure Mousseau, Luis F. Artigas, Sophie Marro, Nicole Garcia, Ornella Passafiume, and Michel J. Denis. Onset of the spring bloom in the northwestern Mediterranean Sea: influence of environmental pulse events on the in situ hourly-scale dynamics of the phytoplankton community structure. *Frontiers in microbiology* 5: 387, https://doi.org/10.3389/fmicb.2014.00387, 2014.

Tzortzis, R., Doglioli, A. M., Barrillon, S., Petrenko, A. A., d'Ovidio, F., Izard, L., Thyssen, M., Pascual, A., Barceló-Llull, B., Cyr, F., Tedetti, M., Bhairy, N., Garreau, P., Dumas, F., and Gregori, G.: Impact of moderately energetic fine-scale dynamics on the phytoplankton community structure in the western Mediterranean Sea, *Biogeosciences*, 18, 6455–6477, https://doi.org/10.5194/bg-18-6455-2021, 2021.

Zhang, Y., Jiao, N., and Hong, N.: Comparative study of picoplankton biomass and community structure in different provinces from subarctic to subtropical oceans, *Deep-Sea Res. Pt. II*, 55, 1605–1614, https://doi.org/10.1016/j.dsr2.2008.04.014, 2008.

---

## Author Response (AR1)

**Authors's reply to Anonymous Referee #1 comments on on egusphere-2022-1008 Tzortzis et al.**

Dear referee,

Thank you very much for your constructive comments and suggestions, as well as your corrections of the grammar and spelling in order to improve the quality of the manuscript. According to the Biogeosciences guidelines, we reworked the manuscript following your suggestions and those of the other referee. Previously we addressed the main points you raised. Hereafter, we provide an update of our previous response: Your comments are in black and our previous answers in blue, we add revised versions of each section in green.

**General Comments**

In the manuscript "The contrasted phytoplankton dynamics across a frontal system in the southwestern Mediterranean Sea", Tzortzis et al. compare the phytoplankton communities at two different water masses separated by a frontal region. The work is original. The sampling design to analyze the two water masses separated by a front, and the phytoplanktonic community that characterizes them, is very interesting. Especially having a tool like the CytoBuoy.

We thank you for these positive comments about our work.

However, the manuscript needs improvement in many aspects before it can be considered for publication. In general, it is a disorganized text. The story does not flow, the paragraphs do not focus on clear topics, there are very long and confusing sentences, there are methodological descriptions in the results and results in the discussion... All sections should be carefully reviewed and improved. Especially the introduction and discussion.

We apologize for not being clear enough. We will rework each section in order to make our scientific message flow, taking into account your useful suggestions and those of the second referee.

The study area is barely described or named, why is it so relevant to focus on that particular front (besides the scope of the satellite)?

We will improve the description of the study area, including more information on the general dynamics of the Algerian sub-basin and on the specific situation encountered during our 2018 cruise. The importance of the frontal area studied in the present article is due to the fact that most of the in-situ studies related to the physical-biological coupling at finescale have focused on extreme situations occurring in boundary currents, where intense fronts and dramatic contrasts in water properties are found but are not representative of the global ocean. Indeed, vast oceanic regions are dominated by weak fronts continuously created,

moved and dissipated, which separate different water masses with similar properties. Very few studies exist in these regions due to the difficulty of performing in situ experiments over these short-lived and small features. In our previous work Tzortzis et al. (2021) we showed that the fine-scale front observed during our 2018 cruise in an oligotrophic and moderately energetic region (the SW Mediterranean) also drove phytoplankton diversity patterns. In the present study we aim at explaining how, by combining the available hydrological and cytometric data recorded at high spatio-temporal frequency.

Moreover, in the first paragraph of the introduction, the authors mention the context of climate change. How will climate change affect the presence and intensity of the fronts? And in turn, how a possible change in the intensity and frequency of the fronts will affect the associated phytoplankton communities? As a reader, I feel like I am being shown an interesting image, but in black and white instead of full color.

The impact of climate change on the ocean is a topic of current research, but many uncertainties remain (Collins et al., 2010 ; Cheng et al., 2022 ; Yao et al., 2022, etc). The predictions concerning the consequences of global warming on ocean circulation are already observable in the Southern Ocean, where the Subantarctic Front is particularly affected (Jia-Rui Shi et al., 2021).
Future climate-driven changes to ocean circulation patterns have the potential to alter dispersal pathways, potentially affecting the ability of species to track and adapt to climate change (Wilson et al., 2016). This impact may be very important in the Mediterranean Sea (MerMex Group, 2011).
The fact that the links between finescale and biological processes are still unknown (see below for more explanation and references) implies that the impact of climate change on them is speculative.
The objective of this study is to contribute to improving our knowledge of the finescale physical-biological coupling, by focusing on the distribution of the phytoplankton and its possible explanation (cell cycle). Our study does not aim to understand the consequence of climate change-induced changes in fronts. Since our study is not related to climate change, we decided to remove the reference to climate change in the Introduction in the reworked manuscript.

**Specific Comments**

**Introduction**

The authors describe in ~15 lines the relevance of phytoplankton and ocean fronts. In my opinion, more information is needed. Knowing the abundance and diversity of phytoplankton is important, but its role in the carbon cycle should also be highlighted, which changes depending on whether the community is dominated by small or large species. Moreover, an oceanic front is not defined, the authors describe briefly the

physical-biological interaction. I also miss an intro to the study region.

We have reworked this part (see below the revised version of the Introduction).

After that brief introduction, the difficulty of an in situ study is described, and without continuing the story fluently, they begin to talk about an oceanographic campaign/project. I think the information is relevant, although the more technical details should be indicated in the methodology.

We will move the technical details of the campaign in the methodology, and improve the transition between the difficulty of an in situ study and our particular campaign.

Finally, the last paragraph is confusing. There is a lot of information, but not all makes sense, and it is kind of disorganized. The last paragraph of the introduction should clearly define the objectives of the study and how the authors would answer them.

We will clarify this part in the reworked manuscript following these suggestions:
"The objectives of this study are to assess whether the observed contrasted abundances across the front are due to different growth and loss rates. Using high-frequency flow cytometry measurements across the front dividing two water masses, we were able to separately analyze each phytoplankton functional group and reconstruct their biovolume dynamics over a diel cycle in each water mass."

Ln 16: Phytoplankton are essential for marine ecosystems, but not really for the functioning of the oceans... Oceans can function without life.
Ln 16-17: Revise the sentence. The $CO_2$ assimilated by phytoplankton can be exported to deep waters when they die or are partially eaten, being decomposed at depth; that's the biological pump. But not when they are eaten by higher trophic layers.

Our sentence was unclear and we plan to modify it as follow:

"Phytoplankton plays a crucial role in the oceans by regulating climate and forming the base of the food chain (Sterner et al., 1994). Its capacity to perform photosynthesis influences the global carbon cycle, by fixing $CO_2$ and exporting it into the ocean depth through the biological pump. Phytoplankton production also supports higher trophic levels, impacting ecosystem functioning."

Ln 20-22: Add references.

We plan to add here the following references: (Clayton et al., 2014 ; Mahadevan 2016 ; Lévy et al., 2018).

Ln 22-23: Revise the sentence. It seems that the idea the authors are trying to convey

is that the temporal scale of growth/evolution of the phytoplankton community is due to a fine-scale coupling. It seems that the fronts are the ideal environment for phytoplankton when it is not necessarily true.

We plan to better explain our point of view by adding a sentence as follows:
"Indeed, the phytoplankton dynamics temporal scale is of the same order of magnitude as the one of finescale processes such as eddies, filaments or fronts, suggesting the possibility of a close coupling between phytoplankton growth and finescale forcing. This does not necessarily imply that phytoplankton can grow only in presence of fronts, but fronts create favorable conditions for phytoplankton and life in general (Clayton et al., 2014 ; Mahadevan 2016 ; Lévy et al., 2018)."

Ln 27: Here there is a change of topic, please, start a new paragraph.

We will rework the manuscript to take into account your suggestions.

Ln 24-27: Revise sentence. First, the sentence is too long.
In my opinion, the use of "could" makes the facts described less solid. It is established that fine-scale frontal structures induce vertical velocity. Is there any study where no vertical velocity is associated with these structures?

We will modify it in the manuscript, following your grammar suggestion.

Different physical processes associated with finescale structures are able to generate vertical velocities, such as deformations of the flow and spatial inhomogeneities (Giordani et al., 2006), eddy perturbation (Martin & Richards, 2001 ; Pilo et al., 2018), linear Ekman pumping (McGillicuddy et al., 1998 ; Gaube et al., 2015), or eddy-wind interactions (McGillicuddy et al., 2007). That is why to our knowledge the majority of studies about vertical velocities have focused on finescale structures (fronts or eddies) because these are suitable places for the formation of vertical motions (Rudnick, 1996 ; Pascual et al., 2015 ; Rousselet et al., 2019 ; Barceló-Llull et al., 2021).

Vertical velocities do not modulate light availability. Vertical velocities move the phytoplankton cells along the water column and depending on the "resulting" depth they will have more or less light.

We plan to add refs and to modify the sentence as follow:
"Previous studies have well established that vertical motions impact biogeochemistry (Mahadevan & Tandon, 2006 ; Mahadevan, 2016 ; McGillicuddy, 2016). Upward vertical velocities drive deep nutrients into the euphotic layer and also move the phytoplankton cells along the water column resulting in changing light conditions."

Ln 33-35: I do not understand this sentence. Are you saying that little is known about the phytoplankton diel cycle? Not only there are laboratory experiments, but also

models, in particular individual-based models, that study this fact. For example, several studies by Geider et al.

We just want to highlight the importance of performing in situ measurements at high frequency to sample the phytoplankton cycle in natural conditions. Although a lot of knowledge has been obtained from laboratory experiments or models, only a few studies have performed in situ measurements at high frequency and resolution. This is a reason to lead cruises using Lagrangian sampling strategy and automated flow cytometer.
Thank you for your reference, we will include it in the sentence as follows:
"Although progress in the understanding of phytoplankton cell cycle has been obtained from incubation, sample manipulation (Worden and Binder, 2003) and models (Geider et al., 1997 ; MacIntyre et al., 2000), performing in situ measurements at high frequency and resolution is a necessity to better understand these biological processes and their responses to the environment. An efficient solution is to lead Lagrangian cruises using automated flow cytometers sampling at high frequency in order to resolve the phytoplankton diurnal cycle in situ, which is challenging using more conventional methods such as cultures or counting by optical microscopy (Thyssen et al., 2008 ; Fontana et al., 2018)."

Ln 44-48: Please, rephrase these sentences with a clearer and simpler message.

We plan to modify the sentence as follow:
"In our previous work, we identified the two water masses as both Atlantic Water (AW), but at different stages of mixing due to their different routes in the Western Mediterranean Sea. The AW located south of the front, characterized by absolute salinity ($S_A$) between 37 g kg$^{-1}$ and 37.5 g kg$^{-1}$, corresponds to AW more recently entered into the Mediterranean Sea by the Gibraltar strait, thus being named "younger AW". The AW found north of the front is characterized by a higher $S_A$ (37.5 g kg$^{-1}$ to 38 g kg$^{-1}$) and corresponds to the surface water having circulated in the cyclonic gyre of the western Mediterranean for some years and then is referred to as "older AW". This water is also referred to as "local AW" (Barceló-Llull et al., 2019) or "resident AW" (Balbín et al., 2012)."

Ln 46-48: Could the authors provide some details about the nutrient concentration of both water masses?

Unfortunately, it was not possible for both technical and funding reasons to perform nutrient measurements during the 2018 cruise, that is why we cannot provide nutrient concentrations of both water masses. Nevertheless we acknowledge the importance of this information and these measurements are planned for our future cruise planned this spring (mid-April to mid-May 2023).

Ln 49-50: This sentence looks like part of the results section. I understand that you are referring to Tzortzis et al. (2021), but it is not clear.

We will modify the sentence as follows:

"Coupled with these hydrological measurements, measurements of phytoplankton abundance by flow cytometry were also described in our previous study. Tzortzis et al. (2021) showed that contrasted phytoplankton abundances were observed in these two water masses, with the smallest phytoplankton such as *Synechococcus* dominating south of the front in the younger AW, while microplankton is more abundant north of the front in the older AW."

Ln 51: This study, or Tzortzis et al. (2021)? I imagine is Tzortzis et al. (2021), then this first sentence and probably the open questions should be in the previous paragraph.

We apologize for the lack of clarity. We will not change the paragraph but we will modify the sentence as follows:

[revised manuscript text omitted]

Ln 70-73: I don't think this information is relevant.

We will move the sentence about SWOT in the perspective section.

Ln 73-76: These facts should be in the introduction.

As suggested, we will move this part in the new version of the introduction (see above)

Ln 77: Please, consider indicating that the measures have a high spatial and temporal resolution.
Ln 77: I am not sure if you can use in situ sensors when they are on board. It is kind of repetitive.
Ln 77-70: Here you are describing in situ measurements, that are described in the next subsection.
Ln 80-82: Repetitive information (introduction).
Ln 76 and 83: Please, describe the sampling strategy details in a single paragraph.

Ln 84 and 85: Please, mention the source of the remote sensing datasets.

Following your suggestions we will modify Figure 1 (see page 21 of this document). We also modified this section as follows:

**The Sampling strategy (Ln 81-101 in the reworked manuscript)**

The PROTEVSMED-SWOT cruise, dedicated to the study of finescale dynamics, was conducted in the south of the Balearic Islands between April 30th and May 18th 2018, on board the R/V Beautemps-Beaupré (Fig. 1a). This cruise followed an adaptive Lagrangian strategy to measure at high spatial and temporal resolution several physical and biological variables with both in situ sensors and analysis of the sea surface water intake. The vessel route was designed ad-hoc on the basis of daily remote sensing dataset provided by the Software Package for an Adaptive Satellite-based Sampling for Oceanographic cruises (SPASSO, https://spasso.mio.osupytheas.fr, last access: April 22, 2023). SPASSO used altimetry-derived currents from the Mediterranean regional product (nrt_med_allsat_phy_l4) AVISO ("Archiving, Validation and Interpretation of Satellite Oceanographic", https://www.aviso.altimetry.fr, last access: April 22, 2023) and ocean color observations. Chlorophyll a concentrations ([chla], level 3, 1 km resolution, MODISAqua and NPPVIIRS sensors combined (after May 27, 2017) into a new product called MULTI) were provided by CMEMS, "Copernicus Marine Environment Monitoring Service", https://marine.copernicus.eu, last access: April 22, 2023. In addition, CLS provided the surface Chl concentration composite products, with the support of the CNES. They were constructed using a simple weighted average over the previous 5 days of data gathered by the Suomi/NPP/VIIRS sensor. SPASSO generated maps of dynamical and biogeochemical structures in both near real time (NRT) and delayed time (DT). Maps of [chla] allowed us to identify two water masses, characterized by distinct [chla] values and separated by a zonal front at around 38° 30' N. This front was also detected using in situ horizontal velocities, temperature and salinity, as described in Tzortzis et al. (2021). These two water masses were sampled along a designated route of the ship, represented in black in Fig. 1b. Special attention was paid to adapting the temporal sampling in order to measure the phytoplankton diel cycle in each water mass. This was achieved by continuously sampling across both water masses along transects. While the ship did not remain in each water mass for 24h, day-to-day variability remained low and measurements from several days were combined into one diel cycle (Fig. 1c). The shape depicted by the ship's track led us to call these areas north–south (NS) hippodrome (bold black line in Fig. 1b) performed between 11 May and 13 May 2018.

Ln 91-94: What are the temporal and spatial resolution of the temperature and salinity measurements?

Temperature and salinity were measured thanks to a thermosalinograph (TSG). The TSG was equipped with two sensors: a CTD Sea-Bird Electronics SBE 45 sensor installed in the wet lab, connected to the surface water and which continuously pumped seawater at 3 m depth ; and an SBE 38 temperature sensor installed at the entry of the water intake. The TSG

measurements are achieved **each 30 min,** which corresponds to a **~ 2 km spatial resolution** at typical ship speeds.

Ln 94-115: One paragraph. Moreover, revise the information provided, there is some repetitiveness regarding the optical signals.
Ln 104: 1.5 cm$^3$ is the water volume analyzed? Please, consider expressing the volume in mL, in my experience, it is a more common unit used in this kind of study.
Ln 112: Again, please consider using cell per mL
Ln 115-116: Totally out of place.

We used a subunit of the international system (SI) as requested by Biogeosciences, but below we will convert it into mL as in the reworked manuscript, following your suggestions.

**In situ measurements ( Ln 103-131 in the reworked manuscript)**

During the cruise, the irradiance (wavelengths between 400 and 1000 nm) was measured by a CMP6 pyranometer (Kipp and Zonen; https://www.campbellsci.fr/cmp6, last access: April 22, 2023). Temperature and salinity were measured by a thermosalinograph (TSG). The TSG was equipped with two sensors: a CTD Sea-Bird Electronics SBE 45 sensor installed in the wet lab, connected to the surface water and which continuously pumped seawater at 3 m depth ; and an SBE 38 temperature sensor installed at the entry of the water intake. The TSG measurements were taken every 30 min, which corresponds to around 2 km spatial resolution at typical ship speeds. The data were converted into conservative temperature ($\Theta$) and absolute salinity ($S_A$) using the TEOS-10 standards of McDougall et al. (2012). To automatically sample and analyze phytoplankton cells, an automated CytoSense flow cytometer (CytoBuoy, b.v. ; (Dubelaar et al., 1999; Dubelaar and Gerritzen, 2000)) was installed on board and connected to the seawater circuit of the TSG. The flow cytometer sampled the seawater in a dedicated small container called "subsampler". The subsampler isolates the seawater every 30 min which allows us to ignore the movement of the ship, while the flow cytometer performed its analysis. Between two consecutive samples the subsampler was flushed continuously by the seawater circuit of the ship in order to clean and renew the seawater. A sheath fluid made of 0.1 μm filtered seawater stretched the sample in order to separate, align, center and drive the individual particles (i.e. cells) through a laser beam (488 nm wavelength). Several optical signals were recorded when each particle crossed the laser beam: the forward angle light scatter (FWS) and 90° side-ward angle scatter (SWS), related to the size and the structure (granularity) of the particles. Two distinct fluorescence emissions induced by the light excitation were also recorded, a red fluorescence (FLR) induced by chlorophyll a content and an orange fluorescence (FLO) induced by the phycoerythrin pigment content. The CytoUSB software (Cytobuoy b.v.) was used to configure and control the flow cytometer and set two distinct protocols. The first protocol (FLR6) was dedicated to the analysis of the smaller phytoplankton, using a red fluorescence (FLR) trigger threshold fixed at 6 mV, and a volume analyzed set up at 1.5 mL. The second protocol (FLR25) targeted nanophytoplankton and microphytoplankton with a FLR trigger level fixed at 25 mV and an analyzed volume of 4 mL. The FLR trigger was used to discriminate the red

fluorescing phytoplanktonic cells from other particles (such as heterotrophic prokaryotes, nanoflagellates, ciliates, etc.). Recorded data were analyzed with the CytoClus software (Cytobuoy b.v.) which retrieves information from the 4 pulse shapes curves (FWS, SWS, FLO, FLR) obtained for every single cell. These curves were then projected into distinct two-dimensional planes (cytograms) by computing the curves' integral. Using a combination of various cytograms (e.g., FWS vs. FLR, FLO vs. FLR) allows us to determine optimal cell clusters (i.e, cells sharing similar optical properties). These clusters have been demonstrated in the literature to represent phytoplankton functional groups (PFGs) (Dubelaar and Jonker, 2000; Reynolds, 2006; Thyssen et al., 2008; Edwards et al., 2015; Thyssen et al., 2022). Finally, the PFGs abundance (cells per milliliter) and mean light scatter and fluorescence intensities were extracted from each sample.

Ln 118-124: Please, make it clear that this size-structured population model was applied to every phytoplankton population/group identified previously using the CytoBuoy.

We applied the size-structured population model of Sosik et al. (2003) to every phytoplankton group identified by the CytoSense flow cytometer. However, we only analyzed the results obtained for *Synechococcus* and the two nanophytoplankton groups because the size distributions of the picophytoplankton groups were very noisy. Furthermore, microphytoplankton and Cryptophytes were not abundant enough to allow a reliable determination of their abundances and cell cycle.

Ln 126: To use the model, the light scatter signal (FWS) recorded for each cell by the flow cytometer must be converted to size (diameter) using a power law relationship (Sosik et al., 2003), and then to biovolume (v).
I imagine that to convert size into volume you are considering that all the species are spherical. Then, please consider indicating this fact and that you are converting the FWS signal to Equivalent Spherical Diameter.Please, indicate the units of both measurements. Also for the rest of the variables (t, E, g, μ*,...).
Ln 128: I am not sure if N is the number of cells in all the size classes or at each size class.
Ln 133: How many size classes were determined and how? Does it follow a log distribution?
Ln 135: I am not sure what exactly is "this probability". Is it the probability of cells growing in a time interval? Is it a probability or a proportion?
Ln 141: Instead of however, besides seems more appropriate.
Ln 141-142: Repetitive information.
Ln 147: I consider kind of inappropriate the use of a "decrease in cell size", it is a division. A phytoplanktonic cell decreases in size if the growth conditions are not optimal, and that is not an indication that there is a doubling event.
Ln 150-151: This sentence is confusing. Why do you talk about N(0) when is not used in the equations 5. (Two equations = two labels, please. Similarly, with equations 8)
Ln 153: A(t) is a tridiagonal transition matrix that contains.

Ln 159: Could you elaborate on what you mean by optimal parameters, please?

Ln 160: Standard deviations of the errors?

Ln 166: There is no information about this equation.

Ln 167-169: I do not understand this explanation.

Moreover, the definition of "bar l" (I do not know how to write the loss symbol here) confuses me. If it is the daily average population loss rate, how dt is 1 hour? On the other hand, what do you mean exactly by loss? The number of cells moving from one size class to another, or death?

What is the description of T1day NT0?

I have no experience using this kind of model, but any reader should be able to understand the methodology followed in the study without having to read previous studies. So please, review this section carefully and try to make it as clear as possible.

In order to follow all your comments above, we will completely rework section 2.3 of the manuscript, concerning the size-structured population model. We hope that this new version here below will clarify this approach.

**The size-structured population model (Ln 132-233 in the reworked manuscript)**

We used the size-structured population model described by Sosik et al. (2003) and adapted by Dugenne et al. (2014) and Marrec et al. (2018), to estimate the in situ growth rates of every phytoplankton group identified by the CytoSense flow cytometer, in the older AW and the younger AW. Before applying the model, we reconstructed a daily cycle of 24 h in the two water masses for each phytoplankton group. We use the term reconstruction because the ship did not spend 24 h in a row in each water mass but sailed along two routes, each forming a sort of racetrack passing alternately through the two water masses (Fig. 1b, 1c). By eliminating the dates and keeping the associated sampling times, the 24-hour diel cycle can be reconstructed for each water body (Fig. 1c). This relies on the hypothesis that the phytoplankton community and dynamics remained similar over the two days, and that hydrology and physics for each water mass remained alike during sampling. We also reconstructed the 24-hour irradiance in the two water masses (Fig. A2), because one of the most important parameters of this model is irradiance, since cell growth is dependent on light exposure due to photosynthesis.

The model of Sosik et al. (2003) uses as input the phytoplankton cell volume (biovolume) derived from cell light scatter intensities (FWS) (Eq. 1). Biovolumes were estimated using coefficients previously obtained by measuring a set of silica beads with the flow cytometer following the same settings used for phytoplankton analysis. The coefficients $\beta_0$ and $\beta_1$ used to convert FWS (arbitrary units, a.u.) to biovolume $v$ ($\mu m^3$) were derived from a log-log regression between FSW and silica bead volumes.

$v = exp(\beta_0) \times FWS^{\beta_1}$     with in our case $\beta_1 = 0.9228$ and $\beta_0 = -5.8702$     (Eq. 1)

In the size-structured population model, cells are classified into several size classes according to their dimensions at time t. Classes are logarithmically spaced as follows: for $i$ in $1,2,...,m$ $v_i = v_1 2^{(i-1)\Delta v}$ where $\Delta v$ is constant and chosen to ensure that size classes cover the entire observed biovolume $v$, from $v_1$ to $v_m$ (see figure below). For Synechococcus, $\Delta v = 1/6$ with $\Delta v$ constant and $m = 40$, so that the model size classes encompassed our full measured size distributions (0.0279-2.5209 μm).

[Figure]

Figure : Cell cycle stages in the size-structured population model. Cells may grow to the next size class ($\gamma$) or be at equilibrium $(1 - \gamma(t))(1 - \delta(v, t))$. Above a particular size, cells are large enough to divide in two daughter cells with probability ($\delta$). Figure adapted from Sosik et al. (2003).

At any time $t$, the number of cells in size classes **N** (and **w** its corresponding normalized distribution), was projected to $t + dt$ via matrix multiplication (Eq. 2):

$$\textbf{\textit{N(t + dt)}} = \textbf{\textit{A(t) N(t)}} \qquad \text{and} \qquad \textbf{\textit{w(t + dt)}} = \frac{A(t)\,N(t)}{\Sigma\,A(t)\,N(t)} \qquad \text{(Eq. 2)}$$

We chose $dt = 10$ min (i.e., $\frac{10}{60}$ h) as Sosik et al. (2003) and Dugenne et al. (2014), because for this time step, cells are unlikely to grow more than one size class.

*A(t)* is a tridiagonal transition matrix that contains:
   1) $\gamma$: the probability of cellular growth

2)  $\delta$: the probability of cells entering mitosis
3)  the cells stasis, i.e., the probability for cells to maintain their state (i.e size) in equilibrium during the temporal projection.

**Probability of cellular growth**

The probability of cells growing to the next size class ($\gamma$) depends only on the light intensity (irradiance) necessary for photosynthesis, expressed as (Eq. 3):

$$\gamma(t) = \gamma_{max} \left[ 1 - exp\left(- \frac{E(t)}{E^*}\right)\right] \qquad \text{(Eq. 3)}$$

$\gamma_{max}$: maximum proportion of cells growing (dimensionless quantity)
E: irradiance ($\mu$E m$^{-2}$ s$^{-1}$)
E*: irradiance normalizing constant ($\mu$E m$^{-2}$ s$^{-1}$)

**Probability of cells entering mitosis**

According to Dugenne et al. (2014), $\delta$ expresses a proportion (between 0 and 1) modeled by the combination of two Normal distributions ($\mathcal{N}$). One is linked to the cell size, the other is linked to the time of cell division. Both imply an optimum, reached at $\bar{v}$ and $\bar{t}$ respectively, for cell division above which the cell size and the timing of division is suboptimal (Eq. 4).

$$\delta(t, v) = \delta_{max} \, \mathcal{N}(\bar{v},\sigma_v^2) \, \mathcal{N}(\bar{t},\sigma_t^2) \qquad \text{(Eq. 4)}$$

$\delta_{max}$: maximum proportion of cells entering mitosis (dimensionless quantity)

$\bar{v}$: mean of the size Normal distribution ($\mu$m$^3$)

$\sigma_v$: standard deviation of the size Normal distribution ($\mu$m$^3$)

$\bar{t}$: mean of the time Normal distribution (h)

$\sigma_t$: standard deviation of the time Normal distribution (h)

**Cells stasis**

A third functional proportion is included in the transition matrix $A(t)$, to represent cell stasis. Since this function illustrates a non-transition, it is modeled by the proportion of cells that neither divided nor grew between $t$ and $t + dt$.

$$[1 - \gamma(t)] \, [1 - \delta(t, v)]$$

Optimal parameters

The set of parameters, $\theta$ is estimated by maximum likelihood function, assuming errors between observed $\boldsymbol{w}$ and predicted $\widehat{w}$ normalized size distributions (Eq. 6, 7, 8). Their standard deviations are estimated by a Markov Chain Monte Carlo approach (Geyer, 1992 ; Neal, 1993) that sample $\theta$ from their prior density distribution, obtained after running 200 optimizations on bootstrapped residuals to approximate the parameter posterior distribution using the normal likelihood. (The likelihood function represents the probability of random variable realizations conditional on particular values of the statistical parameters).

$$\theta = [\gamma_{max}, E^*, \delta_{max}, \bar{v}, \sigma_v, \bar{t}, \sigma_t] = argmin(\textstyle\sum(\theta)) \qquad \text{(Eq. 6)}$$

$$\sum(\theta) = \sum_{t}^{t+dt} \sum_{i=1}^{m} (w(t) - \widehat{w}(t,\theta))^2 \qquad \text{(Eq. 7)}$$

$$\widehat{N}(t,\theta) = A(t\text{-}dt, \theta)\, N(t\text{-}dt) \qquad \text{(Eq. 8)}$$

$\widehat{w}$ is computed from $\widehat{N}$ following Eq. 2. The fit of the model is quantified using two numbers:

the loss rate ($\sum(\theta)$, lower indicates better fit), and the correlation between the observed and

modeled mean biovolumes $\bar{v}_{obs}$ and $\bar{v}_{mod}$ over the diel cycle (corr($\bar{v}_{obs}$ , $\bar{v}_{mod}$), higher indicates better fit). Table 1 provides the model parameters being optimized.

Table 1: Model parameters being optimized.

| Parameters | Definition | Interval | Units |
|---|---|---|---|
| $\gamma_{max}$ | Max proportions of cells in growing phase | [0, 1] | Ø |
| $E^*$ | Irradiance normalizing constant | [0,∞[ | µE m$^{-2}$ s$^{-1}$ |
| $\delta_{max}$ | Max proportion of cells entering mitosis | [0,1] | Ø |
| $\bar{v}$ | Mean of the size Normal distribution | [$v_{min}$ , $v_{max}$] | µm$^3$ |

| $\sigma_v$ | Standard deviation of the size Normal distribution | $[10^{-06}, \infty[$ | $\mu m^3$ |
|---|---|---|---|
| $\bar{t}$ | Mean of the time Normal distribution | $[1, 24 \frac{1}{dt} + 1]$ | hours |
| $\sigma_t$ | Standard deviation of the time Normal distribution | $[10^{-06}, \infty[$ | hours |

Growth rate and loss rate

Once optimal parameters are identified, the model estimates a population intrinsic growth rate $\mu_{size}$ , and a specific loss rate $l$, integrated over a 24 h period. The method uses the fact that the observed size distribution **N** is the result of both growth and loss processes, while the time projection of the initial size distribution **N**(0) using the model, **N̂,** is only the result of growth processes. The growth rate is calculated at each time step following Eq. 9, and integrated over 24 h. 200 iterations by a Markov Chain Monte Carlo were run to estimate the standard deviation of group-specific growth rates.

$$\mu_{size} = \frac{1}{dt} ln\left(\frac{\sum\limits_{i=1}^{m} \widehat{N_i}(t+dt)}{\sum\limits_{i=1}^{m} \widehat{N_i}(t)}\right)$$

$i$: i$^{th}$ size class

$\widehat{N}$: predicted size distribution (cells cm$^{-3}$)

$m$: number of size classes

$dt$: time step (h)

$\mu_{size}$: growth rates (day$^{-1}$)

An independent growth rate estimation was obtained as $\mu_{ratio} = ln(\bar{v}_{max} / \bar{v}_{min})$ where $\bar{v}_{min}$ and $\bar{v}_{max}$ are the minimum and maximum of the mean observed biovolume $\bar{v}_{obs}$ over the diel cycle (Marrec et al., 2018). $\mu_{ratio}$ represents a minimum estimate of the daily growth rate, that would be observed if cells synchronously only grew from the time $\bar{v}_{min}$ is observed (typically dawn) to the time $\bar{v}_{max}$ is observed (typically dusk), and only divided while $\bar{v}$ decreases. Since the model allows for any cell to grow, divide or be at equilibrium over the entire integration period (asynchronous populations), $\mu$ size is expected to be higher than $\mu_{ratio}$. In practice, $\mu_{ratio}$ is sensitive to noise in the data and is only provided here as an alternative estimate of the growth rate that does not rely on the model.

The population loss rate *l* is obtained by difference between the intrinsic growth rate $\mu_{size}(t)$ and the temporal change in logarithmic observed size distribution **N**, which represents the net growth rate $r(t) = \mu_{size}(t) - l(t)$ so that:

$$\bar{l} = \int\limits^{t} \mu_{size}(dt) \ - \ \frac{1}{dt} \ ln(\frac{N(t+dt)}{N(t)})$$

**Results**

Ln 176-188: This information should be included in the methodology section. Also, at the end of this explanation, it will be interesting to indicate how to convert the scatter signal to size and volume.The details about how every species was differentiated, in my opinion, are not necessary, therefore I propose the authors move it to the supplementary, together with Figure 2.

Following your suggestions, we will modify the text by moving: the description of how to identify the phytoplankton groups in the methodology section, and the remaining information and the figure to the **Appendices**.

**Spatio-temporal distribution of phytoplankton abundances in the two water masses (Ln 234-246 in the reworked manuscript)**

The sampling strategy adopted during PROTEVSMED-SWOT enabled us to sample two water masses with different properties. The map of the satellited-derived surface [chla] shows higher concentration in the Northern part of the sampling route, corresponding to older AW, than in the the Southern part, corresponding to younger AW (Fig. 1b). Figure 3 shows the properties of the sea surface water as a function of time (from 11 May 00:00 to 13 May 12:00 UTC) along the sampling route. The older AW is characterized by a colder temperature and higher values of salinity than the younger AW. Figure 3 also displays the abundances of each phytoplankton group over these two water masses. *Synechococcus* and Pico2 are the most abundant. They present a clear surface distribution pattern, with high abundances in the warm and low salinity water, corresponding to the young AW. A similar distribution is observed for Pico1, Pico3 and RNano but with lower abundances than *Synechococcus* and Pico2. The abundances of SNano, PicoHFLR and Cryptophyte show less contrasts along the cruise than the previous groups, nonetheless the highest abundances can be distinguished in the younger AW, in particular in the second and third passage (transect) across this water mass. Finally, microphytoplankton is the less abundant group, but it clearly shows a contrast between the two water masses, opposite to the one of the other phytoplankton groups.

Ln 210-211: The information about the figures does not fit here. It will be more appropriate to move to the beginning of the next paragraph.

We will move it in the reworked version of the manuscript.

On the other hand, please explain the background information. Does it make reference to the proportion (percentage) of cells of each biovolume? If it is a percentage, why it does not vary between 0 and 1?

We will change the terms, instead of proportion we will use frequency; indeed, the background colors represent the number of particles of a specific biovolume. For a given time, the sum of the frequency is equal to the total number of particles.

Ln 211-216: How was reconstructed the 24-hour irradiance curve should be explained in the methodology.

We will move this part in the methodology.

Ln 221, 222, 231, and 232: Please, include the standard deviation value together with the mean value.

We will add that in the text and maintain it in the table, too.

Ln 217-233: Please, explain in this section why there is no information about the other 6 groups identified.

We will move the explanation from the Discussion in this section:
"We have also modeled the diurnal cycle for the picophytoplankton groups, i.e., Pico1, Pico2, Pico3, PicoHFLR. However, we obtained very noisy size distributions and couldn't obtain a valid measurement of the growth rates, hence these distributions are not considered further in this study. As microphytoplankton and Cryptophytes were not abundant enough to allow a reliable determination of their abundances and cell cycles, they were not taken into consideration in this study."

**Phytoplankton cellular growth and division in the two water masses (Ln 247-272 in the reworked manuscript)**

The phytoplankton diurnal cycle was reconstructed in the two water masses using the size-structured population model originally developed by Sosik et al. (2003). Figures 4, 5, 6 represent the phytoplankton size distribution (i.e., biovolume) observed in situ and predicted by the model over 24 h for *Synechococcus*, RNano and SNano, respectively. From the predicted biovolume it is possible to derive specific growth ($\mu_{size}$) and a loss ($l$) rates, summarized in Table 2 for the different phytoplankton groups in the two water masses, along

with metrics of model performance. We also attempted to model the diurnal cycle for the picophytoplankton groups, i.e., Pico1, Pico2, Pico3, and PicoHFLR. However, their very noisy size distributions prevented us from obtaining reliable growth rate estimates. Similarly, microphytoplankton and Cryptophytes were not abundant enough to allow a reliable determination of their abundances and cell cycles. These cytometric groups are thus not considered further in this study.

For *Synechococcus*, in the older AW the prediction of the model (i.e., predicted biovolume) is similar to the observed size distribution (i.e., observed biovolume). Both display a day-long large size-class distribution centered approximately on 0.3 $\mu m^3$. In the younger AW (Fig. 4a, c) the distributions of observed and predicted biovolume are narrower than in the older AW and centered approximately on 0.2 $\mu m^3$ (Fig. 4b, d). As a consequence, the older AW is populated by larger cells of *Synechococcus* (mean observed biovolume $\bar{v}$ obs = 0.38 ± 0.04 $\mu m^3$ ) than in the younger AW (mean biovolume $\bar{v}_{obs}$ = 0.21 ± 0.04 $\mu m^3$ ) (Table 2). Growth and loss rates also differ between the two water masses. In the older AW, the large cells of *Synechococcus* have a growth rate $\mu_{size}$ = 0.24 ± 0.91 $d^{-1}$ and a loss rate $l$ = 0.36 d −1 , whereas in younger AW the smaller cells are characterized by higher growth ($\mu_{size}$ = 0.68 ± 1.56 $d^{-1}$) and loss ($l$ = 0.48 $d^{-1}$) rates.

Relative to *Synechococcus*, cell size distribution and growth and loss rates are less contrasted between the older and younger AW for SNano (Fig. 6) and even more so RNano (Fig. 5). The mean observed RNano biovolumes are similar in the older and younger AW (63.5 ± 2.67 $\mu m^3$ and 61.2 ± 5.23 $\mu m^3$ , respectively) (Table 2). For SNano, similar to *Synechococcus*, the older AW is predominantly composed of larger cells ($\bar{v}_{obs}$ = 85.0 ± 1.98 $\mu m^3$ ) than in the younger AW ($\bar{v}_{obs}$ = 63.8 ± 4.45 $\mu m^3$ ). For both Nano groups, growh rates are generally very low in both water masses ($\mu_{size}$ < 0.1 $d^{-1}$ ). Loss rates are higher than growth rates, except for RNano in the younger AW (negative loss rate implying an external input of cells such as by advection). However, the corresponding optimization factor is the highest observed across the 6 modelisations, indicating this result is subject to caution.

**Discussion**

Ln 236: Please, add some references.

Although it has been clearly demonstrated that phytoplankton plays a fundamental role in the ocean ecosystem functioning **(Watson et al., 1991 ; Field et al., 1998 ; Allen et al., 2005)**, numerous questions remain about their population dynamics in relation to finescale structures.

Ln 245-246: What do you mean by transiting in all the cell cycle stages? That they are growing and dividing?
Ln 254: What do you mean by extensive distribution?

We apologize for the lack of clarity, our intent here is to compare the *Synechococcus* distributions in the two different water masses. We will modify the text as follows:

"In the older AW the cells have all along the day a large size-class distribution centered approximately at 0.3 μm$^3$ while in the younger AW (Fig. 4a, 4c) the distribution is narrower and centered approximately at 0.2 μm$^3$ (Fig. 4b, 4d)."

Ln 265: Is it really the only reference for this fact?

There are others references, we will modify the sentence as follows:
"Picophytoplankton is often characterized by the presence of several taxa with potentially different effects on the population dynamics, whereas nanophytoplankton is mostly dominated by diatoms in the Mediterranean Sea **(Siokou-Frangou et al., 2010 ; Marty et al., 2002 ; Navarro et al., 2014 ; El Hourany et al., 2019)**."

Ln 269-274: Basically the same was said in the introduction.
Ln 275: Please, revise this sentence.

We will remove these sentences.

Ln 284: The fact that light and irradiance are essential for phytoplankton growth was known before 2001.

Here we referred to the fact that vertical velocities impact biogeochemistry, driving nutrients in the euphotic layers and the phytoplankton cells along the water column where these organisms will receive more or less light as a function of the depth (see our previous answer).

We agree that our sentence was not clear, so we will rework these sentences as follows:
"The early experimental works of Huisman, 1999 ; Jenkin et al., 1937 and Marshall et al., 1928 have well established that the light and nutrients are essential for phytoplankton growth. The availability of these two variables for the phytoplankton is driven by physical dynamics such as vertical velocities (Lévy et al., 2001 ; Pidcock et al., 2016 ; Mahadevan, 2016)."

Ln 285: Then is expected a higher nutrient concentration in the old AW? For that reason, there is a higher contribution of larger cells?

Yes, indeed we expect a higher nutrient concentration in the older AW since this water probably experiences more upwelling  and longer temporal vicinity of the coast along its route (Bethoux, 1989 ; Schroeder et al., 2010 ; see introduction). Moreover, it is known that a higher concentration of nutrients is favorable to larger cells. 
[revised manuscript text omitted]

**Conclusions and perspectives**

Ln 305-309: This is not a conclusion.

We used to begin the conclusion section with a short summary of the thematics. We completely reworked the Conclusion taking into account your comments and those of the other referee:

Phytoplankton structure and dynamics are a complex result of many interacting biological and physical phenomena. Finescale structures, and in particular fronts, generate vertical velocities which displace phytoplankton cells and nutrients in the water column, thus influencing phytoplankton communities. These mechanisms are only partially understood because the spatial scale of these structures and their ephemeral nature make them particularly difficult to study in situ; as a consequence only a few studies have been performed in finescale frontal regions. The estimates of specific growth rates for the various phytoplankton groups is one of the keys to better understand how environmental conditions affect phytoplankton dynamics. In this study, we followed the dynamics of several phytoplankton groups in two distinct water masses both in terms of hydrology and phytoplankton abundances, in order to explain their particular distribution.

The originality of our work resides in the fact that we used a size-structured population model applied in two water masses identified using a Lagrangian sampling strategy. To our knowledge this had never been done before. This strategy allowed us to reconstruct the diurnal cycle of several phytoplankton groups and to identify contrasted dynamics in the two water masses. For *Synechococcus* and nanophytoplankton, we found higher cell size in the older AW located north of the front, associated with lower abundances. A possible explanation is that the older AW is more enriched in nutrients than the younger AW, thus favoring larger cells. This remains a hypothesis because of a lack of nutrient data. Another novelty of our study is that we applied the Sosik et al. (2003) model on several phytoplankton groups identified by flow cytometry, whereas previous studies only applied it to *Synechococcus* and *Prochlorococcus* (Ribalet et al., 2010; Hunter-Cevera et al., 2014; Marrec et al., 2018; Fowler et al., 2020) or to certain types of diatoms (Dugenne et al., 2014). We obtained good results for *Synechococcus* and nanophytoplankton. However, our results were noisy for picophytoplankton groups probably because they contain several taxa with differing dynamics (Siokou-Frangou et al., 2010; Le Moal et al., 2011).

Our work paves the way for many research perspectives. Direct integration of growth rates in biogeochemical models (Cullen et al., 1993) should be taken into account for a better assessment of the biogeochemical contribution of phytoplankton in oligotrophic ecosystems and to better forecast its evolution in the context of global change. Furthermore, we plan future experiments again in the South Western Mediterranean in spring 2023, during the fast-sampling phase of the SWOT satellite mission which provides high resolution altimetry-derived currents. Involving high-resolution, high-precision nutrient measurements (necessary considering the oligotrophy of the Mediterranean Sea), coupled with DNA metabarcoding (to address phytoplankton biodiversity), zooplankton and virus sampling, we

will improve the understanding of zooplankton grazing and viral lysis on the different phytoplankton groups. Furthermore, we aim to explore how the biogeochemical and ecological role of finescale structures in regions of weak circulation differ from those documented in highly energetic regions like boundary currents. In the Mediterranean sea, the low nutrient content is indeed the perfect condition when addressing this question, because even weak horizontal or vertical nutrient redistributions associated with the finescale circulation are likely to result in a biological response (Talmy et al., 2014; Hashihama et al., 2021).

**Technical corrections**

Ln 6: Delete the space between "numerous" and ";".
Ln 21: Add a comma after (days-weeks).
Ln 41: Delete parenthesis after altimetry.
Ln 87: Once the front "is" localized.
Ln 103: 1164 samples "were" analyzed.
Ln 122: Maybe light availability is more adequate?
Ln 129: Please, consider changing investigated by counted.
Ln 134 and141: ... between the time interval t...
Ln 135-136: ... necessary to carry out photosynthesis?
Ln 157: Is there a typo? The probability of division is not denoted by $\gamma$?
Ln 163-164: You already defined those symbols; it is kind of redundant to do it again.
Ln 185: [chla]?
Ln 207: Please, consider using disregarding instead of eliminating.
Ln 227: Observed biovolume (observed and in situ are kind of repetitive), and predicted biovolume (check also Ln 219).
Ln 219-220: all species populations in both water masses?
Ln 222: No comma before the parenthesis.
Ln 221, 223, 224, 229, 230, 240: l or "bar l"?
Ln 239-240: The structure of the phytoplankton community.
Delete the point after the manuscript title and after the abstract.
Please, use 1 or 2 decimal numbers for all the variables measurements, to keep the format along the manuscript (e.g., Ln 46, 48, 221).
Please, use the same format for the dates along the manuscript (e.g., Ln 70 and 93).
Please, revise the use of the word indeed, it is repeated quite often throughout the text.

Thank you, we corrected that in the reworked manuscript.

**Figure 1.**
Panel a is very small, impossible to appreciate the information. Moreover, the colormap scale is minuscule and does not indicate the variable (and units) that represents.
In panel b, it would be interesting to indicate where the sampling events took place.
In panel c, in my opinion, the clock diagrams are not necessary.

Legend: The purple box encloses a (b) zoom of the sampling region with overlaid chlorophyll-a concentration (units). _______. The red line represents _____, the dark blue box _______, and the light blue box ________.
I am not an English native, but I think that the lines and boxes are superimposed to the chl map. The other way around will not allow you to see lines and boxes.

We have modified Figure 1 (see below). In panel a, we have increased the size of the map and colormap. We have also changed the color of the route of the ship for more visibility, and we added the units. In panel b, we have also increased the map. The sampling events took place all along the transects, that is why for us it is not necessary to indicate that in addition. Concerning panel c, we kindly disagree, because we think that the clock diagrams help the reader to understand the reconstruction of a day of 24 h period thanks to the 4 transects in each water mass.

[Figure]

Figure 1: (a) Route of the RV Beautemps-Beaupré during the PROTEVSMED-SWOT cruise. The purple box encloses a (b) zoom of the sampling region with overlaid chlorophyll-a concentration (µg L⁻¹) of 11 May 2018. In panel (b) black dotted line represents the route of the ship and the bold black line represents the route of the Lagrangian sampling across the older AW (delimited by the box in dark blue) and the younger AW (delimited by the box in light blue). (c) Dates of the transects across the older AW and the younger AW, used to reconstruct a day of 24 h period in each water mass.

**Figure 2.**

As previously indicated, I do not consider this figure of relevance to the main text.

We moved this figure in Appendices.

**Figure 2.**
Legend: Background colors indicate the two water masses…

We think that you speak about figure 3 ? In this case, indeed the background colors indicate the two water masses.

**Figures 4-6.**
Explain what represents the red line and the background color.
Correct all the color bars (by figure) to vary all in the same range.

We modified these figures following your comments. Furthermore, the y-axis is now in log and we added the mean of biovolume.

[Figure]

Figure 4: The background color represents the Synechococcus cell size distribution (i.e., biovolume in µm 3 ) observed (a, b) and predicted by the model (c, d) in the older AW (a, c) and in the younger AW (b, d) during 24 h. The black dots represent the mean of the biovolume ($\bar{v}_{obs}$ and $\bar{v}_{mod}$) and the red line represents the irradiance ($\mu E\ m^{-2}\ s^{-1}$).

[Figure]

Figure 5: Same as Fig. 4 for RNano.

[Figure]

Figure 6: Same as Fig. 4 for SNano.

**Figure 7.**

A very small figure, with some details difficult to appreciate. Even the legend is difficult to read.

We hope this new arrangement makes the figure clearer.

[Figure]

Figure 7: The contrasted distribution of phytoplankton in the frontal area. The circles represent the abundances of the several phytoplankton groups in the two water masses separated by the front. The boxes indicate the biovolume observed ($\bar{v}_{obs}$) and the growth rates ($\mu_{size}$) for each phytoplankton group, as estimated from the model. Figure adapted from Tzortzis et al. (2021).

**Table 2.**

Indicate also that there is information about the standard deviation.

Table 2. Means of biovolumes observed ($\bar{v}_{obs}$) and modelized ($\bar{v}_{mod}$) in $\mu m^3$, growth rates ($\mu_{size}$, $\mu_{ratio}$ in $d^{-1}$) and loss rate ($l$, in $d^{-1}$) for the phytoplankton groups, in the older and younger AW, as well as model fit parameters (see section 2.3).

| | *Synechococcus* | RNano | SNano |
|---|---|---|---|

| | | | |
|---|---|---|---|
| Older AW | $\bar{v}_{obs} = 0.38 \pm 0.04$
 $\bar{v}_{mod} = 0.38 \pm 0.02$
 $\mu_{size} = 0.24 \pm 0.91$
 $\mu_{ratio} = 0.59$
 $l = 0.36$

 $\sum(\theta) = 0.05$

 $corr(\bar{v}_{obs}, \bar{v}_{mod}) = 0.60$ | $\bar{v}_{obs} = 63.5 \pm 2.67$
 $\bar{v}_{mod} = 63.5 \pm 1.79$
 $\mu_{size} = 0.02 \pm 0.20$
 $\mu_{ratio} = 0.17$
 $l = 0.07$

 $\sum(\theta) = 0.139$

 $corr(\bar{v}_{obs}, \bar{v}_{mod}) = 0.46$ | $\bar{v}_{obs} = 85.0 \pm 1.98$
 $\bar{v}_{mod} = 84.7 \pm 1.38$
 $\mu_{size} = 0.04 \pm 0.26$
 $\mu_{ratio} = 0.11$
 $l = 0.11$

 $\sum(\theta) = 0.067$

 $corr(\bar{v}_{obs}, \bar{v}_{mod}) = -0.05$ |
| Younger AW | $\bar{v}_{obs} = 0.21 \pm 0.04$
 $\bar{v}_{mod} = 0.22 \pm 0.03$
 $\mu_{size} = 0.68 \pm 1.56$
 $\mu_{ratio} = 0.63$
 $l = 0.48$

 $\sum(\theta) = 0.153$

 $corr(\bar{v}_{obs}, \bar{v}_{mod}) = 0.65$ | $\bar{v}_{obs} = 61.2 \pm 5.23$
 $\bar{v}_{mod} = 60.6 \pm 2.17$
 $\mu_{size} = 0.04 \pm 0.28$
 $\mu_{ratio} = 0.33$
 $l = -0.12$

 $\sum(\theta) = 0.417$

 $corr(\bar{v}_{obs}, \bar{v}_{mod}) = 0.56$ | $\bar{v}_{obs} = 63.8 \pm 4.45$
 $\bar{v}_{mod} = 59.1 \pm 0.61$
 $\mu_{size} = 0.06 \pm 0.19$
 $\mu_{ratio} = 0.24$
 $l = 0.23$

 $\sum(\theta) = 0.247$

 $corr(\bar{v}_{obs}, \bar{v}_{mod}) = 0.15$ |

Define every variable on its own.
This will be done.

$\mu$ ratio should not be adimensional? The equation and its meaning are already defined in the text.
As described in the new version of the methods (see above) $\mu_{ratio}$ represents a minimum estimate of the daily growth rate, that would be observed if cells synchronously only grew from the time $\bar{v}_{min}$ is observed (typically dawn) to the time $\bar{v}_{max}$ is observed (typically dusk), and only divided while $\bar{v}$ decreases. Since the model allows for any cell to grow, divide or be at equilibrium over the entire integration period (asynchronous populations), $\mu$ size is expected to be higher than $\mu_{ratio}$. In practice, $\mu_{ratio}$ is sensitive to noise in the data and is only provided here as an alternative estimate of the growth rate that does not rely on the model.

Define the acronym PFG.
We mentioned in the manuscript (Ln 111) that PFG(s) means "phytoplankton functional groups".


Dear referee,

Thank you very much for your constructive comments and suggestions, as well as your corrections of the grammar and spelling in order to improve the quality of the manuscript. According to the Biogeosciences guidelines, we reworked the manuscript following your suggestions and those of the other referee. Previously we addressed the main points you raised. Hereafter, we provide an update of our previous response: Your comments are in black and our previous answers in blue, we add revised versions of each section in green.

**General Comments**

Fine-scale physical processes affect the community structures and productivity of marine plankton at various time scales, but the study to explore them are relatively limited due to technical difficulties. This study aims at untangling this problem by applying the combination of a semi-Lagrangian survey, semi-continuous sampling, and biomathematical models. This approach is novel, and the results obtained from the field survey in the Mediterranean Sea seem to be reasonable. I believe that their approach may open the door to the elucidation of complex physical processes that affect marine microbial ecology, though there are still some problems to be answered.

We thank you for this encouraging comment on our scientific approach.

The first problem is that the objective of the present study (this article) is ambiguous. I understand the final goal of their study, but the results obtained this time are too primitive for that. Based on the results obtained, the authors should reconstruct the objective(s) of the "present" study. The authors should be clear about whether this manuscript concentrated on the development of a new method or aimed to elucidate the effects of the frontal structure observed in the South Mediterranean Sea on phytoplankton structures to some degree. In addition to Introduction and Abstract, the title of the article possibly should be changed in association with that.

We acknowledge that we did not sufficiently detail our scientific questions. Our present study is closely related to our previous work (Tzortzis et al., 2021) on the description of the physical characteristics of a **frontal finescale structure** and its effects on the distribution of phytoplankton abundances, in the south part of the Balearic Islands (Mediterranean Sea). The importance of the frontal area studied in the present article is due to the fact that most of the in-situ studies related to the physical-biological coupling at finescale have focused on extreme situations occurring in boundary currents, where intense fronts and dramatic contrasts in water properties are met but are not representative of the global ocean. Indeed, on the contrary, vast oceanic regions are dominated by weak fronts continuously created, moved

and dissipated, and which separate different water masses with similar properties. Very few studies exist in these regions due to the difficulty of performing in situ experiments over these short-lived and small features. In our previous work Tzortzis et al. (2021) we showed that the fine-scale front observed during our cruise in 2018 in an oligotrophic region as the SW Mediterranean maintains the driving role on phytoplankton diversity also in these moderately energetic regions. Nevertheless, this first study did not explain this particular distribution of phytoplankton, and open questions remain: **Is it exclusively driven by the dynamics of the front ? Or, do biological processes also play a role ?**

In the present study, we attempt to explain the patterns of phytoplankton abundances by focusing on the cellular dynamic of these organisms using the size-structured population model developed by Sosik et al. (2003). This is possible thanks to the analysis of phytoplanktonic cells at the single cell level.

We agree that we did not develop enough the potential effect of the frontal dynamics on the structure of the phytoplankton community, especially in the Discussion. We will rework this part, detailing more in depth the potential effect of vertical velocities and water masses properties (temperature, salinity, nutrients) on the phytoplankton communities.

Furthermore, we did not sufficiently discuss the novelty of our methodology, and its implication for future oceanographic cruises. Although some studies have already used the size-structured population model (Sosik et al., 2003 ; Ribalet et al., 2010, Dugenne et al., 2014 ; Marrec et al., 2018) or other models (Geider et al., 1997 ; MacIntyre et al., 2000) to compute phytoplankton growth rates, the novelty of our study is its application in a context of a Lagrangian sampling strategy. Moreover, we applied this model on several phytoplankton groups (not only *Synechococcus* or *Prochlorococcus*, like most studies). To our knowledge, this has not been done before (except the study of Dugenne et al. (2014) which applied it to specific diatoms). We think our methodology applied here, paves the way for future studies.

The second problem is about the robustness and significance of the estimates of growth and loss rates. When we compare two or more values, the intervals of confidence or possible standard errors are indispensable. However, in the present manuscript, there are no remarks on that. If possible, please add the statistical information.

We agree that it is important to add intervals of confidence; in the previous literature this is not done, probably due to the difficulties in calculating them. In any case, as suggested also by the other referee, we have completely reworked the section concerning the methodology of the size-structured population model (section 2.3 in the manuscript). We provide this part further in the present document, in the case that you cannot consult our answer for the other referee. We hope that this new version clarifies the principle of this model. Indeed, statistics are already included in the model: the growth rates were estimated using the maximum likelihood function and 200 iterations were run to estimate the standard deviation of group-specific growth rates using a Markov Chain Monte Carlo (Geyer, 1992 ; Neal, 1993).

English grammatical errors are relatively frequent in this manuscript. The authors should

have it checked by a native speaker or some editorial service. For example, "Numerical simulation have shown" (L3), "Since several years" (L5), and "a precious information" (L7).

Thank you, we will rework the manuscrit with the help of a new co-author English speaker.

These are general comments on this manuscript. The followings are minor specific comments.

**Specific Comments**

**Introduction**

**Introduction (see Ln 15-78 in the reworked manuscript)**

[revised manuscript text omitted]

**Material and Methods**

L71 "satellite SWOT will be launched" is correct.

Sorry for the mistake. The SWOT satellite is now in orbit ! It was launched on December 16[th] 2022. We will modify this section following also the suggestions of the other referee.

L74 What do the authors mean by "moderate energy"? Which energy? And in which way is it important in the selection of the present study site?

Here, we mean that Mediterranean frontal structures are often less intense than those found in boundary currents such as the Kuroshio, that are able to generate vertical velocities in order of 30 m day$^{-1}$ (Clayton et al., 2014). By contrast, vertical velocities in the Mediterranean sea are in the order of 8 m day$^{-1}$ (Barceló-Llull et al., 2021, Tzortzis et al., 2021).

Below is a map of the surface eddy kinetic energy by Pascual et al. (2006), where the contrast between the Mediterranean sea and the western boundary currents is evident.
We will add this reference and explanation in the new version of the Methods section.

[Figure]

Figure: Eddy kinetic energy (EKE) estimated with 4 altimetric missions (Jason-1 + T/Pinterlaced + ERS-2/ENVISAT + GFO). Units are cm$^2$ s$^{-2}$ . Figure extracted from Pascual et al. (2006).

**The Sampling strategy (Ln 81-101 in the reworked manuscript)**

The PROTEVSMED-SWOT cruise, dedicated to the study of finescale dynamics, was conducted in the south of the Balearic Islands between April 30th and May 18th 2018, on board the R/V Beautemps-Beaupré (Fig. 1a). This cruise followed an adaptive Lagrangian strategy to measure at high spatial and temporal resolution several physical and biological variables with both in situ sensors and analysis of the sea surface water intake. The vessel route was designed ad-hoc on the basis of daily remote sensing dataset provided by the Software Package for an Adaptive Satellite-based Sampling for Oceanographic cruises (SPASSO, https://spasso.mio.osupytheas.fr, last access: April 22, 2023). SPASSO used altimetry-derived currents from the Mediterranean regional product (nrt_med_allsat_phy_l4) AVISO ("Archiving, Validation and Interpretation of Satellite Oceanographic", https://www.aviso.altimetry.fr, last access: April 22, 2023) and ocean color observations. Chlorophyll a concentrations ([chla], level 3, 1 km resolution, MODISAqua and NPPVIIRS sensors combined (after May 27, 2017) into a new product called MULTI) were provided by CMEMS, "Copernicus Marine Environment Monitoring Service", https://marine.copernicus.eu, last access: April 22, 2023. In addition, CLS provided the surface Chl concentration composite products, with the support of the CNES. They were constructed using a simple weighted average over the previous 5 days of data gathered by the Suomi/NPP/VIIRS sensor. SPASSO generated maps of dynamical and biogeochemical structures in both near real time (NRT) and delayed time (DT). Maps of [chla] allowed us to identify two water masses, characterized by distinct [chla] values and separated by a zonal front at around 38° 30' N. This front was also detected using in situ horizontal velocities,

temperature and salinity, as described in Tzortzis et al. (2021). These two water masses were sampled along a designated route of the ship, represented in black in Fig. 1b. Special attention was paid to adapting the temporal sampling in order to measure the phytoplankton diel cycle in each water mass. This was achieved by continuously sampling across both water masses along transects. While the ship did not remain in each water mass for 24h, day-to-day variability remained low and measurements from several days were combined into one diel cycle (Fig. 1c). The shape depicted by the ship's track led us to call these areas north–south (NS) hippodrome (bold black line in Fig. 1b) performed between 11 May and 13 May 2018.

L91 "have been measured" should be "were measured".

Thank you, we will modify it when we rework the manuscrit.

L109 Was the categorization of phytoplankton populations (functional groups) on cytograms made manually on a somewhat arbitrary criterion or semi-automatedly using something like machine learning? How do the authors guarantee the validity and consistency of the categorization?

We have identified several groups of phytoplankton by flow cytometry. This categorization of phytoplankton groups on 2 dimensional plots (cytograms) was made **manually using the conventional criterion determined by flow cytometrists**. Phytoplankton groups were resolved on the basis of their light scatter (namely forward scatter FWS and sideward scatter SWS) and fluorescence (red FLR and orange FLO fluorescence ranges) properties. For instance, *Synechococcus* was unambiguously put in evidence thanks to its higher FLO intensity induced by the presence of phycoerythrin pigments. The optical characteristics of each phytoplankton cluster (or group) are provided in the literature (Dubelaar and Jonker, 2000 ; Reynolds, 2006 ; Thyssen et al., 2008 ; Edwards et al., 2015). We based our categorization on these previous studies. We identified typical phytoplankton groups of the Mediterranean Sea already found by previous works using flow cytometry (Thyssen et al., 2014 ; Marrec et al., 2018). Most of the publications using flow cytometry data to study planktonic cells perform the same way, and rely entirely on the literature and the expertise of the flow cytometrists.
Application of Artificial Intelligence as machine learning to cytometry data is currently under development in our laboratory and the recent work of Fuchs et al. (2022) provided promising results. Unfortunately, this approach is not yet mature enough, which is why we do not use it here.

L114 Show us the time and space (cruise length) ranges that a single sample covers.

We will provide a detailed information modifying the sentence as follow:
"The flow cytometer was connected to sea surface continuous flow, through the system of the thermosalinograph (TSG), (depth: 3 m). The flow cytometer sampled the seawater in a dedicated small container called "subsampler", that isolates the seawater during its analysis.

Between two consecutive samples the subsampler was flushed continuously by the seawater circuit of the ship in order to clean and renew the seawater. The subsampler isolated the seawater every 30 min, and two distinct protocols (FLR6 and FLR25) were run sequentially: for FLR6 about 1.3 mL were analyzed in 420 s and for FLR25 about 4 mL were analyzed in 600 s. The use of the subsampler to isolate the volume of seawater subsampled by the flow cytometer allowed us to ignore the movement of the ship, while the flow cytometer performed its analysis. This way the volume analyzed corresponds to a point location rather than a volume spread on the ~2 km covered by the ship in 30 min".

**In situ measurements ( Ln 103-131 in the reworked manuscript)**

[revised manuscript text omitted]

$v = exp(\beta_0) \times FWS^{\beta_1}$      with in our case $\beta_1 = 0.9228$ and $\beta_0 = -5.8702$     (Eq. 1)

In the size-structured population model, cells are classified into several size classes according to their dimensions at time t. Classes are logarithmically spaced as follows: for $i$ in 1,2,...,$m$ $v_i = v_1 2^{(i-1)\Delta v}$ where $\Delta v$ is constant and chosen to ensure that size classes cover the entire observed biovolume $v$, from $v_1$ to $v_m$ (see figure below). For Synechococcus, $\Delta v = 1/6$ with $\Delta v$ constant and $m = 40$, so that the model size classes encompassed our full measured size distributions (0.0279-2.5209 $\mu m$).

[Figure]

Figure : Cell cycle stages in the size-structured population model. Cells may grow to the next size class ($\gamma$) or be at equilibrium $(1 - \gamma(t))(1 - \delta(v, t))$. Above a particular size, cells are large enough to divide in two daughter cells with probability ($\delta$). Figure adapted from Sosik et al. (2003).

At any time $t$, the number of cells in size classes **N** (and **w** its corresponding normalized distribution), was projected to $t + dt$ via matrix multiplication (Eq. 2):

$$N(t + dt) = A(t)\, N(t) \qquad \text{and} \qquad w(t + dt) = \frac{A(t)\, N(t)}{\sum A(t)\, N(t)} \qquad \text{(Eq. 2)}$$

We chose $dt = 10$ min (i.e., $\frac{10}{60}$ h) as Sosik et al. (2003) and Dugenne et al. (2014), because for this time step, cells are unlikely to grow more than one size class.

$A(t)$ is a tridiagonal transition matrix that contains:
1) $\gamma$: the probability of cellular growth
2) $\delta$: the probability of cells entering mitosis
3) the cells stasis, i.e., the probability for cells to maintain their state (i.e size) in equilibrium during the temporal projection.

Probability of cellular growth

The probability of cells growing to the next size class ($\gamma$) depends only on the light intensity (irradiance) necessary for photosynthesis, expressed as (Eq. 3):

$$\gamma(t) = \gamma_{max} \left[ 1 - exp\left(- \frac{E(t)}{E^*}\right)\right] \qquad \text{(Eq. 3)}$$

$\gamma_{max}$: maximum proportion of cells growing (dimensionless quantity)
E: irradiance ($\mu$E m$^{-2}$ s$^{-1}$)
E*: irradiance normalizing constant ($\mu$E m$^{-2}$ s$^{-1}$)

Probability of cells entering mitosis

According to Dugenne et al. (2014), $\delta$ expresses a proportion (between 0 and 1) modeled by the combination of two Normal distributions ($\mathcal{N}$). One is linked to the cell size, the other is linked to the time of cell division. Both imply an optimum, reached at $\overline{v}$ and $\overline{t}$ respectively, for cell division above which the cell size and the timing of division is suboptimal (Eq. 4).

$$\delta(t, v) = \delta_{max} \, \mathcal{N}(\overline{v}, \sigma_v^2) \, \mathcal{N}(\overline{t}, \sigma_t^2) \qquad \text{(Eq. 4)}$$

$\delta_{max}$: maximum proportion of cells entering mitosis (dimensionless quantity)

$\overline{v}$: mean of the size Normal distribution ($\mu$m$^3$)

$\sigma_v$: standard deviation of the size Normal distribution ($\mu$m$^3$)

$\overline{t}$: mean of the time Normal distribution (h)

$\sigma_t$: standard deviation of the time Normal distribution (h)

Cells stasis

A third functional proportion is included in the transition matrix *A(t)*, to represent cell stasis. Since this function illustrates a non-transition, it is modeled by the proportion of cells that neither divided nor grew between *t* and *t + dt*.

$$[1 - \gamma(t)] \, [1 - \delta(t, v)]$$

Optimal parameters

The set of parameters, $\theta$ is estimated by maximum likelihood function, assuming errors between observed *w* and predicted $\widehat{w}$ normalized size distributions (Eq. 6, 7, 8). Their standard deviations are estimated by a Markov Chain Monte Carlo approach (Geyer, 1992 ;

Neal, 1993) that sample $\theta$ from their prior density distribution, obtained after running 200 optimizations on bootstrapped residuals to approximate the parameter posterior distribution using the normal likelihood. (The likelihood function represents the probability of random variable realizations conditional on particular values of the statistical parameters).

$$\theta = [\gamma_{max}, E^*, \delta_{max}, \bar{v}, \sigma_v, \bar{t}, \sigma_t] = argmin(\textstyle\sum(\theta)) \qquad \text{(Eq. 6)}$$

$$\sum(\theta) = \sum_{t}^{t+dt} \sum_{i=1}^{m} (w(t) - \widehat{w}(t,\theta))^2 \qquad \text{(Eq. 7)}$$

$$\widehat{N}(t,\theta) = A(t\text{-}dt, \theta) \, N(t\text{-}dt) \qquad \text{(Eq. 8)}$$

$\widehat{w}$ is computed from $\widehat{N}$ following Eq. 2. The fit of the model is quantified using two numbers:

the loss rate ($\sum(\theta)$, lower indicates better fit), and the correlation between the observed and

modeled mean biovolumes $\bar{v}_{obs}$ and $\bar{v}_{mod}$ over the diel cycle (corr($\bar{v}_{obs}$, $\bar{v}_{mod}$), higher indicates better fit). Table 1 provides the model parameters being optimized.

Table 1: Model parameters being optimized.

| Parameters | Definition | Interval | Units |
|---|---|---|---|
| $\gamma_{max}$ | Max proportions of cells in growing phase | $[0, 1]$ | $\emptyset$ |
| $E^*$ | Irradiance normalizing constant | $[0,\infty[$ | $\mu E\ m^{-2}\ s^{-1}$ |
| $\delta_{max}$ | Max proportion of cells entering mitosis | $[0,1]$ | $\emptyset$ |
| $\bar{v}$ | Mean of the size Normal distribution | $[v_{min}, v_{max}]$ | $\mu m^3$ |
| $\sigma_v$ | Standard deviation of the size Normal distribution | $[10^{-06},\infty[$ | $\mu m^3$ |
| $\bar{t}$ | Mean of the time Normal distribution | $[1, 24\frac{1}{dt} + 1]$ | hours |

| $\sigma_t$ | Standard deviation of the time Normal distribution | $[10^{-06}, \infty[$ | hours |
|---|---|---|---|

Growth rate and loss rate

Once optimal parameters are identified, the model estimates a population intrinsic growth rate $\mu_{size}$ , and a specific loss rate $l$, integrated over a 24 h period. The method uses the fact that the observed size distribution **N** is the result of both growth and loss processes, while the time projection of the initial size distribution **N(0)** using the model, $\hat{\textbf{N}}$, is only the result of growth processes. The growth rate is calculated at each time step following Eq. 9, and integrated over 24 h. 200 iterations by a Markov Chain Monte Carlo were run to estimate the standard deviation of group-specific growth rates.

$$\mu_{size} = \frac{1}{dt} ln(\frac{\sum_{i=1}^{m} \widehat{N}_i(t+dt)}{\sum_{i=1}^{m} \widehat{N}_i(t)})$$

$i$: i$^{th}$ size class

$\widehat{N}$: predicted size distribution (cells cm$^{-3}$)

$m$: number of size classes

$dt$: time step (h)

$\mu_{size}$: growth rates (day$^{-1}$)

An independent growth rate estimation was obtained as $\mu_{ratio} = ln(\bar{v}_{max} / \bar{v}_{min})$ where $\bar{v}_{min}$ and $\bar{v}_{max}$ are the minimum and maximum of the mean observed biovolume $\bar{v}_{obs}$ over the diel cycle (Marrec et al., 2018). $\mu_{ratio}$ represents a minimum estimate of the daily growth rate, that would be observed if cells synchronously only grew from the time $\bar{v}_{min}$ is observed (typically dawn) to the time $\bar{v}_{max}$ is observed (typically dusk), and only divided while $\bar{v}$ decreases. Since the model allows for any cell to grow, divide or be at equilibrium over the entire integration period (asynchronous populations), μ size is expected to be higher than $\mu_{ratio}$. In practice, $\mu_{ratio}$ is sensitive to noise in the data and is only provided here as an alternative estimate of the growth rate that does not rely on the model.

The population loss rate $l$ is obtained by difference between the intrinsic growth rate $\mu_{size}(t)$ and the temporal change in logarithmic observed size distribution **N**, which represents the net growth rate $r(t) = \mu_{size}(t) - l(t)$ so that:

$$\bar{l} = \int\limits^{t} \mu_{size}(dt) \ - \ \frac{1}{dt} \ ln(\frac{N(t+dt)}{N(t)})$$

**Results**

L183 What do the authors mean by "put in evidence"?

We will change it by "identify": "Four eukaryotic picophytoplankton groups **were identified**."

L197 "A similar distribution is observed" should be "A similar distribution was observed". Most of the sentences in this paragraph should be rewritten to past tense.

Thank you, we will carefully check verb tenses in the reworked manuscript.

**Spatio-temporal distribution of phytoplankton abundances in the two water masses (Ln 234-246 in the reworked manuscript)**

The sampling strategy adopted during PROTEVSMED-SWOT enabled us to sample two water masses with different properties. The map of the satellited-derived surface [chla] shows higher concentration in the Northern part of the sampling route, corresponding to older AW, than in the the Southern part, corresponding to younger AW (Fig. 1b). Figure 3 shows the properties of the sea surface water as a function of time (from 11 May 00:00 to 13 May 12:00 UTC) along the sampling route. The older AW is characterized by a colder temperature and higher values of salinity than the younger AW. Figure 3 also displays the abundances of each phytoplankton group over these two water masses. *Synechococcus* and Pico2 are the most abundant. They present a clear surface distribution pattern, with high abundances in the warm and low salinity water, corresponding to the young AW. A similar distribution is observed for Pico1, Pico3 and RNano but with lower abundances than *Synechococcus* and Pico2. The abundances of SNano, PicoHFLR and Cryptophyte show less contrasts along the cruise than the previous groups, nonetheless the highest abundances can be distinguished in the younger AW, in particular in the second and third passage (transect) across this water mass. Finally, microphytoplankton is the less abundant group, but it clearly shows a contrast between the two water masses, opposite to the one of the other phytoplankton groups.

L204 "In addition to the cell abundances measured along the route of the ship, the phytoplankton diurnal cycle in the two water masses was also reconstructed" This sentence means that the cell abundances were reconstructed first. But, of course, they were not "reconstructed". Rewrite.

We will rework this sentence as follows: "The phytoplankton diurnal cycle in the two water masses was reconstructed [...]."

L205 "each water mass" should be "either water mass"?

Thank you, we will modify it when we rework the manuscrit.

L207 "This adaptive Lagrangian approach allows sampling of the different functional groups of phytoplankton in each water mass" Different functional groups of phytoplankton in different water masses can be sampled using another approach. I think that this is not the benefit of the adaptive Lagrangian approach. Explain it more appropriately.

We are sorry for the lack of clarity. Following the suggestions of the other referee, in the reworked manuscript we will move this part from the Discussion to the Methods section.

L217 "Furthermore, the comparison between the biovolume observed in situ and the biovolume predicted by the model is sound and confirms that the model-predicted cell size distributions well recapitulated the diurnal cycle reflecting either growth or cell division." Could the authors show any data or figure to support this?

We apologize for the lack of clarity. This sentence refers to Figure 4 in the manuscript. In order to also take into account the suggestion of the other referee, we will modify the text as follows:
"For *Synechococcus*, in the older AW the observed size distribution (i.e., observed biovolume) is similar to the prediction of the model (i.e., predicted biovolume). Both display a day-long large size-class distribution centered approximately on 0.3 μm$^3$. In the younger AW (Fig. 4a, 4c) the distributions of biovolume observed and predicted are narrower than in the older AW and centered approximately on 0.2 μm$^3$ (Fig. 4b, 4d)."

L223 As mentioned in my General Comments, I request the authors to show the interval of confidence or something that can evaluate the robustness of the estimates presented by the present method. This will enable us to compare the values of different phytoplankton groups and water masses on a statistical basis. I can find something like that in Table 2, but I fail to see what it means. When the authors consider the interval, is it significant to discuss the "difference" between the two water masses?

In the manuscript, the standard deviation of the growth rates is indicated in Table 2. As previously mentioned, we completely reworked the methodological section concerning the size-structured population model, following the comments of the other referee.

Table 2. Means of biovolumes observed ($\bar{v}_{obs}$) and modelized ($\bar{v}_{mod}$) in μm$^3$, growth rates ($\mu_{size}$, $\mu_{ratio}$ in d$^{-1}$) and loss rate ($l$, in d$^{-1}$) for the phytoplankton groups, in the older and younger AW, as well as model fit parameters (see section 2.3).

| | *Synechococcus* | RNano | SNano |
|---|---|---|---|
| Older AW | $\bar{v}_{obs} = 0.38 \pm 0.04$ | $\bar{v}_{obs} = 63.5 \pm 2.67$ | $\bar{v}_{obs} = 85.0 \pm 1.98$ |

| | | | |
|---|---|---|---|
| | $\bar{v}_{mod} = 0.38 \pm 0.02$
 $\mu_{size} = 0.24 \pm 0.91$
 $\mu_{ratio} = 0.59$
 $l = 0.36$

 $\sum(\theta) = 0.05$

 $\text{corr}(\bar{v}_{obs}, \bar{v}_{mod}) =$
 0.60 | $\bar{v}_{mod} = 63.5 \pm 1.79$
 $\mu_{size} = 0.02 \pm 0.20$
 $\mu_{ratio} = 0.17$
 $l = 0.07$

 $\sum(\theta) = 0.139$

 $\text{corr}(\bar{v}_{obs}, \bar{v}_{mod}) =$
 0.46 | $\bar{v}_{mod} = 84.7 \pm 1.38$
 $\mu_{size} = 0.04 \pm 0.26$
 $\mu_{ratio} = 0.11$
 $l = 0.11$

 $\sum(\theta) = 0.067$

 $\text{corr}(\bar{v}_{obs}, \bar{v}_{mod}) =$
 -0.05 |
| Younger AW | $\bar{v}_{obs} = 0.21 \pm 0.04$
 $\bar{v}_{mod} = 0.22 \pm 0.03$
 $\mu_{size} = 0.68 \pm 1.56$
 $\mu_{ratio} = 0.63$
 $l = 0.48$

 $\sum(\theta) = 0.153$

 $\text{corr}(\bar{v}_{obs}, \bar{v}_{mod}) =$
 0.65 | $\bar{v}_{obs} = 61.2 \pm 5.23$
 $\bar{v}_{mod} = 60.6 \pm 2.17$
 $\mu_{size} = 0.04 \pm 0.28$
 $\mu_{ratio} = 0.33$
 $l = -0.12$

 $\sum(\theta) = 0.417$

 $\text{corr}(\bar{v}_{obs}, \bar{v}_{mod}) =$
 0.56 | $\bar{v}_{obs} = 63.8 \pm 4.45$
 $\bar{v}_{mod} = 59.1 \pm 0.61$
 $\mu_{size} = 0.06 \pm 0.19$
 $\mu_{ratio} = 0.24$
 $l = 0.23$

 $\sum(\theta) = 0.247$

 $\text{corr}(\bar{v}_{obs}, \bar{v}_{mod}) =$
 0.15 |

L223 What do the authors mean by a negative loss rate? I think that it should be shown as a positive value if the loss term is significant.

The model estimates a population intrinsic growth rate, $\mu_{size}$, and a specific loss rate, $l(t)$, over a 24 h period. The loss term includes both biological losses (grazing or death, always negative) but also physical losses (e.g., advection, which can be positive or negative, see Sosik et al., 2003).

L224 "a low division rate" should be "a low loss rate"?

Indeed it is "a low loss rate".

L225 We are not able to judge whether the difference is "significant", without an appropriate statistical figure. Did the authors conduct a statistical test? In which way? What was the level of significance?

We did not conduct a statistical test, but in the reworked manuscript we plan to measure the "fit" of the model, i.e. compare observed vs predicted cell distributions (e.g. Fig. 4a vs 4c). The idea is to recover the error used to define the "best fit" and the best parameters (i.e. error of the selected model).

**Phytoplankton cellular growth and division in the two water masses (Ln 247-272 in the reworked manuscript)**

The phytoplankton diurnal cycle was reconstructed in the two water masses using the size-structured population model originally developed by Sosik et al. (2003). Figures 4, 5, 6 represent the phytoplankton size distribution (i.e., biovolume) observed in situ and predicted by the model over 24 h for *Synechococcus*, RNano and SNano, respectively. From the predicted biovolume it is possible to derive specific growth ($\mu_{size}$ ) and a loss (*l*) rates, summarized in Table 2 for the different phytoplankton groups in the two water masses, along with metrics of model performance. We also attempted to model the diurnal cycle for the picophytoplankton groups, i.e., Pico1, Pico2, Pico3, and PicoHFLR. However, their very noisy size distributions prevented us from obtaining reliable growth rate estimates. Similarly, microphytoplankton and Cryptophytes were not abundant enough to allow a reliable determination of their abundances and cell cycles. These cytometric groups are thus not considered further in this study.

For *Synechococcus*, in the older AW the prediction of the model (i.e., predicted biovolume) is similar to the observed size distribution (i.e., observed biovolume). Both display a day-long large size-class distribution centered approximately on 0.3 $\mu m^3$. In the younger AW (Fig. 4a, c) the distributions of observed and predicted biovolume are narrower than in the older AW and centered approximately on 0.2 $\mu m^3$ (Fig. 4b, d). As a consequence, the older AW is populated by larger cells of *Synechococcus* (mean observed biovolume v̄ obs = 0.38 ± 0.04 $\mu m^3$ ) than in the younger AW (mean biovolume $\bar{v}_{obs}$ = 0.21 ± 0.04 $\mu m^3$ ) (Table 2). Growth and loss rates also differ between the two water masses. In the older AW, the large cells of *Synechococcus* have a growth rate $\mu_{size}$ = 0.24 ± 0.91 d$^{-1}$ and a loss rate *l* = 0.36 d −1 , whereas in younger AW the smaller cells are characterized by higher growth ($\mu_{size}$ = 0.68 ± 1.56 d$^{-1}$) and loss (*l* = 0.48 d$^{-1}$) rates.

Relative to *Synechococcus*, cell size distribution and growth and loss rates are less contrasted between the older and younger AW for SNano (Fig. 6) and even more so RNano (Fig. 5). The mean observed RNano biovolumes are similar in the older and younger AW (63.5 ± 2.67 $\mu m^3$ and 61.2 ± 5.23 $\mu m^3$ , respectively) (Table 2). For SNano, similar to *Synechococcus*, the older AW is predominantly composed of larger cells ($\bar{v}_{obs}$ = 85.0 ± 1.98 $\mu m^3$ ) than in the younger AW ($\bar{v}_{obs}$ = 63.8 ± 4.45 $\mu m^3$ ). For both Nano groups, growh rates are generally very low in both water masses ($\mu_{size}$ < 0.1 d$^{-1}$ ). Loss rates are higher than growth rates, except for RNano in the younger AW (negative loss rate implying an external input of cells such as by advection). However, the corresponding optimization factor is the highest observed across the 6 modelisations, indicating this result is subject to caution.

**Discussion**

L244 "largest cells of *Synechococcus* are dominant" This sounds unnatural. "large cells" or "larger cells" may sound more natural.

Thank you for your English corrections.

L245 "This is due to the fact that the older AW is composed of *Synechococcus* cells transiting in all the cell cycle stages all day long". That the older AW is composed of *Synechococcus* cells transiting in all the cell cycle stages all day long is not a "fact", but a suggestion or speculation derived from the present observation. The authors should be more careful about it.

The sentence will be reformulated as follows: "The model results suggest that in the older AW *Synechococcus* cells transit in all the cell cycle stages all day long."

L250 "The patchiness of a distribution" laterally means how frequently "patches" are observed in that distribution. It does not mean how dispersed it is over a wide range. This misunderstanding may be critical in this discussion.

We apologize for the lack of clarity. We will remove this sentence, as we don't think this information is relevant because it is kind of repetitive with what we wrote before, and modify this part also taking into account the comments of the other referee:
"The model results suggest that in the older AW *Synechococcus* cells transit in all the cell cycle stages all day long. Furthermore, in the older AW the cells display a day-long large size-class distribution centered approximately on 0.3 $\mu m^3$ while in the younger AW (Fig. 4a, 4c) the distribution is narrower and centered approximately on 0.2 $\mu m^3$ (Fig. 4b, 4d)."

L257 Avoid using any contraction (including "couldn't") in academic writing.

Thank you for your English correction.

L258 What is an "important biodiversity"? I believe that biodiversity is always important.

We apologize for the misuse of the adjective "important". Of course, we do agree on the importance of biodiversity! We will modify the sentence as follows:
"Picophytoplankton is often characterized by the presence of several taxa with potentially different effects on the population dynamics, whereas nanophytoplankton is mostly dominated by diatoms in the Mediterranean Sea (Siokou-Frangou et al., 2010 ; Marty et al., 2002 ; Navarro et al., 2014 ; El Hourany et al., 2019). "

L261 Does this mean that the authors should have conducted molecular analysis (e.g. metabarcoding) to elucidate which taxonomic group each flow cytometric population is composed of? Although it requires flow sorting before analysis, is it a possible future plan? Anyway, the authors mention "this hypothesis" here, but I could not find any hypothesis to be tested from this paragraph. Please reconsider the issues to be discussed here and rearrange this paragraph.

As mentioned in the methodology section, several phytoplankton groups were identified by flow cytometry. This analysis allowed us to detect various groups of eukaryotic nanophytoplankton (RNano and SNano) and eukaryotic picophytoplankton (Pico1, Pico2, Pico3, PicoHFLR). *Synechococcus* is a prokaryotic picophytoplankton, but we have made a distinction between the picophytoplankton group and *Synechococcus* group because this latter was unambiguously resolved by flow cytometry thanks to its higher FLO intensity induced by the presence of phycoerythrin pigments. Idem for Cryptophytes which also have a peculiar and unambiguous optical signature.

Unfortunately, we did not conduct molecular analysis, that is why we are not able to identify the taxa contained in pico- and nanophytoplankton groups. In our future cruise (spring 2023), we plan to use metabarcoding and metagenomic analysis to address the biodiversity of phytoplankton. We will also perform zooplankton and virus sampling to understand the effect of zooplankton grazing and viral lysis on the different phytoplankton groups.

Following the suggestions of the other referee, we will move the sentences (L 256 - 258) in section 3.3. In the reworked manuscript, we will focus on the explanation of why the size distribution of picophytoplankton is noisy whereas we obtained a clear pattern for nanophytoplankton.

**The phytoplankton diurnal cycle (Ln 274-298 in the reworked manuscript)**

Although it has been clearly demonstrated that phytoplankton plays a fundamental role in the ocean ecosystem functioning (Watson et al., 1991; Field et al., 1998; Allen et al., 2005), numerous questions remain about their population dynamics in relation with finescale structures.

Coupling high-resolution in-situ flow cytometry measurements in two contrasted water masses with the size-structured population model developed by Sosik et al. (2003) allowed us to characterize the structure of phytoplankton and to reconstruct its diel cycle of cell growth and division on both sides of a finescale front. The growth and loss rates ($\mu_{size}$ and $l$) found for *Synechococcus* are of the same order of magnitude as those obtained by Marrec et al. (2018) in the northwestern Mediterranean Sea using the same method. In section 3.2, we showed that the largest cells of *Synechococcus* were found in the older AW. These *Synechococcus* cells are characterized by a larger range of biovolume and lower growth and loss rates than those located in the younger AW (Table 2). The cells are in average larger than in the younger AW as they grow slower at the population scale and divide less. Conversely, in the younger AW the distribution of the *Synechococcus* biovolume is narrower, which could be explained by cells being more active, more homogeneous in terms of size (biovolume) and better synchronized, leading to a smaller spread of the cell biovolume (Fig. 4b,d) with a dominance of small *Synechococcus* cells (Fig. 3). This also explains why higher abundances of *Synechococcus* are found in the younger AW (Fig. 3). Interestingly, the resulting net growth rate (growth minus loss) is negative in the older AW, positive in the younger AW.

Results are more difficult to interpret for the nanoplankton groups RNano and SNano, expected to be mostly dominated by diatoms in the Mediterranean Sea (Marty et al., 2002;

Siokou-Frangou et al., 2010; Navarro et al., 2014; El Hourany et al., 2019), especially in frontal systems (Claustre et al., 1994). RNano and SNano diel cycles are not as well-defined as for *Synechococcus*, leading to very small estimates of growth rates by the model. Optimization factors (linked to the mean squared difference between observed and predicted normalized size distributions) are relatively high and/or temporal correlations between observed and predicted mean biovolume relatively low, indicating these results must be considered with caution. Nevertheless, results suggest much lower growth and loss rates for nanoplankton than for *Synechococcus* and potentially higher growth rates in the younger AW, similar to *Synechococcus* (excluding the likely unrealistic loss rate obtained for RNano in the younger AW).

L269 The authors have used the term "finescale" and the rough definition appears here for the first time. From which have the authors derived this definition? We often used the term "mesoscale" to show this spatial scale in marine processes (Dickey and Bidigare, 2005, Scientia Marina). If this term was originally defined, the authors should have shown that in Introduction.

In the manuscript, we defined the term "finescale" in the Introduction (L 20-21): "ocean structures characterized by horizontal scale of the order of 1-100 km, with a short lifetime (days-weeks)". Following your comments, in the reworked manuscript we will develop this definition further.
Although several studies used the term "mesoscale", in our case "finescale" seems more appropriate. Indeed, by using this term, we include a fraction of the mesoscale processes (e.g. eddies), with scales close to the first internal Rossby radius, and the submesoscale processes, with scales smaller than the first internal Rossby radius (e.g. fronts) (Capet et al., 2008a ; Capet et al., 2008b ; McWilliams, 2016 ; Lévy et al., 2018).

L272 What are "many important oceanic processes including biogeochemical cycles and biodiversity"? Unless specified, we cannot judge whether "this suggests the possibility of a close coupling between the finescale forcing and the phytoplankton distribution and growth." Honestly, I could not understand what the authors are to discuss in this paragraph. In different water masses, phytoplankton community structures are different almost every time. We usually attribute this to different water properties that can affect phytoplankton physiology, including salinity, temperature, turbidity, and nutrient concentrations, rather than to temporal and/or spatial scales of physical processes. I am afraid that there may be a large discrepancy between the final goal of this (overall) study and possible conclusions extracted from the present results.

We agree that phytoplankton is affected by water masses properties. In this part, we were not clear enough, we propose to reformulate these sentences as follows:
"The temporal scale of finescale processes (days-weeks) is of the same magnitude as biogeochemical processes and phytoplankton cellular cycle. The rapid evolution of these

finescale structures influence the phytoplankton community, suggesting the possibility of a close coupling between finescale forcing and phytoplankton distribution and growth.”

L284 How much of the two figures (Figs. 7 and A2) was extracted from the original version in Tzortzis et al. (2021)? If it is a copy of the original, the authors should not use it again but should just cite it. And the authors say “in the frontal area upwellings and downwellings occur with different intensities”, but I think that it is not reflected in Fig. 7. From this figure, I could not find any difference in the vertical velocity of the two water masses.

Figures 2 and A2 were indeed extracted from Tzortzis et al. (2021). Following your suggestion and those of the first referee, we will move these figures into supplementary information. Concerning Figure 7, we have adapted this figure from Tzortzis et al. (2021), which is why we think that it should stay in the manuscript. Following the suggestions of the other referee, we have reworked this figure (see below) for clarity.

[Figure]

Figure 7: The contrasted distribution of phytoplankton in the frontal area. The circles represent the abundances of the several phytoplankton groups in the two water masses separated by the front. The boxes indicate the biovolume observed ($\bar{v}_{obs}$) and the growth rates ($\mu_{size}$) for each phytoplankton group, as estimated from the model. Figure adapted from Tzortzis et al. (2021).

L286 The authors intended to say "spatial", not "special"? Even if so, the authors did not show "spatial" distribution in this paper. They just showed "temporal variations" in phytoplankton populations while covering two water masses.

We intended to say "special", but "particular" is more appropriate.

L289 "high phytoplankton size" is not an appropriate term.

Thank you for your corrections, maybe "the largest cells" is better.

L290 "picophytoplankton are more abundant in oligotrophic regions". This is a problematic description. First, it is true that the proportion of picophytoplankton in the total phytoplankton biomass becomes higher in the oligotrophic region compared with that in the mesotrophic or eutrophic regions. However, the absolute biomass or abundance of picophytoplankton is not always higher in the oligotrophic area. Generally speaking, *Prochlorococcus*, which are adapted to ultraoligotrophic environments, are most abundant in oligotrophic waters. However, *Synechococcus* and eukaryotic picophytoplankton are more abundant in the mesotrophic region. Additionally, within the narrow trophic variation of the oligotrophic regions (typically < 0.1 μM of nitrate), a higher concentration of nutrients is sometimes related to the higher abundance of these picophytoplankton populations. Because the Mediterranean Sea is widely depleted with surface nutrients, discussion is not such a simple one as "picophytoplankton are more abundant in oligotrophic regions." I admit that this description is true for the study area, as shown in previous studies (Jacquet et al., 2010; Mena et al., 2016) as well, but it is not always related to generalization.

Thank you for your analysis. We will rework this part taking into account your comments, as follows:

"In our study, the older AW is characterized by larger cells of *Synechococcus* and nanophytoplankton with low abundances, whereas the younger AW is dominated by small cells with high abundances. Furthermore, microphytoplankton (i.e largest type of phytoplankton) is more abundant in older AW than in the younger AW. A possible explanation is that these two water masses do not have the same nutrient concentration, thus favoring certain phytoplankton groups.
Bethoux (1989) and Schroeder et al. (2010) have observed that the older AW is slightly more enriched with nutrients than the younger AW because during its circulation across the Mediterranean basin, the older AW receives nutrient inputs from the continent (river discharges, rain, wind). While in our study we do not have nutrient data, we can suppose that the nutrient distribution across the two water masses should be similar to the one measured during the previous studies of Bethoux (1989) and Schroeder et al. (2010).
We propose that the enhancement in nutrient in the older AW explains the corresponding phytoplankton cell size and abundances distributions. Our hypothesis is supported by the fact that our results are in agreements with those of Jacquet et al. (2010) and Mena et al. (2016) which also found the highest abundances of the small phytoplankton (*Synechococcus* and

picophytoplankton) in the most oligotrophic waters, i.e., the younger AW. Furthermore, previous studies have shown that the proportion of picophytoplankton in the total phytoplankton biomass is higher in the oligotrophic region compared with that in the mesotrophic or eutrophic regions (Zhang et al., 2008 ; Cerino et al., 2012). Indeed, their better surface:size ratio due to their small size confers them a better capacity to inhabit areas with very low nutrient concentration compared to larger phytoplankton (Kiørboe, 1993 ; Marañón, 2015). Since our study area is always oligotrophic (Moutin et al., 2012),  a small variation of the nutrient concentration (typically < 0.1 μM of nitrate) is sufficient to generate higher abundance of picophytoplankton."

L295 "If in our study we do not have nutrient data" I do not understand the intention. Are the authors unclear whether they have nutrient data themselves?

We will modify the sentence as follows:
"Unfortunately, it was not possible for both technical and funding reasons to perform nutrient measurements during the 2018 cruise, which is why we cannot provide nutrient concentrations of both water masses. We acknowledge the importance of this information and these measurements are planned for our future cruise this year."

**Influence of the frontal system on the phytoplankton dynamics (Ln 299-339 in the reworked manuscript)**

[revised manuscript text omitted]

**Conclusion**

L305 Here the authors abandoned the trial to estimate the effects of physical processes on irradiance received by phytoplankton, but is it impossible to compare them from the results of vertical velocity in the two water masses?

Following also the suggestion of the other referee, we will modify the beginning of this section as follows:

"Previous studies have well established that vertical motions impact biogeochemistry (Mahadevan & Tandon, 2006 ; Mahadevan, 2016 ; McGillicuddy, 2016). Upward vertical velocities drive deep nutrients into the euphotic layer and also move the phytoplankton cells along the water column resulting in changing light conditions."

L309 Although the authors succeeded in estimating intrinsic growth rates of various phytoplankton populations in the two different water masses using novel methodologies, the conclusion remarks here seem too superficial and primitive. The authors did not discuss the validity or robustness of the methodology or did not discuss the interactive connections among physical fields, chemical environments, and phytoplankton growth with quantitative comparisons.

We will completely rework the Conclusion taking into account your comments and those of the other reviewer:

Phytoplankton structure and dynamics are a complex result of many interacting biological and physical phenomena. Finescale structures, and in particular fronts, generate vertical velocities which displace phytoplankton cells and nutrients in the water column, thus influencing phytoplankton communities. These mechanisms are only partially understood because the spatial scale of these structures and their ephemeral nature make them particularly difficult to study in situ; as a consequence only a few studies have been performed in finescale frontal regions. The estimates of specific growth rates for the various phytoplankton groups is one of the keys to better understand how environmental conditions affect phytoplankton dynamics. In this study, we followed the dynamics of several phytoplankton groups in two distinct water masses both in terms of hydrology and phytoplankton abundances, in order to explain their particular distribution.
The originality of our work resides in the fact that we used a size-structured population model applied in two water masses identified using a Lagrangian sampling strategy. To our knowledge this had never been done before. This strategy allowed us to reconstruct the diurnal cycle of several phytoplankton groups and to identify contrasted dynamics in the two water masses. For *Synechococcus* and nanophytoplankton, we found higher cell size in the older AW located north of the front, associated with lower abundances. A possible explanation is that the older AW is more enriched in nutrients than the younger AW, thus favoring larger cells. This remains a hypothesis because of a lack of nutrient data. Another novelty of our study is that we applied the Sosik et al. (2003) model on several phytoplankton groups identified by flow cytometry, whereas previous studies only applied it to *Synechococcus* and *Prochlorococcus* (Ribalet et al., 2010; Hunter-Cevera et al., 2014; Marrec et al., 2018; Fowler et al., 2020) or to certain types of diatoms (Dugenne et al., 2014). We obtained good results for *Synechococcus* and nanophytoplankton. However, our results were noisy for picophytoplankton groups probably because they contain several taxa with differing dynamics (Siokou-Frangou et al., 2010; Le Moal et al., 2011).
Our work paves the way for many research perspectives. Direct integration of growth rates in biogeochemical models (Cullen et al., 1993) should be taken into account for a better assessment of the biogeochemical contribution of phytoplankton in oligotrophic ecosystems

and to better forecast its evolution in the context of global change. Furthermore, we plan future experiments again in the South Western Mediterranean in spring 2023, during the fast-sampling phase of the SWOT satellite mission which provides high resolution altimetry-derived currents. Involving high-resolution, high-precision nutrient measurements (necessary considering the oligotrophy of the Mediterranean Sea), coupled with DNA metabarcoding (to address phytoplankton biodiversity), zooplankton and virus sampling, we will improve the understanding of zooplankton grazing and viral lysis on the different phytoplankton groups. Furthermore, we aim to explore how the biogeochemical and ecological role of finescale structures in regions of weak circulation differ from those documented in highly energetic regions like boundary currents. In the Mediterranean sea, the low nutrient content is indeed the perfect condition when addressing this question, because even weak horizontal or vertical nutrient redistributions associated with the finescale circulation are likely to result in a biological response (Talmy et al., 2014; Hashihama et al., 2021).

---

## Author Response (AR2)

**Comments on the revised manuscript of "The contrasted phytoplankton dynamics across a frontal system in the southwestern Mediterranean Sea", by Roxane Tzortzis, Andrea M. Doglioli, Monique Messié, Stéphanie Barrillon, Anne A. Petrenko, Lloyd Izard, Yuan Zhao, Francesco d'Ovidio, Franck Dumas, and Gérald Gregori.**

Dear referee,

We are grateful for your interest in our study and for the attention that you have given to our work for a second time. Your suggestions and constructive comments helped us to further improve our work. We hope that our actual revised version will be accepted for publication. Below you find the point-by-point reply (in blue, with line numbers of the latest version) to your comments (in black).

**General Comments**

I found that in the revised version of "The contrasted phytoplankton dynamics across a frontal system in the southwestern Mediterranean Sea", what I pointed out has been largely incorporated. I feel a little bit disappointed that there was no additional discussion about the validity of the methodology applied, but I am largely satisfied that the logical consistency of the manuscript has been improved. In particular, the revisions made in Introduction and Conclusion have successfully clarified the location of the present study in a series of past and future expeditions conducted in the Mediterranean Sea.

Hopefully it is plausible that the authors will expand Discussion by adding more about the validation, possibility, and limitation of this methodology. This is just an option, but I believe that it will improve the current manuscript.

Thank you very much for these generally positive comments about our work.

Following your suggestions, we have reworked the Discussion (see new section 4.3 "Limitations of the study and recommendations" (L341-L368)) and the Conclusion, adding comments about the implications and the limitations of our study.

**4.3 Limitations of the study and recommendations**

[revised manuscript text omitted]

L299 "cells...grow more than one size class" This description is should be more specified.
We rewrote it as: (L161-L162) "cells of a specific phytoplankton group are unlikely to grow more than one size class over such a small time duration."

L316 "Normal" should be written in lower case.
It has been done.

L371 If the authors move the detailed description of identification of the phytoplankton functional groups by flow cytometry to appendices, they should refer to it somewhere in the main text, possibly in materials and methods.
It has been done (L128-L129).

L385 Renumber the sections.
It has been done.

L474 Section numbers should be changed.
It has been done.

L625 Although I admit that this is an important point of this study, it is not a point that has been clarified from the present results. I think that the authors had better mention it in any precedent section. It is very important to emphasize the novelty of the study in concluding remarks, but repeated mentioning on the lack of previous studies sounds redundant.
We have reworked the Conclusion, we have also removed the sentence "To our knowledge this had never been done before".